# Global Dust Optical Depth Climatology Derived from CALIOP and MODIS Aerosol Retrievals on Decadal Time Scales: Regional and Interannual Variability

Qianqian Song[1,2], Zhibo Zhang[1,2,\*], Hongbin Yu[3], Paul Ginoux[4], Jerry Shen[3,#]

1. Physics Department, UMBC, Baltimore, Maryland, USA
2. Joint Center of Earth Systems Technology, UMBC, Baltimore, Maryland, USA
3. Earth Sciences Division, NASA Goddard Space Flight Center, Greenbelt, Maryland, USA
4. NOAA Geophysical Fluid Dynamics Laboratory, Princeton, New Jersey, USA.

\*Correspondence to: Zhibo Zhang, Zhibo.Zhang@umbc.edu
# who worked as summer intern at NASA Goddard Space Flight Center during June–August 2020.

# 1. Abstract

We derived two observation-based global monthly mean dust aerosol optical depth (DAOD) climatological datasets from 2007 to 2019 with a $2° (latitude) \times 5° (longitude)$ spatial resolution, one based on CALIOP and the other on MODIS observations. In addition, the CALIOP climatological dataset also includes dust vertical extinction profiles. Dust is distinguished from non-dust aerosols based on particle shape information (e.g., lidar depolarization ratio) for CALIOP, and on dust size and absorption information (e.g., fine-mode fraction, Ångström exponent, and single-scattering albedo) for MODIS, respectively. The two datasets compare reasonably well with the results reported in previous studies and the collocated AERONET coarse mode AOD. Based on these two datasets, we carried out a comprehensive comparative study of the spatial and temporal climatology of dust. On multi-year average basis, the global (60°S-60°N) annual mean DAOD is 0.032 and 0.067 according to CALIOP and MODIS retrievals, respectively. In most dust active regions, CALIOP DAOD generally correlates well (correlation coefficient R > 0.6) with the MODIS DAOD, although CALIOP value is significantly smaller. CALIOP DAOD is 18%, 34%, 54% and 31% smaller than MODIS DAOD over Sahara Desert, the tropical Atlantic Ocean, the Caribbean Sea, and the Arabian Sea, respectively. Applying a regional specific lidar ratio (LR) of $58sr$ instead of the $44sr$ used in the CALIOP operational retrieval reduces the difference from 18% to 8% over the Sahara and from 34% to 12% over Tropical Atlantic Ocean. However, over Eastern Asia and the Northwestern Pacific Ocean (NWP), the two datasets show weak correlation. Despite these discrepancies, CALIOP and MODIS show similar seasonal and interannual variations in regional DAOD. For dust aerosol over NWP, both CALIOP and MODIS show a declining trend of DAOD at a rate of about 2% $yr^{-1}$. This decreasing trend is consistent with the observed declining trend of DAOD in the southern Gobi Desert at a rate of 3% $yr^{-1}$ and

5% $yr^{-1}$ according to CALIOP and MODIS, respectively. The decreasing trend of DAOD in the southern Gobi Desert is in turn found to be significantly correlated with increasing vegetation and decreasing surface wind speed in the area.

# 1 Introduction

Mineral dust, referred to as dust for short, is one of the most abundant type of atmospheric aerosol in terms of dry mass (Textor et al. 2006; Yu et al. 2012; Kok et al. 2017). Dust aerosol directly interacts with both solar and thermal infrared radiation, known as the direct radiative effect, and thereby influences the Earth's radiative energy budget (Kok et al, 2017; Song et al., 2018; Di Biagio et al. 2020). Dust also influences the life cycle and properties of clouds by altering the thermal structure of the atmosphere (known as semi-direct effects) (Hansen et al., 1997) and acting as cloud condensation nuclei (CCN) and ice nuclei (IN) (known as indirect effects) (Albrecht 1989; Rosenfeld and Lensky 1998; Twomey 1977). Dust storms and plumes can degrade air quality affecting human health (Griffin, 2007; Querol et al., 2019). Dust deposition provides essential nutrients to marine and terrestrial ecosystems (Jickells et al. 2005; Yu et al., 2015b) but reduces the snow albedo increasing snow melt (Painter et al., 2007). All these impacts manifest the important role of mineral dust in the Earth systems (e.g. Evan et al., 2006; Lau & Kim, 2007; Miller & Tegen, 1998; Shao et al., 2011)

Dust production is sporadic in nature. Dust aerosol can be transported on intercontinental, hemispherical, and even global scales (Grousset et al. 2003; Uno et al. 2009; Yu et al. 2012, 2013). Thus, global and routine measurements of dust spanning over years or even decades are vital for studying dust transport and deposition, estimating the dust radiative effects, and evaluating and constraining dust simulations in numerical weather and climate models. Satellite remote sensing

is the only means to observe dust on regional to global scales. Satellite remote sensing techniques usually retrieve the optical depth or extinction profile for total aerosol in the atmosphere with additional retrievals of particle size, shape, or absorption properties that are sensor specific. Passive sensors have been used to detect dust sources and track dust plumes at global scales. A few examples are the Total Ozone Mapping Spectrometer (TOMS) (Prospero et al., 2002), Ozone Monitoring Instrument (OMI) (Chimot et al. 2017), Multi-angle Imaging SpectroRadiometer (MISR) (Ge et al., 2014 and Y. Yu et al. 2019), Moderate Resolution Imaging Spectroradiometer (MODIS) (Ginoux et al., 2010; Remer et al.,2005; Yu et al., 2009), multi-angular and polarimetric POLarization of Directionality of the Earth's Reflectances / Polarization and Anisotropy of Reflectances for Atmospheric science coupled with Observations from a Lidar (POLDER/PARASOL ) measurements (Chen et al. 2018) and the International Association of Structural Integrators (IASI) (Klüser et al., 2011; Clarisse et al. 2019). On one hand, these passive sensors provide global or quasi global coverage of column integrated properties of aerosol with satisfactory temporal resolution. On the other hand, they do not provide the vertical structure of aerosol that is critical for studying aerosol-cloud interactions and aerosol influences on the thermal structure of the atmosphere. Space-borne lidar systems, such as the Cloud-Aerosol Lidar with Orthogonal Polarization (CALIOP) onboard the Cloud-Aerosol Lidar and Infrared Pathfinder Satellite Observation (CALIPSO) spacecraft (Winker et al., 2010) and the Cloud-Aerosol Transport System (CATS) onboard the International Space Station (Yorks et al. 2015) are able to provide the vertical structure of aerosol and clouds, albeit with limited spatial coverage. All these passive and active remote sensing observations have been used extensively in studies of the spatial and temporal evolution of aerosol over the past decade (e.g., Proestakis et al. 2018).

A significant hurdle of applying satellite remote sensing measurements for dust studies is how to distinguish dust from other aerosol types in a quantitative way. While many studies have used total aerosol retrievals by focusing on the regions and seasons where dust dominates, some studies have developed sensor-specific methods of partitioning total aerosol into dust and non-dust components with varying assumptions (Kaufman et al., 2005; Kalashnikova et al. 2005; Dubovik et al. 2006; Ginoux et al., 2010; Yu et al., 2009, 2013, 2015a, 2019). In general, the dust separation methods are based on dust physical and optical properties such as their large size, their irregular or non-spherical shape, and absorption characteristics. For example, CALIOP dust classification is mainly based on the fact that dust aerosols are non-spherical in shape and their lidar depolarization ratio is significantly larger than those spherical aerosols. In contrast, the wide spectral coverage of MODIS measurements enables the retrieval of aerosol particle size information, such as effective radius, fine-mode fraction (FMF), and aerosol extinction Angstrom exponent, as well as spectral gradient of absorption (decreasing of absorption from UV to red) (Remer et al., 2005). The combinations of these retrievals provide the basis for dust separation and DAOD retrievals from MODIS. Some recent studies have also characterized dust distribution through integrating satellite measurements with other data sources and model simulations. For example, Voss and Evan (2020) (referred to as VE20 hereafter) developed a dust optical depth record from MODIS retrievals, similar to Kaufman et al. (2005) over ocean and Ginoux et al. (2012) over land. Unlike Kaufman et al. (2005) and Yu et al. (2020) that derived characteristic FMF values for combustion, dust, and marine aerosol from MODIS retrievals, VE20 determined these characteristic FMFs from AERONET measurements. VE20 also extended the MODIS-based method to AVHRR over-ocean retrievals with some assumptions and produced the long-term (1981-2018) record of dust optical depth. Gkikas et al. 2021 developed a global fine resolution

(0.1° x 0.1°) DAOD dataset for the period 2006-2017 by scaling MODIS retrieved Collection 6.1 Aerosol Optical Depth (AOD) with the DAOD-to-AOD ratios provided by MERRA-2 (Modern-Era Retrospective analysis for Research and Applications, Version 2) reanalysis (Gelaro et al. 2017). Given that MODIS and other remote sensing measurements (e.g., MISR and AERONET) have been assimilated in the MERRA-2 reanalysis to constrain the aerosol optical depth, the DAOD-to-AOD ratio reported by MERRA-2 is the same as that from the underlying GOCART aerosol transport model in the MERRA-2 reanalysis system.

In this study, we focus on the DAOD derived from CALIOP and MODIS with two major objectives. First, we produce a decadal (2007-2019) record of global DAOD and dust vertical extinction coefficient profile climatology from the CALIOP observations, which represents an extension of the trans-Atlantic dust transport and deposition studies by Yu et al. (2015a, 2015b, 2019), both in terms of spatial and temporal coverages. Second, we compare the CALIOP DAOD climatology with the MODIS DAOD over both land and ocean (Yu et al. 2020; Pu and Ginoux, 2018) to identify and understand their differences in terms of global dust distribution and interannual variabilities including trend in key dust regions. Our analysis goes beyond broad dust-laden regions by zooming into potential dust source areas, which provides important insights into local dust activities. A systematic comparison and better understanding of DAOD from the two sensors based on distinct retrieval algorithms is critical for applying satellite measurements to evaluate global dust modeling (Kim et al. 2019). In comparison to some most recent studies (Voss and Evan, 2020; Gkikas et al. 2021), our dust climatology is derived using the satellite observations in a self-consistent way without blending in other measurements (e.g., AERONET) or models (e.g., MERRA-2). As discussed in Yu et al. (2009), the self-consistent use of MODIS data could

minimize the introduction of additional biases due to discrepancies in FMF between MODIS and AERONET. Furthermore, we use the latest version 4.2 CALIOP products and version 6.1 MODIS products to characterize the spatial and temporal distributions of dust. The rest of the paper is organized as follows. Section 2 provides a description of the methodology of deriving dust climatology from CALIOP and MODIS. In Section 3, we compare our DAOD datasets with previous studies and collocated AERONET retrievals. In Section 4, we compare and study the DAOD climatology from CALIOP and MODIS. Section 5 provides a summary of the study along with the main conclusions.

## 2 Dust Detection and AOD Partition Schemes

### 2.1 CALIOP Dust Detection and AOD Partition

CALIPSO is in a sun-synchronous polar orbit with an equator crossing time of around 13:30 local time and 98° orbit inclination. CALIOP is a two-wavelength (532nm and 1064nm) polarization-sensitive lidar onboard CALIPSO. CALIPSO orbit track repeats every 16 days, CALIOP sensor never provides global coverage due to its small footprint. At Earth's surface, the diameter of CALIOP footprint is around 70m, with spacing distance of 333m between two adjacent footprints along the orbit track. CALIOP utilizes three receiver channels (one measuring the 1064nm backscatter intensity and two measuring orthogonally polarized components of the 532nm backscatter) to provide high vertical resolution 30-60m of aerosol and cloud structure profiles (Winker et al., 2009).

Aerosol subtype classification and a priori assumption of LR (extinction to backscatter ratio) for specific aerosol type are critical for CALIOP aerosol retrievals. CALIOP Level 2 product has

been validated by comparing with ground-based measurements. The comparison between aerosol subtypes in CALIOP level 2 V2.01 and NASA Aerosol Robotic Network (AERONET) aerosol types shows that 70% of the CALIOP and AERONET aerosol types are in agreement and best agreement is achieved for dust and polluted dust (Mielonen et al. 2009). Schuster et al. (2012) compared CALIOP AOD to the collocated AERONET AOD measurements and found a CALIPSO bias of $-13\%$, corresponding to an absolute bias of $-0.029$ relative to AERONET AOD on global average. Further comparison between CALIPSO AOD measurements and the collocated AERONET AOD measurements for the columns that contain the dust subtype exclusively showed a larger bias (i.e., $-29\%$ and corresponding absolute bias of $-0.1$), although they show a relatively high correlation of R=0.58; this indicates that the assumed LR of 40 $sr$ for the CALIPSO dust retrievals is too low. Omar et al. 2013 showed that CALIOP AOD are lower than AERONET AOD especially for low AOD. Furthermore, they found that the median of relative AOD difference between CALIOP and AERONET (500nm) is 25% of AERONET AOD for AOD $> 0.1$.

CALIOP observations have been used widely in previous studies of the spatial and temporal evolution of dust aerosols over the past decade (Huang et al. 2007, 2008; Yang et al. 2012; Xu et al. 2016; Kim et al., 2019). It is important to note that these studies are regional in scope and they use the standard CALIPSO product and aerosol subtype classification algorithm (Omar et al. 2009). In the standard CALIPSO product, each detected aerosol layer is classified as one of the six subtypes: dust, polluted dust, polluted continental, smoke, clean marine and clean continental. In the latest CALIOP version, another sub-type "marine-dust" is introduced (Kim et al. 2018). In these studies, the "dust" subtype or a combination of "dust" and "polluted dust" subtypes is

categorized as dust. While the former assumption leads to an underestimate of dust due to neglecting dust component in the "polluted-dust" subtype, the latter assumption results in an overestimate of dust because of accounting for non-dust component in the "polluted-dust" subtype. In order to better distinguish dust component from each CALIOP detected aerosol layers, Yu et al. (2015a) developed an algorithm independent of the standard aerosol subtype classification to distinguish dust from non-dust aerosol by using their respective thresholds of particulate depolarization ratio (Table 1). The depolarization-based dust separation algorithm is based on the method developed by Shimizu et al. 2004, Hayasaka et al. 2007 and Tesche et al. 2009. The algorithm has been implemented in the framework of surface lidar network such as European Aerosol Research Lidar Network (EARLINET) (Ansmann et al. 2011) and also applied to CALIOP observations (Yu et al., 2012; Amiridis et al. 2013; Yu et al., 2015a). They further used the derived three-dimensional distribution of dust extinction to quantify the trans-Atlantic dust transport and deposition and its implications for Amazon rainforest (Yu et al., 2015b, 2019).

In this study, we use the methodology in Yu et al. (2015a) to derive the monthly mean dust extinction profile under clear-sky conditions from the latest V4.20 CALIOP products on a global scale from 2007 to 2019. First, we select the cloud-free columns based on the CALIOP cloud layer product. In order to increase the sampling, we define clear-sky cases in this study either as columns that are completely cloud-free or with the presence of optically thin (cloud optical depth < 0.2) and high-level (cloud base > 7km) clouds. This is justified that the presence of high-level optically thin clouds does not significantly affect the retrieval of aerosol layers below the clouds (Yu et al. 2015a). After clear-sky screening, we use the operational 5 km level 2 CALIOP aerosol profile product that contains aerosol depolarization, backscatter and extinction profiles over a global scale

(Young et al. 2018) to derive the dust extinction profile. The depolarization ratio from CALIOP is a key variable for detecting and distinguishing dust from non-dust aerosol. Backscatter by spherical particle largely retains the polarization of the incident light, resulting in a depolarization ratio of nearly zero. In contrast, dust particles are generally non-spherical in shape and large in size, which gives them non-zero depolarization ratio that is significantly larger than other types of aerosol. The cloud-aerosol discrimination (CAD) score in the products gauges the level of confidence for a feature being classified as aerosol or cloud. In this study, in order to screen out low-confidence aerosol and cloud discrimination, we select layers with CAD scores between −90 and −100 (high level of confidence for aerosol feature) by following Yu et al. (2019). Aerosol profile product also provides extinction quality control flag (Ext_QC) to indicate problematic retrievals. This study only uses layers with Ext_QC values of 0, 1, 18, and 16 (Winker et al., 2013). Only nighttime data are used to avoid sunlight interference in aerosol signals.

For each aerosol backscatter coefficient profile, we derive the fraction of dust backscatter to total backscatter ($f_d$) at each altitude from the following equation

$$f_d = \frac{(\delta - \delta_{nd})(1 + \delta_d)}{(\delta_d - \delta_{nd})(1 + \delta)} \, ,$$

(1)

where $\delta$ is CALIOP observed particulate depolarization ratio, $\delta_d$ and $\delta_{nd}$ is a priori knowledge of depolarization ratios of dust and non-dust aerosols respectively. Clearly, the calculations of $f_d$ in Eq. (1) rely on the a priori depolarization ratios of dust and non-dust aerosols (i.e., $\delta_d$ and $\delta_{nd}$). To account for various types of non-dust aerosols with different depolarization ratio, we follow Yu et al. 2015a and assume 0.02 and 0.07 as lower and upper bounds for $\delta_{nd}$ (Burton et al., 2012; Fiebig et al., 2002; Sakai et al., 2010). Dust aerosols have significantly larger depolarization ratio compared to non-dust aerosols. To account for the variability of dust shape and size, we use 0.2 and 0.3 as lower and upper bounds for $\delta_d$ (Ansmann et al., 2012; Esselborn et al., 2009; Sakai et

al., 2010). Given an observed dust depolarization ratio $\delta$, the $f_d$ based on Eq. (1) has the minimum value when $\delta_d = 0.30$ and $\delta_{nd} = 0.07$ and the maximum value when $\delta_d = 0.20$ and $\delta_{nd} = 0.02$. To account for this variability, the final $f_d$ is based on the mean of the lowest (i.e., $\delta_d = 0.30$ and $\delta_{nd} = 0.07$) and the highest (i.e., $\delta_d = 0.20$ and $\delta_{nd} = 0.02$) dust scenario.

In each 2º (latitude) ×5º (longitude) grid, at each altitude, dust backscatter coefficient for per clear-sky overpass is derived by multiplying CALIOP total backscatter coefficient with the calculated $f_d$ from Eq. (1). To derive dust extinction coefficient from dust backscatter coefficient, we assume dust LR, i.e., extinction to backscatter ratio, of $44 \pm 9 sr$ at 532nm, consistent with CALIOP Version 4.20 operational retrieval (Kim et al., 2018). The monthly mean dust extinction coefficient is calculated at each altitude when overpass samples within the month is larger than 5. Then DAOD is calculated by integrating the monthly mean extinction coefficient profile for each grid. The use of globally uniform LR and the selection of $\delta_d$ and $\delta_{nd}$ could induce uncertainty to the derived DAOD. This is discussed in section 3.

It is important to note that in this study we use only nighttime CALIOP observations for DAOD retrievals. This is because the daytime CALIOP observations are often contaminated by background solar noise (Getzewich et al. 2018). As shown in Figure S1 in the supplementary material, when the above DAOD retrieval method is applied to daytime CALIOP observation, there is a widespread non-zero DAOD retrieval over remote ocean regions where dust should be scarce. This is apparently an artifact caused by solar contamination on CALIOP daytime observations, which motivates and justifies our use of nighttime CALIOP observations. On the other hand, however, this leads to an inconsistency with the MODIS DAOD retrieval which is based on daytime observations (see section 2.2). Although the diurnal cycle of dust has been

investigated using model simulations (e.g., Yue et al. 2009), it is extremely difficult to assess dust diurnal variation from polar orbiting remote sensing observations, especially using elastic lidar in visible region like CALIOP, due to the inherent instrument limitation. For example, a recent study by Yu et al. 2021 attempted to use the retrievals from the Cloud-Aerosol Transport System (CATS) lidar to study the diurnal cycle of dust. The 51.6-degree inclination orbit allows CATS to sample the tropical and midlatitude regions multiple times a day, which make it more advantageous than CALIOP for diurnal variability studies. Unfortunately, after a validation comparison with AERONET observations (i.e., solar-based during daytime and lunar-based during nighttime), they found a significant day–night inconsistency in their retrieval quality. Because of this inconsistency, they concluded that diurnal variability in dust and dust mixture characteristics have to be examined separately for daytime and nighttime periods. Nevertheless, Yu et al. 2021 plotted the daytime and nighttime DAOD together for several dust active regions (see their Figures 3 and 10-13). The contrast between daytime and nighttime DAOD based on these plots is roughly between 10-15%, which is smaller than other uncertainties in CALIOP retrievals as analyzed in section 3. Again, it has to be emphasized that this contrast is partly due to the day–night inconsistency in CATS data quality.

## 2.2  **MODIS Dust Detection and AOD Partition**

As described above, the CALIOP-based DAOD derivation mainly makes use of dust non-sphericity in shape to separate dust aerosol from others. Another important difference of dust aerosol from other types of aerosols is their relatively large size. This difference provides the basis for the dust separation. DAOD derivation scheme based on the Moderate Resolution Imaging Spectroradiometer (MODIS) retrievals is introduced in this section.

MODIS sensors onboard of the Aqua and Terra satellites measure radiances at 36 spectral bands ranging from 0.41 to 14 $\mu m$, with a 2330 km swath that provides near-global coverage every day. MODIS aerosol retrievals employ two complementary algorithms to achieve the global coverage. The Dark Target (DT) algorithm is applicable for the retrieval of aerosol loading and properties over dark surfaces, including ocean-water and vegetated land. The MODIS aerosol AOD retrievals over ocean are found within the retrieval errors of $\Delta\tau_a = \pm 0.03 \pm 0.05\tau_a$ relative to AERONET AOD measurements (Remer et al. 2005). An approach was developed in previous studies to separate DAOD from other types of aerosol by using aerosol optical depth ($\tau$) and fine mode fraction ($f$) retrieved from MODIS DT retrieval over ocean. Both $\tau$ and $f$ refer to properties at 550nm hereafter, unless specified otherwise. In this approach, both $\tau$ and fine-mode AOD ($f\tau$) are assumed to be composed of marine aerosol, dust and combustion aerosols, i.e.,

$$\tau = \tau_m + \tau_d + \tau_c \ ,$$
(2)

$$f\tau = f_m\tau_m + f_d\tau_d + f_c\tau_c \ ,$$
(3)

Where the subscripts m, d, and c represent marine aerosol, dust and combustion aerosol, respectively. Based on Eq. (2) and (3), $\tau_d$ can be calculated from MODIS-retrieved $\tau$ and $f$, with appropriate parameterizations for $f_m, f_d, f_c$ and $\tau_m$. More specifically, $f_m, f_d, f_c$ were determined from retrieved $f$ in selected regions and seasons for which a specific aerosol type dominates, $\tau_m$ was parameterized as a function of wind speed (details can be found in Kaufman et al. 2005; Yu et al., 2009, 2020).

Over land, MODIS aerosol properties including AOD, Ångström exponent, SSA are retrieved from the Deep Blue (DB) algorithm (Hsu et al. 2004, 2013). The MODIS aerosol AOD retrievals over land are found within the retrieval errors of $\Delta\tau_a = \pm 0.05 \pm 0.15\tau_a$ relative to AERONET

AOD measurements (Remer et al. 2005). DAOD over land is derived from the AOD using one criterion based on size distribution (to distinguish fine and coarse modes) and the other criterion based on absorption (to distinguish between scattering sea salt and absorbing dust). To apply first criterion, we use the following formula established by Anderson et al. 2005 using in-situ data:

$$COD_M = AOD \times (0.98 - 0.5089\alpha + 0.051\alpha^2) \ ,$$

(4)

Where $\alpha$ is the Ångström exponent (a measure of the wavelength dependence of optical depth) which has been shown to be highly sensitive to particle size (Eck et al. 1999), $COD_M$ is the coarse mode fraction (aerodynamic diameters larger than $1\mu m$) of AOD retrieved from MODIS, with a contribution from absorbing (DAOD) and scattering aerosols (sea salt aerosol optical depth). The second criterion requires the single-scattering albedo at 470nm to be less than 0.99 for the retrieval of DAOD (more details can be found in Pu and Ginoux, 2018).

Overall, multi-wavelength observations from MODIS contains aerosol size information such as fine-mode fraction and Ångström exponent in the observed reflectance spectral pattern, which was used to separate dust aerosol from others in MODIS dust retrieval over ocean and land (Table 1). In this study, the latest retrieved aerosol properties from MODIS Collection 6.1 are used. We use data from Aqua MODIS only, because Terra MODIS retrievals may generate spurious dust trend (Yu et al. 2020). In order to minimize cloud contamination and avoid the infrequent sampling to bias DAOD in MODIS dust retrieval over ocean, we screen the data by requiring a minimum of 10 DAOD retrievals in a month.

The relevant variables and the quality assurance procedures used in CALIOP- and MODIS-based DAOD retrievals are summarized in Table 1 and Table S1, respectively.

# 3  Comparison with previous studies and Uncertainty Analysis

Based on the dust detection and separation schemes of two sensors described in section 2, we derived the following three datasets:

1.  The monthly mean CALIOP-based total aerosol optical depth (TAOD) and DAOD, as well as the vertical extinction profile on a 2º (latitude) ×5º (longitude) spatial resolution grids for the period of 2007 – 2019. This relatively coarse resolution is limited by CALIOP's sampling.

2.  We combine the monthly mean Aqua MODIS over-ocean (Yu et al., 2020) and over-land (Pu and Ginoux, 2018) TAOD and DAOD on a 1º ×1º spatial resolution grids to get the monthly mean MODIS-based TAOD and DAOD from 2003 to 2019. In order to compare with CALIOP-based dust climatology data, we aggregate the 1º×1º MODIS-based data to 2º×5º resolution grids.

3.  For evaluation and comparison purpose (see section 4.1), we also produce a seasonal global distribution of conditionally sampled DAOD from CALIOP (Marinou et al. 2017, Proestakis et al. 2018). While the standard climatological DAOD includes all cloud-free cases in the average of dust extinction and DAOD regardless of the presence of dust, the conditionally sampled DAOD calculation only averages those cases where dust is detected (i.e., DAOD and dust extinction are non-zero). Therefore, the conditionally sampled DAOD is directly related to the intensity of the detected dust events, whereas the climatological DAOD is determined by a number of factors including not only the intensity of the detected dust events but also the frequency of the dust events as well as the capability of the instrument to sample the dust events.

## 3.1  Comparison with previous studies

Before we compare and study the DAOD climatology from MODIS and CALIOP in detail in the next section, we first evaluate our retrievals through comparisons with the regional and global DAOD values reported in the previous studies and explore the potential reasons for the differences.

Table 2 summarizes a comprehensive comparison of our DAOD datasets with previous studies. In Ridley et al. 2016, DAOD is first estimated in 14 dust-laden regions from the combination of AERONET measurements, MODIS and MISR retrievals. Then the observation-based, regional DAOD estimates are estimated to the global scale based on the model-estimated regional-to-global DAOD ratio. Using this method, they estimated that the global (90°S~90°N) DAOD@550nm is $0.03 \pm 0.005$. Using the DAOD-to-AOD ratio from MERRA-2, Gkikas et al. 2021 converted the MODIS AOD retrievals to DAOD and found a similar global (90°S~90°N) DAOD@550nm around 0.033. In contrast, as shown in Table 2 our MODIS-based global (90°S~90°N) DAOD is 0.057. However, it is important to note that the global mean DAOD values from these studies are not directly comparable to our global mean results because of the methodology differences. In particular, both of aforementioned studies used model simulations to aid their global DAOD estimate, while our estimates are completely based on observations (More precisely, DAOD of the scope 60°S~60°N are completely based on observations, while outside of the scope, DAOD is assumed to be zero). Nevertheless, to gain a more insightful understanding of the differences, we select the same 14 dust-laden regions as in the Ridley et al. 2016 (see Figure S2 in the supplementary material) and derive the corresponding regional DAOD (see Figure S3 and Table S2 in the supplementary material). As aforementioned, in Ridley et al. 2016 the DAOD in these dust-laden regions is based on AERONET measurements and satellite retrievals, and

therefore more comparable with our results. As shown in the supplementary material (Figure S3), our regional MODIS-based DAOD values are in excellent agreement with those reported in Ridley et al. 2016 (relative bias Br $= -5.8\%$ in DJF, $-0.2\%$ in MAM, $-2.5\%$ in JJA and $-10.4\%$ in SON). This regional comparison suggests that the difference in global DAOD between our study and Ridley et al. 2016 is probably because we used different methods to derive the DAOD in the regions with less frequent dust activities (i.e., observation-based vs. model-based).

Recently, VE20 used a method similar to our MODIS-based DAOD estimate methodology to derive the global DAOD. Because of the use of similar methodology and data, VE20 is more comparable to our study than Ridley et al. 2016 or Gkikas et al. 2021. They estimated that the long-term mean DAOD to be 0.1 over land between $50^{\circ}$S and $60^{\circ}$N, which is almost identical to our estimate of 0.103 ($60^{\circ}$S $\sim 60^{\circ}$N) as shown in Table 2. However, when averaged over the ocean, their DAOD estimate (0.03 $\pm$0.01) is significantly smaller than our result (0.055). As explained in the supplementary material, this difference is probably because different parameterizations of $f_m, f_d, f_c$ and $\tau_m$ in Eq. (3) used in the two studies (see Table S4 and discussions in supplementary material).

A recent study by Proestakis et al. 2018 used a method similar to ours as described in section 2.1 to derive CALIOP-based regional DAOD in five dust-laden regions in Asia. We compared our CALIOP-based regional DAOD for the same regions (Figure S4) and compare the results with the values reported in Proestakis et al. 2018. As shown in Figure S5 of the supplementary material, the two studies are in excellent agreement with relative difference Br $= 5.5\%$ in DJF, $-6.0\%$ in MAM, $-6.9\%$ in JJA and 0.8% in SON, respectively.

Overall, the above comparisons indicate that our DAOD retrievals are in reasonable agreement with previous studies (where directly comparable). However, none of the aforementioned previous studies performed a systematic comparison between MODIS- and CALIOP-based DAOD, which is one of the motivations for this study and will be addressed in the Section 4.

## 3.2 Uncertainty Analysis

In order to understand the differences between the MODIS- and CALIOP-based DAOD, it is important to identify and quantify the uncertainties in each retrieval. The uncertainty of CALIOP DAOD retrieval come from several sources: An important source is the inherent uncertainty associated with CALIOP observations and its retrieval algorithm, such as instrument calibration errors (Kar et al. 2018), errors in discriminating cloud from aerosol and failure to detect aerosol layers (including tenuous aerosol layer and the lower part of heavy dust layer. For example, Thorsen and Fu (2015) estimated that CALIOP may have underestimated 30%-50% in the magnitude of aerosol direct radiative effect due to its low sensitivity to tenuous layer), which is likely to translate into low bias in DAOD. In heavy aerosol conditions (e.g., strong dust storms in source regions and outflow regions), CALIOP laser cannot penetrate to the bottom of aerosol layer due to the laser attenuation (Rajapakshe et al., 2017), which could also lead to a low bias in CALIOP DAOD.

CALIOP-based DAOD is also subject to the uncertainty associated with the assumed dust LR. Different deserts produce dust with different minerology, size and shape, and thus different LRs. Voss et al., (2001) measures LR for African dust as $41 \pm 8$ sr using a micropulse lidar system

and Liu et al. (2002) measures LR for Asian dust as 42-55 sr. Globally observed LRs are summarized in Müller et al., (2007) and Baars et al., (2016). In this study, we assume dust LR to be $44 \pm 9\ sr$ at 532nm to be consistent with the value used in the CALIOP V4 product (Kim et al. 2018). This LR range is also comparable to previous studies and basically covers the range of typical dust LRs from $35\ sr$ to $55\ sr$ (Muller et al. 2007, Baars et al. 2016). The $\pm 9\ sr$ induces $\pm 20\%$ DAOD uncertainties. When separating dust from non-dust aerosol, the choice of depolarization ratio (DPR) for dust aerosols and non-dust aerosols also introduces uncertainty in DAOD. To quantify the uncertainty caused by DPR selection, we also calculated DAOD in the lowest ($\delta_d = 0.30$ and $\delta_{nd} = 0.07$) and the highest ($\delta_d = 0.20$ and $\delta_{nd} = 0.02$) dust fraction scenarios. The uncertainty induced by DPR is region dependent (Figure S6). The uncertainty is much lower in dust dominant regions than other regions. The averaged uncertainty for regions with DAOD>0.05 is 20%, while the averaged uncertainty for other regions is 38%.

MODIS dust detection is also subject to a number of uncertainties. Over ocean, the persistent presence of clouds, especially broken clouds, poses a great challenge to the MODIS aerosol retrievals (Martins et al. 2002). If a cloud is mistaken as aerosol, it would lead to a high AOD and low FMF bias, and thereby a high DAOD bias. In addition, DAOD was calculated from the MODIS-retrieved AOD ($\tau$) and FMF ($f$) with appropriate parameterizations of marine aerosol AOD ($\tau_m$), FMF of dust ($f_{dust}$), combustion ($f_c$) and marine ($f_m$) aerosols (see Table 2 in Yu et al. 2020 for the parameterization values). All the parameterizations could also introduce uncertainty in the derived DAOD, in particular on a regional basis (see details in Yu et al. 2020). Over land, the derived MODIS DAOD represents the coarse-mode fraction (aerodynamic diameters larger than $1\mu m$) of dust only. The exclusion of submicron dust aerosol could induce

around 3% underestimation of the global atmospheric dust mass load and around 15% underestimation of the global DAOD (see Figure S1 in Kok et al. 2017).

One way to evaluate these uncertainties and validate the two dust detection methods is to compare with an independent measurement of DAOD. AERONET measurements have been considered as ground truth and often used to evaluate satellite aerosol optical depth retrievals. However, so far there is not a valid method to derive DAOD from AERONET AOD measurements to compare our results with. Therefore, we use coarse-mode AOD (COD) from AERONET measurements as a proxy for DAOD (Pu and Ginoux, 2018) to compare with our DAOD datasets and further estimate the absolute expected errors (EE) associated with our DAOD datasets. The fine mode and coarse mode AOD in AERONET product are defined optically, rather than in terms of a microphysical cutoff of the associated particle size distribution at some specific radius (see details in O'Neill et al. 2003). Over land especially dust source regions, dust aerosols are predominantly in coarse mode, therefore, AERONET COD could be considered as a good proxy of DAOD over land. Over ocean, the exclusion of fine mode DAOD could be partially cancelled by the inclusion of coarse sea salt AOD in AERONET COD retrievals. Therefore, AERONET COD is considered as a proxy of DAOD over ocean as well.

We use AERONET monthly mean COD retrieved at 500nm from the level 2 (cloud screened and quality assured) Spectral Deconvolution Algorithm (SDA) version 4.1 in this study. The AERONET COD is converted to 550nm and 532nm using Angstrom Exponent to compare with MODIS and CALIOP DAOD retrievals, respectively. In addition, we produce a finer resolution (1° × 1°) CALIOP-based DAOD retrieval to compare with AERONET COD.

For overland dust retrievals, between 2007 and 2019, there are 16653 MODIS, CALIOP monthly mean DAOD retrievals collocated with 761 AERONET sites located within a 1-degree MODIS and CALIOP grid cell (Figure 1). MODIS DAOD (DAOD$_M$) overall bias high compared to AERONET COD with absolute bias $B_a = 0.01$, and relative bias $B_r = 26.7\%$. While CALIOP DAOD (DAOD$_C$) generally bias low with $B_a = -0.02$ and $B_r = -27.9\%$. Using a methodology suggested in Sayer et al. 2013, the estimated EE (take 68[th] percentiles referring to Sayer et al. 2013) for all collocated MODIS DAOD over land is approximately 0.65×DAOD$_M$+0, and for CALIOP DAOD over land is approximately 0.52×DAOD$_C$+0.02 (Figure 2).

For over-ocean dust retrievals, between 2007 and 2019, there are 7755 MODIS, CALIOP monthly mean DAOD retrievals collocated with 311 AERONET sites located within a 1-degree MODIS and CALIOP grid cell (Figure 3). MODIS DAOD overall bias high compared with AERONET COD with absolute bias $B_a = 0.01$, and relative bias $B_r = 18.1\%$. While CALIOP DAOD generally bias low with $B_a = -0.02$ and $B_r = -35\%$. The estimated EE for all collocated MODIS DAOD over land is approximately 0.50×DAOD$_M$+0, and for CALIOP DAOD over land is approximately 0.54×DAOD$_C$+0.02 (Figure 4).

We further analyze the statistical parameters and EE by continents for MODIS and CALIOP DAOD (Table 3). The lowest EE, $B_r$ and highest correlation (R) are estimated over Africa, followed by Asia, Europe, Americas and Australia. This implies that our DAOD retrievals are subject to higher bias under high AOD in polluted regions. Overall, MODIS-based monthly mean DAOD retrievals are larger than AERONET COD measurements, while CALIOP-based

DAOD retrievals are smaller than AERONET COD, which seems to suggest that the true DAOD fall between the MODIS and CALIOP DAOD products.

# 4   Global Dust Climatology

In this section, we compare CALIOP global dust retrieval against MODIS dust retrieval, more specifically MODIS ocean dust retrieval from Yu et al. (2009, 2020) and land dust retrieval from Pu and Ginoux (2018), we analyze the similarities and differences between two dust climatological datasets and furthermore study the seasonal cycle and trend of dust aerosols based on these datasets.

## 4.1   Comparison between CALIOP and MODIS DAOD Climatology

The DAOD climatology datasets derived from the CALIOP and MODIS observations, as described in Section 3, have two major sources of uncertainty:

1) The uncertainty associated with the TAOD retrieval. The primary uncertainty sources in MODIS TAOD retrieval include instrument calibration errors, cloud-masking errors, inappropriate assumption of surface reflectance and aerosol model selection (Remer et al. 2005; (Levy et al. 2013, 2018). Uncertainty sources in CALIOP aerosol retrieval include instrument calibration errors, errors in discriminating cloud from aerosol, uncertainties associated with the a priori assumption of LRs, under detection of tenuous aerosol layers, and overestimation of the elevation height of heavy aerosol plume base (Winker et al. 2009; Yu et al., 2010; Schuster et al., 2012; Thorsen and Fu, 2015; Rajapakshe et al. 2017).

2) The uncertainty associated with dust detection and separation. As explained in section 2, CALIOP- and MODIS-based dust detection and separation methods are based on different

characteristics of dust aerosols in comparison with other types of aerosols, as summarized in Table 1. The CALIOP-based method makes use of the fact that depolarization ratio of dust aerosols is much higher than other types of aerosols, primarily because of irregular non-spherical shape and also to a lesser extent because of coarse size of dust particles (Gasteiger et al. 2011, Järvinen et al. 2016). MODIS-based method is largely based on the characteristics of coarse particle size. Over ocean, DAOD is derived from MODIS-retrieved TAOD and Fine Mode Fraction (FMF) with a priori characteristic FMF for individual aerosol types. Over land, DAOD is derived using spectral dependence of aerosol extinction (i.e., Angstrom exponent) and single scattering albedo. In other words, MODIS retrieves overland DAOD based on dust size supplemented by absorption characteristics.

Given these retrieval uncertainties and methodological differences, some discrepancies between the two DAOD climatological datasets are expected. In this section, we will compare the two datasets to identify and understand their similarities and differences. Since the mechanisms of dust generation, dust transport and dust removal processes all have a seasonal cycle (Mbourou et al. 1997; Parrington et al. 1983), we first present and discuss dust spatial distributions for each season in this section. Table 4 summarizes the seasonal and annual mean DAOD and TAOD values averaged over ocean, land and the globe (all limited to $60°$S-$60°$N), respectively, based on MODIS and CALIOP dust retrievals from 2007 to 2019. On multi-year average basis, the global DAOD was found to be 0.055 over the ocean and 0.103 over land based on MODIS, and 0.020 over ocean and 0.068 over land based on CALIOP. The global, annual mean DAOD (TAOD) is 0.032 (0.121) and 0.067 (0.171) according to CALIOP and MODIS retrieval, respectively.

As a comparison of two DAOD retrievals in this study, generally, DAOD from two retrievals differ by a factor of about 3 over ocean and less than 2 over land, while TAOD differ by a factor

of less than 2 over both ocean and land. The ratio of DAOD over land to that over ocean is about

2 and 3 for MODIS and CALIOP, respectively. For TAOD, the land to ocean ratio is about 2 for

both products. Overall, the difference in TAOD between two retrievals is less than their difference

in DAOD. On a global average, both MODIS and CALIOP-based DAOD peaks in boreal summer

(June-July-August). DAOD reaches minimum in boreal Fall (September-October-November) for

MODIS but in boreal Winter (December-January-February) for CALIOP. The MODIS and

CALIOP differences are region dependent, which is discussed as follows.

Figure 5 shows the spatial distribution of seasonal mean DAOD and the percentage of

DAOD to the TAOD based on 13-year (2007-2019) CALIOP and MODIS observations. Note that

this period is chosen because both datasets are available. Generally, MODIS-based DAOD is larger

than CALIOP-based DAOD. As expected, high values are seen from both CALIOP-based and

MODIS-based DAOD over the 'dust belt' regions extending from the west coast of North Africa

to the Middle East, Central Asia, and China, where large-scale dust activities occur persistently

throughout the year. However, the CALIOP-based DAOD is rather low in some other regions that

are known to be dusty in certain seasons, such as the southwestern United States, South America

(Patagonian Desert), Australia, and South Africa (i.e., Kalahari Desert). These regions do stand

out in MODIS DAOD maps (i.e., the second column in Figure 5). Then we plot DAOD-to-TAOD

ratio based on DAOD and TAOD retrievals from two sensors (the last two columns in Figure 5).

These regions indeed show up in the DAOD-to-TAOD ratio plot based on both sensors (i.e., the

last two columns in Figure 5). This means that in those regions both sensor-specific methodologies

are able to distinguish dust aerosol from sensor-detected total aerosol to some extent so that the

DAOD-to-TAOD ratio stands out in those regions for both sensors.

A close examination of Figure 5 revealed a land-to-ocean discontinuity in MODIS-based DAOD along the West African coastlines, especially between 30°S and 0° in the Summer and Fall seasons. This discontinuity could have been caused by several factors. First, we used the MODIS DB and DT products to derive DAOD over land and ocean, respectively. It is known the DB and DT algorithms are based on different methods and implemented by different groups, which inevitably leads to significant differences between the two and contributes to the discontinuity. This discontinuity has been pointed out in previous studies (e.g., Yu et al., 2021). Second, MODIS AOD retrievals are susceptible to cloud contaminations. The southeast Atlantic region has one of largest stratocumulus decks whose cloud amount peaks at Summer and Fall seasons (Klein and Hartmann 1993). Therefore, cloud contamination in MODIS DT retrievals over ocean leads to overestimation of AOD and underestimation of FMF and hence overestimation of DAOD (Yu et al., 2020). In addition, as explained in Section 3 we used different methods to derive the DAOD from the TAOD over land and ocean, which could also contribute to the problem. This land-to-ocean discontinuity is an important limitation of the current method. To mitigate this problem, substantial efforts are needed to improve the MODIS DT AOD and FMF retrievals and better understand the difference between DT and DB algorithm, which is beyond the scope of this study.

The climatological dust product shown in Figure 5 is a measure of the average dust loading over a geographical domain and time interval. It contains information of both the intensity and frequency of dust activities. The seasonal conditionally sampled DAOD shown in the first column of Figure 6 eliminates the impacts from dust frequency by excluding dust-free cases in the average. It is mainly related to the intensity of observed dust events. Therefore, the comparison between climatological and conditionally sampled DAOD sheds a light on the frequency and intensity of dust events detected by CALIOP. Therefore, we further compare the seasonal climatological

DAOD and conditional DAOD product. The second column of Figure 6 shows the seasonal climatological DAOD. The third column in Figure 6 shows the relative difference between conditionally sampled DAOD and climatological DAOD with respect to the climatological DAOD. In 'dust belt' regions, especially in Sahara Desert and Middle East where dust activities are persistent, climatological DAOD is very close to conditional DAOD. In Australia, Southwest United State, South America and South Africa, however, the conditional DAOD (column 1 in Figure 6) and the difference (column 3 in Figure 6) are relatively high. This suggests that dust activities in those regions are highly episodic and/or occur in relatively small scales. The difference also is very large in open oceans, suggesting that dust aerosols are present at a very low frequency.

Having analyzed the conditionally sampled DAOD from CALIOP, we now return to climatological DAOD and comparison between CALIOP and MODIS. Hereafter, all AOD values are climatological without otherwise explicit statement. Figure 7 shows the difference in seasonal mean TAOD, DAOD, and the percentage of DAOD in TAOD between MODIS retrievals and CALIOP retrievals. We first focus on 'dust belt' and its ocean out-flow regions extending from Northeast Atlantic, North Africa to the Middle East, Central Asia, China and Northwest Pacific. We note that in Figure 7 CALIOP-based TAOD and DAOD is generally smaller than MODIS-based ones over North Africa and Saharan dust out-flow region over the tropical Atlantic Ocean. One of the reasons of this large discrepancy is the choose of LR in CALIOP aerosol retrieval in these regions. CALIOP V4 products retrieve dust extinction coefficients with two steps. First, apply a globally uniform LR of $44sr$ for the identified dust aerosol layers to retrieve backscatter coefficients. Second, use the same LR of $44sr$ value to convert backscatter coefficients to extinction coefficients. Amiridis et al. 2013 shows that in the second step applying LR of $58sr$ to

CALIOP dust backscatter coefficients in North Africa improves the resulting aerosol extinction in terms of optical depth comparison with synchronous and collocated AERONET and MODIS measurements. Similarly, over Sahara Desert and the tropical Atlantic Ocean (see Figure 8 (a) and (d)), we apply LR of $58sr$ to the derived backscatter coefficient of dust component to get extinction coefficient of dust component. The resulting DAOD for LR of $58sr$ shows an improvement in comparison with MODIS DAOD relative to LR of $44sr$ (Figure 9 (a) and (d)). Therefore, the choose of LR can largely explain the difference between MODIS and CALIOP DAOD over North Africa and tropical Atlantic Ocean. For other regions, typical values of LR of desert dust aerosols vary between 35 and 55 $sr$, which is basically covered by the range of $44 \pm 9sr$ used in this study. The DAOD uncertainty induced by $\pm 9sr$ is estimated to be around 20% as shown in the shaded area in Figure 9.

In Middle East (the region indicated by Figure 8 (b)), the second column in Figure 7 shows that MODIS-DAOD is generally larger than CALIOP-DAOD in Arabian Peninsula, while opposite in India.

In Arabian Sea (the region indicated by Figure 8 (h)), comparing column 2 and column 4 in Figure 7, we could see that MODIS-DAOD is significantly larger than CALIOP-DAOD during JJA, during which cloud fraction is very high in the region. MODIS aerosol retrieval is more susceptible to cloud contamination. Specifically, the cloud contamination can lead to an overestimation of TAOD but underestimation of FMF. Although the MODIS retrieval algorithm neither assume coarse particles are exclusively from dust aerosols nor assume dust particles are all coarse particles (Yu et al., 2020), coarse mode aerosols are primarily dust. Thus, the overestimation of TAOD and underestimation of FMF will lead to an overestimation in DAOD.

Over Eastern Asia and Asian dust outflow region (Northwest Pacific-NWP), CALIOP-based DAOD is generally smaller than MODIS-based DAOD. There could be several reasons for this. First, this region is a major outflow region of Asian pollution (Yu et al., 2020). It is possible that the internal mixing of dust aerosols with industrial pollution in this region changes the dust morphology making it less non-spherical (Li and Shao 2009, Huang et al. 2020) but larger in size, which leads to smaller depolarization ratio and smaller fine-mode fraction. As a result, CALIOP shape-based DAOD derivation method could not capture the dust particles contained in the mixture, while those dust particles can be captured by MODIS size-based method. Another potential reason could be associated with that dust plumes in this region are vertically dispersed (Yu et al., 2010; Su and Toon, 2011). These tenuous dust layers are likely to go undetected by CALIOP because of its relatively low sensitivity. However, MODIS retrieves aerosol from the columnal integrated reflectance which is not dependent on the vertical distribution of aerosol. The difference may also be caused by uncertainties in MODIS aerosol retrievals. The West Pacific Ocean is cloudy almost all year long (see the last column in Figure 7), which makes MODIS aerosol retrievals bias high due to its more susceptibility to cloud contamination. An exception occurs during winter when cloud fraction is large in NWP. The MODIS-based DAOD is smaller than CALIOP-based DAOD, even though MODIS TAOD is larger than CALIOP TAOD. Similarly, over the southeastern Atlantic Ocean, CALIOP-based DAOD is also generally smaller than MODIS-based DAOD. On one hand, cloud contamination may have biased the MODIS dust retrieval high. On the other hand, CALIOP clear-sky sampling is not large enough to capture some dust events in this region.

We further compare DAOD (Figure 9) and TAOD (Figure S7 in the supplementary) retrievals from CALIOP and MODIS over major dust laden regions (as shown in Figure 8),

including three source regions on land (i.e., Sahara Desert, Middle East and Eastern Asia) and six oceanic outflow regions (i.e., the Tropical Atlantic Ocean - TAT, the Caribbean Basin - CRB, the Mediterranean Sea - MED, the Northwest Pacific Ocean - NWP, the Arabian Sea - ARB as well as the tropical Indian Ocean and the Bay of Bengal - IND). Each data point in the scatter plot represents a monthly mean DAOD (or TAOD) in a 2º × 5º grid. The density of data is represented by different color. To avoid our analysis being biased by some extreme and rare cases, we exclude those data points within the lowest 5% of data density (grey points in Figure 9 and Figure S7). Overall, the DAOD from the two instruments correlate well ($R > 0.75$) and on average CALIOP-based DAOD is 18%, 34%, 54% and 31% lower than MODIS-based DAOD over the Sahara (Figure 9(a)), TAT (Figure 9(d)), CRB(Figure 9(e)) and ARB(Figure 9(h)) regions, respectively. Applying LR of $58sr$ to Sahara dust reduces the difference from 18% to 8% over the Sahara and from 34% to12% over TAT. Over the Sahara Desert, the good agreement in DAOD between the two sensors (bias of 8% and R = 0.78) suggests that over the Sahara Desert dust particles can be adequately characterized by both irregular non-spherical shape and coarse size. As a result, both CALIOP- and MODIS-based methods are able to detect and separate the dust from other types of aerosols. In TAT and ARB regions, two instruments correlate well ($R > 0.8$) in both DAOD and TAOD. For TAOD, CALIOP is smaller than MODIS by 2% in TAT and larger than MODIS by 15% in ARB. Differences in DAOD are larger, with CALIOP DAOD lower than the MODIS DAOD by 12% and 31% in TAT and ARB, respectively. This suggests that the differences in DAOD from the two instruments are mainly resulted from differences in the dust separation method. In East Asia and NWP, on contrast, both TAOD and DAOD show poor correlation between the two methods (Figure 9(c), 9(g), S7(c) and S7(g)). As discussed earlier, the poor correlation between the two methods may be contributed by many factors. For example, the total

TAOD retrievals from MODIS are subject to larger uncertainties due to cloud contamination, or the DAOD retrieval from CALIOP may miss spherical dust particles that are coated by large combustion emissions from East Asia.

## 4.2    Comparison between CALIOP and MODIS DAOD Seasonality

Figure 10 compares annual cycle of MODIS and CALIOP DAOD based on the 13-year (2007-2019) average over the nine dust laden regions.   Each data point represents domain-averaged 13-year mean DAOD for a month while the error bar indicates $\pm1\sigma$ (one standard deviation of DAOD). The seasonal cycles of dust activities and dust transport are consistent with results in literature. For example, Prospero et al. 2002 shows that dust activity peaks in May-July in North Africa and Middle East, while peaks in spring in China. These seasonal cycles are consistent with our results shown in the first row of Figure 10. Yu et al. 2015a shows that DAOD peaks in June-July-August in La Parguera, which is consistent with the seasonal cycle in CRB in this study. Generally, CALIOP and MODIS show very similar seasonality over those dust laden regions. DAOD peaks in summer June-July-August (JJA) over Sahara Desert, Middle East, TAT, CRB, ARB and IND, but in spring March-April-May (MAM) over Eastern Asia, MED and NWP. Over NWP, the seasonal cycle of MODIS DAOD is somewhat different from that of CALIOP DAOD. While CALIOP DAOD peaks in spring, MODIS DAOD shows a peak in late spring or even summer months for some years. This could have resulted from cloud contamination in MODIS retrievals due to the large cloud fraction in summer [Yu et al., 2020].

Compared to the MODIS dust retrieval, CALIOP has a unique capability of detecting dust aerosol vertical distribution. Figure 11 shows seasonal mean dust extinction vertical profile from CALIOP for the nine dust-laden regions. The values on each plot represent the seasonal mean DAOD. Both DAOD and dust vertical structure have a seasonal dependence. In Sahara (a), Middle

East (b) and their dust outflow regions the Tropical Atlantic (d) and the Arabian Sea (h), summertime dust aerosol has the highest DAOD and reaches to the highest altitude extending from surface up to 6km in altitude.

The analysis above has been performed over the broad dust-laden regions. Here we focus on MODIS and CALIOP comparison in major potential source areas (PSAs) for dust in North Africa, namely NAF-1 to NAF-6 as illustrated in Figure 12 (adapted from Fig. 1 in Formenti et al., 2011). Among all dust source regions around the globe, the Sahara Desert and its margins in North Africa are the largest dust emitter. Within this region, prominent dust sources are often associated with topographical lows and foothills of mountains (Prospero et al. 2002). Seasonal variations of DAOD in the six PSAs are shown in Figure 13. Two B values are shown in the upper left of each panel in Figure 13, where B is defined as the average of CALIOP DAOD / MODIS DAOD ratios of all data pairs. B=1, >1, <1 indicates no bias, high bias and low bias. They are calculated based on CALIOP DAOD using dust LR of 44sr and 58sr respectively. The CALIOP DAOD derived using larger LR (58sr) achieve a better agreement (B values are closer to 1) with MODIS DAOD. Striking CALIOP and MODIS differences in DAOD exist in NAF-5 where the mean bias (B) deviate far from 1. NAF-5 (14N-20N, 15E-20E) is located in Bodélé Depression, Western Chad. This region is reported as the most intense dust source in the world (Prospero et al. 2002), and dust activity in the region occurs with a high frequency during all seasons except fall (Mbourou et al., 1997). However, CALIOP DAOD are much smaller than MODIS retrievals in this region. In terms of dust seasonality (Figure 13), the MODIS DAOD indicates intense dust aerosol loading all year long with a lower DAOD in Fall, while CALIOP shows a more distinct seasonality with the highest DAOD of about 0.3 in May-July and the lowest DAOD of <0.1 in

winter. Over other PSAs in North Africa, MODIS and CALIOP DAOD show similar seasonality with B closing to 1 (Figure 13).

In summary, MODIS and CALIOP DAOD show largest differences under the following conditions: (1) highly cloudy oceanic regions and (2) dust-pollution internal mixtures with high relative humidity. Their differences can be explained as follows.

1. Over cloudy ocean, effective cloud screening is critical to the quality of aerosol retrievals. As an active sensor, CALIOP is more reliable in discriminating clouds and aerosols than passive imager MODIS. In addition, active sensor is able to avoid impact from cloud side scattering. Therefore, MODIS is subject to more cloud contamination than CALIOP. Large cloud contamination in MODIS results in overestimation in TAOD and underestimation in FMF, introducing a high bias in DAOD over ocean cloudy regions (e.g., NWP).

2. Pure dust particles are hydrophobic and will not absorb water vapor. However, for dust aerosols coated by other types of aerosols (such as the deliquescent dust-nitrate $Ca(NO_3)_2$) and saline mineral dust particles emitted from saline topsoil in arid and semiarid areas (Tang et al. 2019), those types of dust particles will take up water vapor and grow to be larger in size and more spherical in shape (Wu et al. 2020c). This phenomenon is most prominent for dust aerosols in polluted region (e.g., EAS) as well as with relatively high relative humidity. While such coarse spherical dust particles will not be accounted as dust in CALIOP shape-based method, they are categorized as dust in the MODIS size-based method.

### 4.3  DAOD Inter-annual variation from CALIOP and MODIS observations

In this section we examine the inter-annual variation of DAOD captured by two sensors over several major dust source and outflow regions. Figure 14 shows a global map of DAOD trend derived based on the 13-year (2007-2019) time series of annual mean DAOD from CALIOP and MODIS. DAOD trend are calculated for each 2°×5° grid. Red color indicates positive trend and blue negative trend. Regions where the trend is statistically significant ($p < 0.05$) are marked with symbol '+'. The similar trend map for total aerosol optical depth is shown in Figure S8 in the supplementary. Overall, DAOD global pattern of interannual trend is similar to TAOD in major dust-laden regions. For example, Over Sahara Desert and tropical Atlantic Ocean region, both CALIOP and MODIS do not show statistically significant trend in DAOD and TAOD. In East Asia and the northwest Pacific Ocean, both sensors show negative trend in DAOD and TAOD.

Figure 15 displays interannual variability of annual-mean DAOD for the major dust-laden regions as defined in Figure 8. Seasonal and annual DAOD trends in the nine regions are listed in Table 4. Both MODIS and CALIOP show a clear DAOD trend in certain seasons over the Eastern Asia, ARB and NWP regions. In Eastern Asia, MODIS and CALIOP show a consistent DAOD decreasing trend at a rate of $-1.7\%\ yr^{-1}$ annually. The two sensors show a DAOD decreasing tend of $-3.5\%\ yr^{-1}$ and $-2.5\%\ yr^{-1}$ respectively in Eastern Asia during spring and show a consistent trend of DAOD in ARB during the fall, though with a factor of 2 difference in magnitude. In NWP, both MODIS- and CALIOP-based DAOD shows a decreasing trend of $-1.7\%\ yr^{-1}$ and $-1.6\%\ yr^{-1}$, respectively. The annual DAOD decreasing trend in NWP is mainly attributed to the DAOD decline in spring at a rate of $-2.3\%\ yr^{-1}$ and $-3.0\%\ yr^{-1}$ for MODIS and CALIOP, respectively. For comparison, Shimizu et al. (2017) detect the decreasing DAOD trends of $-4.3\%\ yr^{-1}$ in spring and $-2.5\%\ yr^{-1}$ on annual mean basis from the Asian Dust Network

(AD-Net) lidar observations over Japan (2007-2016). These trends are greater than our results based on MODIS and CALIOP data records.

Dust over NWP comes mainly from East Asian dust sources. The broad East Asian region (ESA defined in Figure 12) shows statistically significant DAOD decreasing trends (Figure 15c) which is consistent with the DAOD decreasing trend in NWP. It is also imperative to further examine which of six major PSAs in East Asia (ESA-1 to ESA-6 in Figure 12) contribute to the decreasing trend of DAOD. As shown in Figure 16, among the six PSAs, the satellite data show consistent interannual declining trend of DAOD in EAS-5 (Southern Gobi Desert) at a rate of $-4.8\%\ yr^{-1}$ and $-2.8\%\ yr^{-1}$ for MODIS and CALIOP, respectively. In spring, DAOD in EAS-5 shows a significantly declining trend at a rate of $-5.6\%\ yr^{-1}$ and $-3.3\%\ yr^{-1}$ for MODIS and CALIOP (Figure S9). Figure 17 assesses the correlation between DAOD in EAS-5 and DAOD in NWP based on MODIS and CALIOP, respectively. For annual mean DAOD from 2007 to 2019, both sensors show a good correlation between EAS-5 and NWP with $R \approx 0.6$ ($p = 0.02$). In spring, the correlation of DAOD from two regions is good based on CALIOP ($R = 0.6$ , $p = 0.03$), while a weaker correlation ($R = 0.53, p = 0.07$) was found based on MODIS. We further examine potential factors contribute to the declining trend of DAOD in ESA-5 (Qian et al. 2002; Kurosaki and Mikami 2003; Lee and Sohn 2011). The first row in Figure 18 shows the springtime trend of MODIS enhance vegetation index (EVI), MERRA2 near-surface (at 10 m) wind speed (Carvalho et al. 2019) and precipitation (Reichle et al. 2017) in EAS-5 region. EVI and precipitation show a statistically significant (p<0.05) increasing trend with R = 0.82 and R=0.58, respectively. Surface wind speed shows a statistically significant (p<0.05) decreasing trend with R = −0.66. The second row in Figure 18 shows the correlations of the three factors with MODIS DAOD and CALIOP DAOD, respectively. Clearly, EVI is anti-correlated with both MODIS and CALIOP DAOD with

|R| > 0.7 and p<0.05. Surface wind speed is correlated with MODIS DAOD and CALIOP DAOD with |R| > 0.6 and p<0.05. While the correlation with precipitation is not statistically significant (p>0.05). Note that EVI and surface wind speed are not independent variables that affect dust emissions. An increase of EVI or vegetation cover could reduce the surface wind speed. However, given the relatively coarse resolution of MERRA2, the surface wind speed trend may largely reflect the change in atmospheric circulations other than local wind decrease induced by more vegetation. The precipitation shows no statistically significant correlation with MODIS and CALIOP DAOD.

As discussed earlier, our results suggest that the decrease of NWP DAOD is likely a result of the decreasing dust events in Asian deserts (i.e., EAS-5 Gobi) in turn likely due to change of vegetation. This is also reported in several recent studies. Sternberg et al. (2015) found that Gobi Desert contracted from 2000 to 2012 due to increased moisture availability. Song et al. (2016) used an Integrated Wind Erosion Modeling System to simulate the spring dust emissions in northern China over the period of 1982 to 2011. They found a significant decrease of the magnitude of spring dust event in China which is attributed to both climate change and local mitigation strategies. Similarly, An et al., (2018) also noted a significant decrease of dust storm event in East Asian after analyzing observational data from ground stations, numerical modeling, and vegetation indices obtained from both satellite and reanalysis data. Over the last few decades, The Chinese government has been taking actions to restore overgrazed land in Inner Mongolia, the enlarged vegetation coverage and the expected earlier vegetation green-up due to global warming could have mitigated dust activity in this region (Fan et al. 2014). Together the results from our analysis, along with the aforementioned recent studies, suggest that the decreasing springtime DAOD trend

in the NWP region is a result of the decline of dust activities in the Inner Mongolia (i.e., EAS-5) which is likely linked to vegetation coverage changes in recent years as a result of China's mitigation projects to hold back desertification.

Some caveats must be mentioned, however, when interpreting the trend analysis here. First of all, due to the limitation of satellite data record, we have only 13 years' CALIOP data and 17 years' MODIS data available. Other climate variabilities, such as the El Nino-Southern Oscillation (ENSO), could confound the trend analysis. For example, Abish and Mohanakumar (2013) shows that La Nina (El Nino) weakens (strengthens) the zonal circulation over the Indian subcontinent, which result in low (high) aerosol concentration over Indian subcontinent transported from Arabian Desert over the period. Gong et al. (2005) also shows the impact of ENSO on the interannual variability of Asian dust loading and deposition. According to the NOAA Oceanic Nino Index (ONI), the climate switched from a strong La Niña phase in 2010-2011 to a strong El Niño phase in 2015-2016. However, the potential impact of ENSO on the dust inter-annual variability is beyond the scope of this study and will be left for the future research.

# 5   Summary and Conclusion

We derive two observation-based global monthly mean dust aerosol optical depth (DAOD) climatological datasets from 2007 to 2019 with a 2° (latitude) ×5° (longitude) spatial resolution, one based on CALIOP and the other on MODIS observations. Our product captures very well as much hot spots along the 'dust belt' region well, as weaker signals in other dust active regions such as Southwestern United States, Patagonian Desert in South America, Central Australia, and South Africa (Figure 5). Since DAOD climatology product contains and mixes the information of the intensity and frequency of dust activities, we introduce the conditional DAOD product, which

diminishes impacts from dust frequency by excluding dust-free cases in the average. The comparison between DAOD climatology data and conditional DAOD data suggests that dust activities in those regions are highly episodic. The two data records compare reasonably well with the results reported in previous studies and the collocated AERONET coarse model AOD. The comparison of our MODIS-based and CALIOP-based DAOD with AERONT COD indicates that MODIS overestimates DAOD, while CALIOP underestimates DAOD. It is highly probably that the true DAOD fall between MODIS and CALIOP DAOD.

CALIOP distinguishes dust aerosols based on its non-spherical shape, whereas MODIS separates dust aerosols from others based on its large size characteristics. The discrepancy in dust retrieval based on two instruments are expected due to the uncertainty associated with their TAOD retrieval and the uncertainty associated with their different mechanism in dust detection and separation. The comparison between CALIOP dust retrieval and MODIS dust retrieval facilitate a better understanding of advantages and limitations of each dust product and also provide some insights on dust morphology and dust size. Through the comparison, we found generally CALIOP-based DAOD correlates well with MODIS-based DAOD over dust-laden regions such as Sahara (R=0.78), TAT (R=0.84), CRB (R=0.75) and ARB (R=0.85), but with CALIOP-based DAOD 18%, 34%, 54% and 31% lower than MODIS-based DAOD over those regions respectively. This result is consistent with the different treatment of the dust-pollution mixtures in the dust separation approaches of two instruments. Applying LR of $58sr$ to Sahara dust reduce the difference from 18% to 8% over the Sahara and from 34% to12% over TAT. Over the Sahara Desert, the good agreement in DAOD between the two sensors (bias of 8% and R = 0.78) suggests that dust aerosols are irregular non-spherical and at the same time large in size in this region. In some regions such

as NWP, the DAOD correlation between two sensors is quite low. There could be many reasons for this, for example, the total TAOD retrievals from MODIS have larger uncertainty due to cloud contamination, or the DAOD retrieval from CALIOP may miss coarse spherical dust-pollution mixtures.

The interannual variability of DAOD over dust-laden regions show a clear trend in Eastern Asia a rate of $-1.7\%$ $yr^{-1}$ based on two sensors. Over the outflow region of Easter Asia, DAOD in NWP region shows a clear trend at a rate of $-1.6\%$ $yr^{-1}$ and $-1.7\%$ $yr^{-1}$ based on CALIOP and MODIS respectively, this trend is mainly attributed to the decreasing trend in spring with a rate of $-3.0\%$ $yr^{-1}$ based on CALIOP and $-2.3\%$ $yr^{-1}$ based on MODIS. Further investigation of DAOD trend in six dust source areas in Eastern Asia where NWP dust aerosols come from shows that there is an obvious decreasing trend in DAOD during 2007 - 2019 over Southern Gobi Desert based on both CALIOP and MODIS dust retrievals. The decreasing trend of DAOD is correlated significantly with the vegetation index and surface wind speed in the area, whereas there is almost no correlation with the precipitation.

The two observation-based DAOD climatological datasets derived in this study can be highly valuable for many dust-related studies. For example, they can be used to assess dust direct radiative effects (e.g., Kok et al. 2017), study aerosol-cloud interactions (e.g., Cho et al. 2010, Tan et al. 2015) and identify global and regional dust trends and variabilities. They may also be used to evaluate the dust simulations in global climate models (e.g., M. Wu et al. 2020, C. Wu et al. 2020). On the other hand, the current study faces several important limitations. For example, as explained in Section 4, the current MODIS based DAOD climatology suffers from the land-to-ocean discontinuity problem due to the use of two AOD products and potential cloud

contaminations. The small horizontal sampling rate of CALIOP is also an important limitation. These problems are beyond the scope of this study and will be left for future research.

*Data availability.* The global DAOD and dust vertical extinction coefficient climatology data derived from CALIOP in this study and the MODIS DAOD retrieval data over land and ocean are available at

'https://drive.google.com/drive/folders/1aQVupe7govPwR6qmsqUbR4fJQsp1DBCX?usp=sharing'.

The MODIS Enhanced Vegetation Index (EVI) data could be downloaded from 'https://lpdaac.usgs.gov/products/myd13c2v006/#tools'. The MERRA2 surface wind speed and precipitation data are available at 'https://disc.sci.gsfc.nasa.gov/datasets/M2T1NXFLX_5.12.4/summary?keywords=%22MERRA-2%22'.

*Acknowledgement.* Qianqian Song and Zhibo Zhang cordially acknowledge the funding support from the Future Investigators in NASA Earth and Space Science and Technology (FINESST). Zhibo Zhang's research is supported by NASA grant (80NSSC20K0130) from the CALIPSO and CloudSat program managed by Dr. David Considine. HY was supported by NASA's the Science of Terra, Aqua, and Suomi-NPP and the CALIPSO/CloudSat Science Team programs administered by Dr. Hal Maring and Dr. David Considine, respectively. The computations in this study were performed at the UMBC High Performance Computing Facility (HPCF). The facility is supported by the US National Science Foundation through the MRI program (grant nos. CNS-0821258 and CNS-1228778) and the SCREMS program (grant no. DMS-0821311), with substantial support from UMBC. The MODIS aerosol data were obtained from the NASA Level-1 and Atmosphere Archive and Distribution System (LAADS) webpage

(https://ladsweb.nascom.nasa.gov/). The CALIOP aerosol products were obtained from NASA

Langley Research Center Atmospheric Science Data Center (https://eosweb.larc.nasa.gov/).

Table 1. Summary of DAOD retrievals from MODIS and CALIOP

| Sensors | Retrieve Scope | Relevant variables used to derive DAOD | References |
|---------|----------------|----------------------------------------|------------|
| MODIS | Ocean | AOD, fine-mode AOD | Yu et al. (2009, 2020) |
| MODIS | Land | AOD, SSA at 470nm, Angstrom exponent | Pu and Ginoux et al. (2018) |
| CALIOP | Globe | Profiles of backscatter, extinction, depolarization ratio | Yu et al. (2015a) |

Table 2. Compare global mean DAOD retrievals in this study with some relevant studies (Note the definition of global scope is different for different studies).

| Region | | DAOD@550nm | Reference |
|---|---|---|---|
| 90°S~90°N | Global | 0.03±0.005 | Ridley et al. 2016<br>Use multiple satellite platforms, in-situ AOD observations and four global models |
| 90°S~90°N | Global | 0.033 | Gkikas et al 2021<br>Use AOD from Aqua MODIS and DOD-to-AOD ratio from MERRA2 |
| 50°S~60°N | Over Ocean | 0.03±0.06 | Voss and Evan 2020<br>Over Ocean: use method in Kaufman et al 2005<br>Over Land: use method in Ginoux et al. 2012 |
| | Over Land | 0.1 | |
| 60°S~60°N | Over Ocean | 0.055, 0.020 | This Study<br>MODIS-based, CALIOP-based DAOD<br>(To calculate global mean DAOD for scope 90°S~90°N, we assume zero DAOD outside of region 60°S~60°N. We weight each grid-cell surface area into ocean, land and global DAOD average) |
| | Over Land | 0.103, 0.068 | |
| 90°S~90°N | Global | 0.057, 0.028 | |

Table 3 Statistical parameters and absolute error by continents using the method indicated in Figure 1. Sites is the number of AERONET sites involved; N is the number of MODIS, CALIOP and AERONET matchups. R is correlation coefficient; C is the intercept of the linear fit; K is the slope of the linear fit; RMSE is root mean square error of the linear fit; Ba is the absolute bias; Br is the relative bias. For cells with two rows of values, the upper row is for MODIS, the lower row is for CALIOP.

| Region | Sites | N | R | C | K | RMSE | Ba | Br (%) | Absolute Error |
|--------|-------|------|------|-------|------|------|-------|-------|----------------|
| Global | 761 | 16653 | 0.72 | 0.01 | 1.05 | 0.08 | 0.01 | 26.7 | $0.65 \times DAOD_M$ |
|        |       |      | 0.70 | −0.01 | 0.90 | 0.07 | −0.02 | −27.9 | $0.52 \times DAOD_C + 0.02$ |
| Africa | 44 | 706 | 0.79 | 0.04 | 0.72 | 0.10 | 0.01 | 4.5 | $0.37 \times DAOD_M + 0.01$ |
|        |    |     | 0.72 | 0.01 | 0.75 | 0.12 | −0.02 | −19.8 | $0.51 \times DAOD_C + 0.02$ |
| Asia | 143 | 2507 | 0.64 | 0.04 | 0.88 | 0.10 | 0.03 | 34.2 | $0.61 \times DAOD_M$ |
|      |     |      | 0.57 | 0.00 | 0.84 | 0.11 | −0.01 | −11.0 | $0.66 \times DAOD_C + 0.01$ |
| Europe | 156 | 4359 | 0.27 | 0.03 | 0.55 | 0.05 | 0.01 | 18.2 | $0.70 \times DAOD_M$ |
|        |     |      | 0.35 | 0.00 | 0.53 | 0.04 | −0.02 | −48.6 | $0.47 \times DAOD_C + 0.02$ |
| Americas | 319 | 6656 | 0.29 | 0.02 | 0.54 | 0.04 | 0.01 | 25.5 | $0.77 \times DAOD_M$ |
|          |     |      | 0.33 | 0.00 | 0.31 | 0.03 | −0.02 | −55.8 | $0.26 \times DAOD_C + 0.02$ |
| Australia | 12 | 507 | 0.51 | 0.0 | 0.57 | 0.03 | −0.02 | −43.9 | $0.37 \times DAOD_M + 0.02$ |
|           |    |     | 0.28 | 0.0 | 0.32 | 0.04 | −0.02 | −59.3 | $0.34 \times DAOD_C + 0.03$ |

Table 4. Global (60° S-60° N) seasonal mean DAOD and TAOD based on MODIS and CALIOP (2007~2019) dust retrievals. Since Earth is a sphere, grid-cell surface area decreases toward the poles. We weight each grid-cell surface area into ocean, land and global DAOD average.

| | | MAM | | JJA | | SON | | DJF | | Annual | |
|---|---|---|---|---|---|---|---|---|---|---|---|
| | | DAOD | TAOD | DAOD | TAOD | DAOD | TAOD | DAOD | TAOD | DAOD | TAOD |
| MODIS | Ocean | 0.057 | 0.151 | 0.062 | 0.153 | 0.047 | 0.143 | 0.052 | 0.144 | 0.055 | 0.148 |
| | Land | 0.131 | 0.283 | 0.119 | 0.270 | 0.079 | 0.206 | 0.085 | 0.217 | 0.103 | 0.244 |
| | Global | 0.075 | 0.183 | 0.077 | 0.183 | 0.055 | 0.159 | 0.059 | 0.160 | 0.067 | 0.171 |
| | Land /Ocean | 2.27 | 1.87 | 1.90 | 1.77 | 1.67 | 1.44 | 1.64 | 1.51 | 1.89 | 1.65 |
| CALIOP | Ocean | 0.022 | 0.098 | 0.025 | 0.104 | 0.015 | 0.092 | 0.016 | 0.090 | 0.020 | 0.096 |
| | Land | 0.086 | 0.196 | 0.086 | 0.228 | 0.051 | 0.186 | 0.047 | 0.157 | 0.068 | 0.192 |
| | Global | 0.039 | 0.124 | 0.041 | 0.137 | 0.025 | 0.117 | 0.024 | 0.107 | 0.032 | 0.121 |
| | Land /Ocean | 3.90 | 1.99 | 3.45 | 2.20 | 3.40 | 2.03 | 2.87 | 1.74 | 3.45 | 2.00 |

Table 5. DAOD trend over major dust-laden regions based on MODIS and CALIOP observations. The changing rate of DAOD trend is shown in a sequence of annual/spring/summer/fall/winter in each cell of the table. Those statistically meaningful trends with p<0.05 are shown in bold.

| | MODIS [% yr⁻¹] | | | | | CALIOP [% yr⁻¹] | | | | |
|---|---|---|---|---|---|---|---|---|---|---|
| | Annual | MAM | JJA | SON | DJF | Annual | MAM | JJA | SON | DJF |
| Sahara Desert (a) | -0.04 | -0.84 | 0.21 | 0.29 | 0.51 | -0.09 | -0.93 | 0.34 | -0.52 | 0.55 |
| Middle East (b) | 0.32 | -0.61 | -0.02 | **1.80** | 1.37 | **-1.84** | -2.36 | -1.86 | **-2.46** | -0.09 |
| Eastern Asia (c) | **-1.74** | **-3.48** | -0.28 | -0.33 | -0.56 | **-1.70** | **-2.46** | **-1.99** | -0.45 | -1.42 |
| TAT (d) | 0.34 | -0.68 | -0.03 | 1.68 | 1.32 | -0.25 | -1.41 | -0.07 | 0.91 | -0.09 |
| CRB (e) | 1.10 | 0.78 | 0.94 | **1.59** | **1.97** | -0.40 | -1.39 | -0.34 | 0.79 | -1.09 |
| MED (f) | 0.10 | 0.32 | 0.49 | -0.71 | 0.03 | -1.09 | -1.07 | -1.63 | -1.20 | -0.52 |
| NWP (g) | **-1.67** | **-2.33** | **-1.93** | 0.63 | -1.35 | **-1.58** | **-3.01** | **-2.89** | -0.40 | -0.19 |
| ARB (h) | -1.42 | -0.72 | -1.81 | **-1.85** | -0.31 | -1.17 | -1.70 | -0.46 | **-3.60** | -0.06 |
| IND (i) | -0.09 | -0.51 | 0.40 | 0.38 | -0.89 | -1.96 | -2.92 | -2.43 | -0.21 | -0.54 |

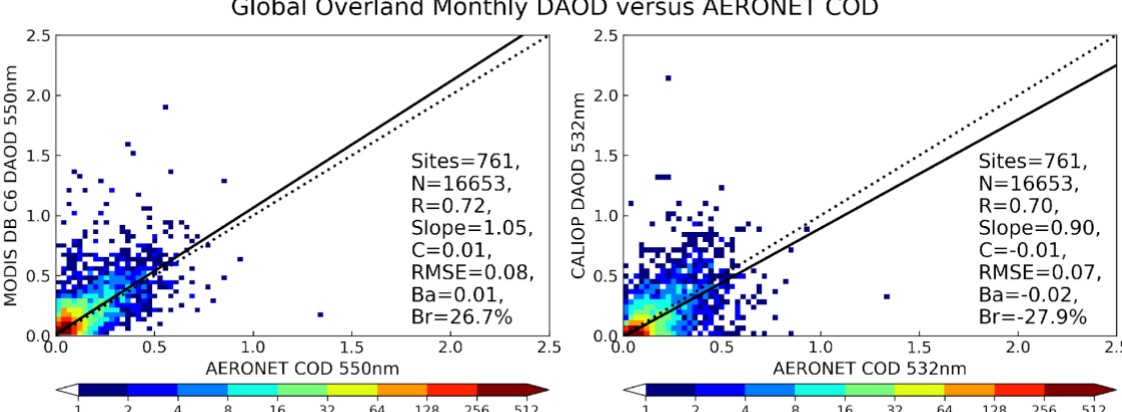

Figure 1. Scatter density histogram comparing monthly (from 2007 to 2019) MODIS DAOD (on the left panel) and CALIOP DAOD (on the right panel) with monthly coarse mode COD$_{SDA}$ retrieved at 550nm for MODIS comparison and at 532nm for CALIOP comparison from the Level 2 (cloud screened and quality assured) Spectral Deconvolution Algorithm (SDA) version 4.1 (O'Neill et al., 2003). The 1 to 1 line and linear regression line are shown by dotted and solid lines, respectively. The number of sites (Sites), of matchups (N), correlation (R), slope (S), constant (C), and root mean square error (RMSE) of the linear regression as well as absolute bias (Ba) and relative Bias (Br) are indicated in the lower right of the panel. Ba, Br and RMSE are defined as:

$$B_a = \underline{DAOD}_{C\ or\ M} - \underline{COD}_{SDA}, B_r = \underline{DAOD}_{C\ or\ M} / \underline{COD}_{SDA} - 1, RMSE =$$

$$\sqrt{\frac{\sum_i \ (\underline{DAOD}_{C\ or\ M,i} - \underline{COD}_{SDA,i})^2}{N}}$$

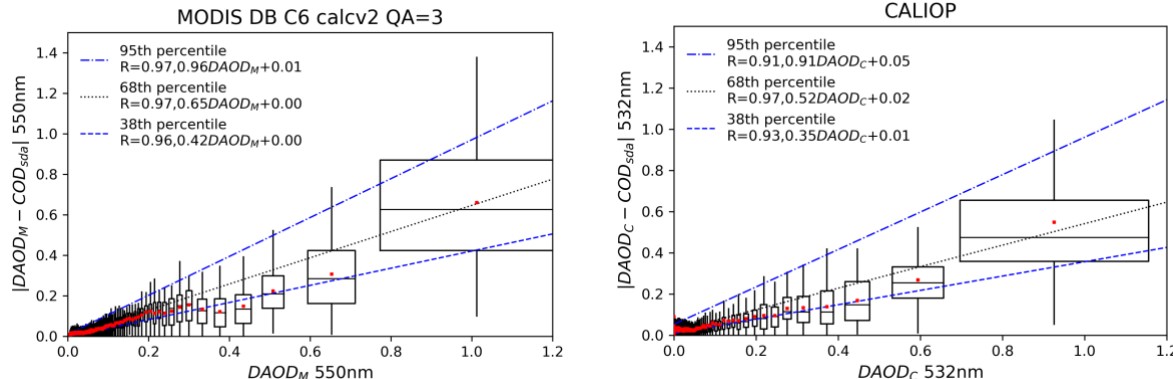

Figure 2 Left panel: the x-axis is the MODIS derived DAOD, y-axis is the absolute MODIS DAOD−AERONET COD difference (without scaling by AMF). Data are sorted by bins of 100 values (we have 16653 matchups in total; therefore, the last bin has 53 values). The means and standard deviations of the MODIS DAOD$_M$ are the centers and half widths of the boxes in the horizontal. The mean, medians, and lower to upper quartile interval of the MODIS –AERONET SDA differences are the red dots, the center, and top-bottom intervals of the boxes. The dotted line is the error estimated from the least squares linear fit of the 68$^{th}$ percentiles for each box. The right panel is the same except for CALIOP DAOD.

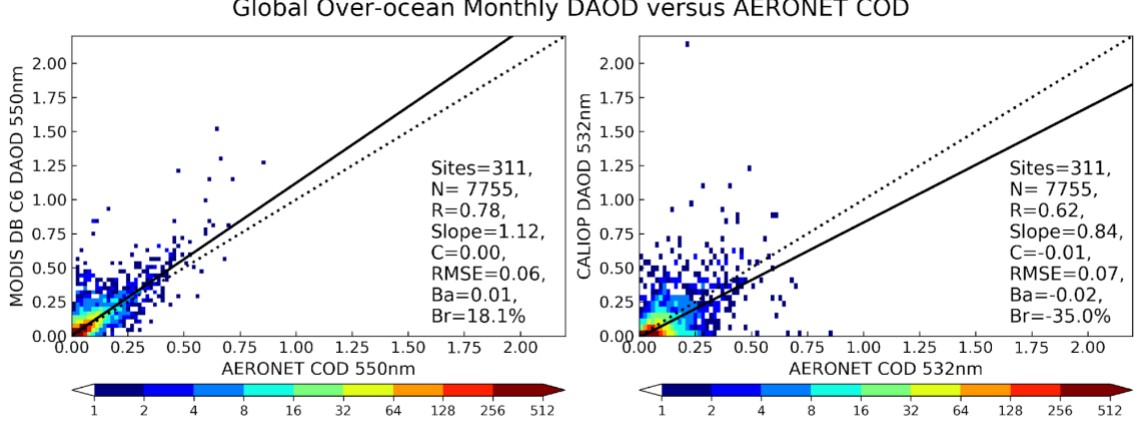

Figure 3 The same as Figure 1 except for over ocean.

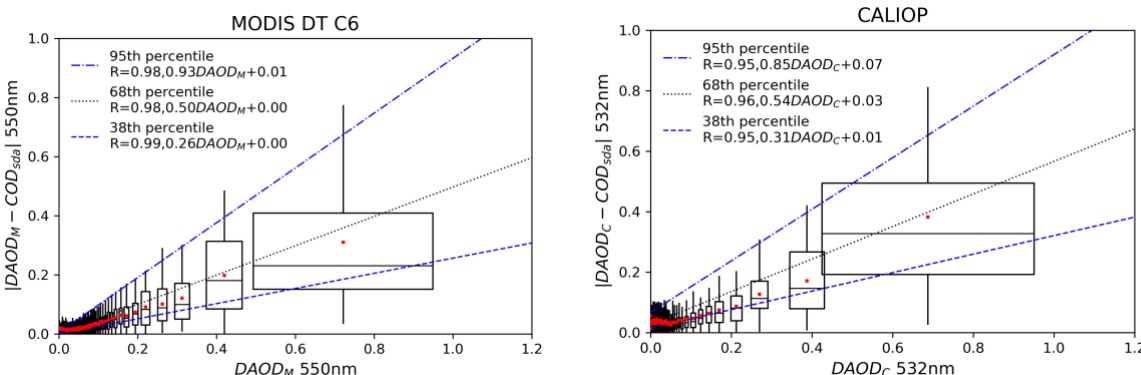

Figure 4 The same as Figure 2 except for over ocean.

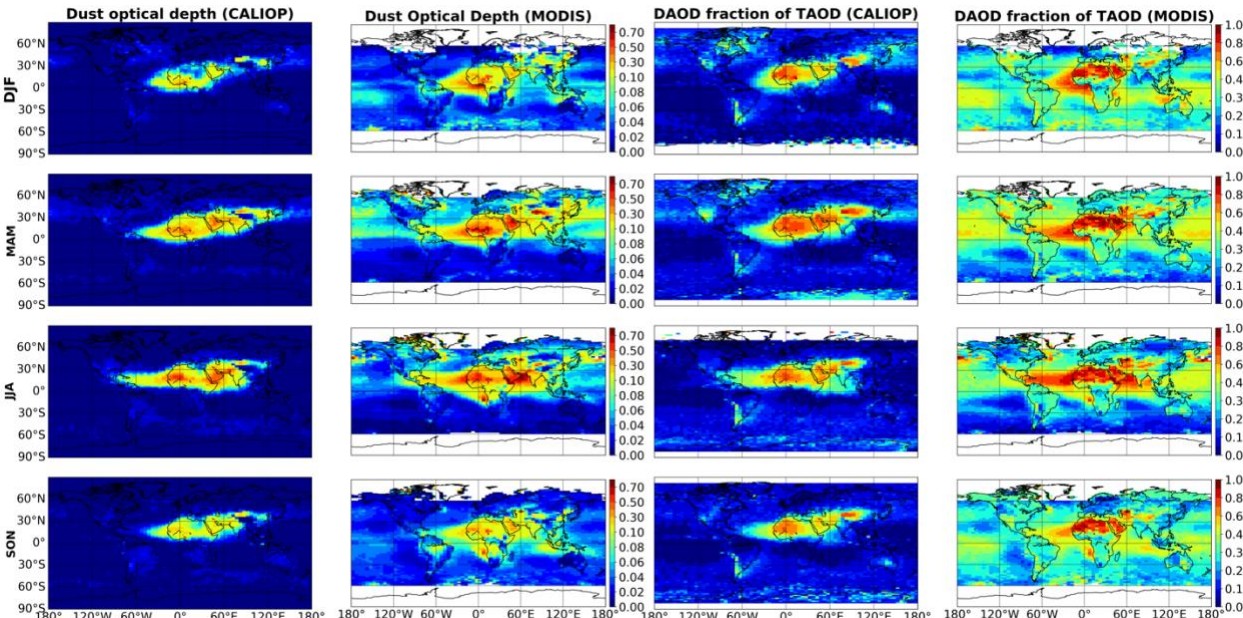

Figure 5. Spatial distribution of the seasonal mean CALIOP-based DAOD, MODIS-based DAOD and the fraction of DAOD with respect to the TAOD based on CALIOP and MODIS respectively for the globe at a 5° $longitude \times 2°\ latitude$ resolution based on 13-year (2007-2019) CALIOP measurements. **DJF**: December from previous year-January-February; **MAM**: March-April-May; **JJA**: June-July-August; **SON**: September-October-November.

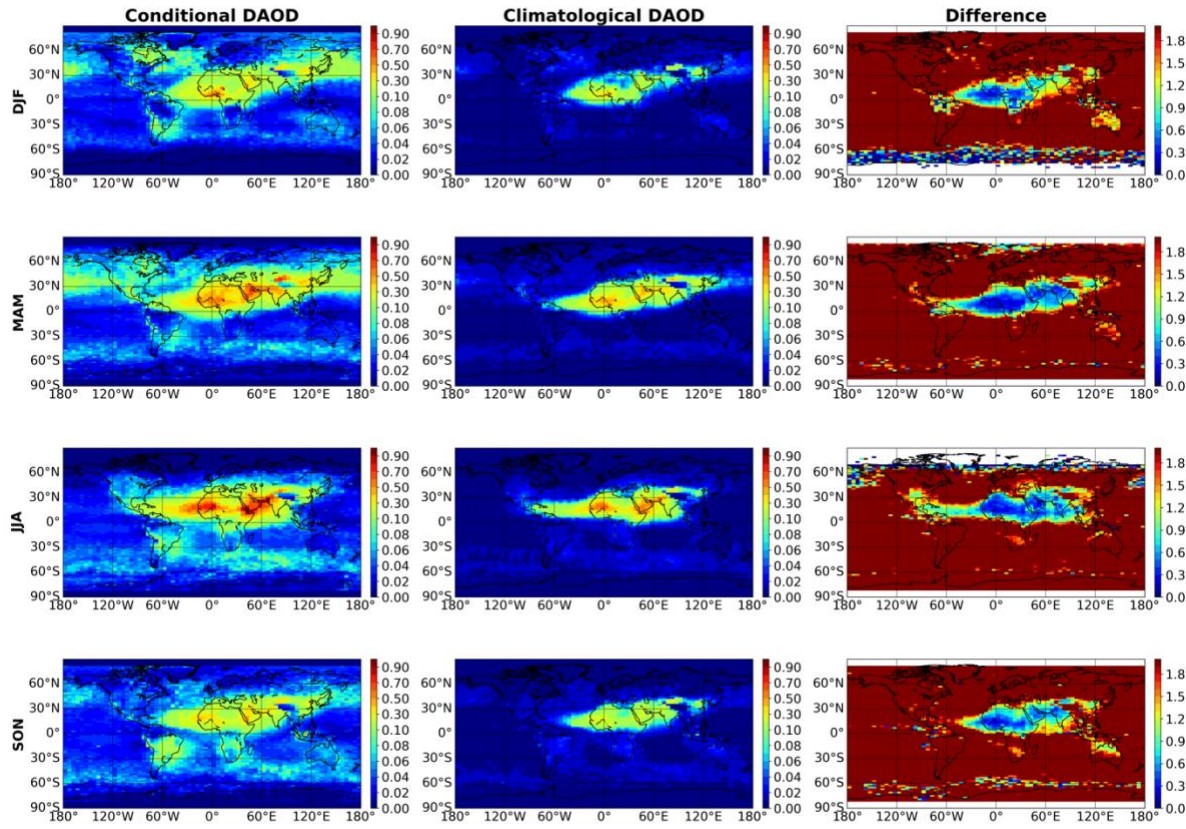

Figure 6. Conditional DAOD (the first column), climatological DAOD (the second column) based on CALIOP dust retrieval from 2007 to 2019. The third column shows the relative difference between conditionally sampled DAOD and climatological DAOD with respect to the climatological DAOD expressed in fraction.

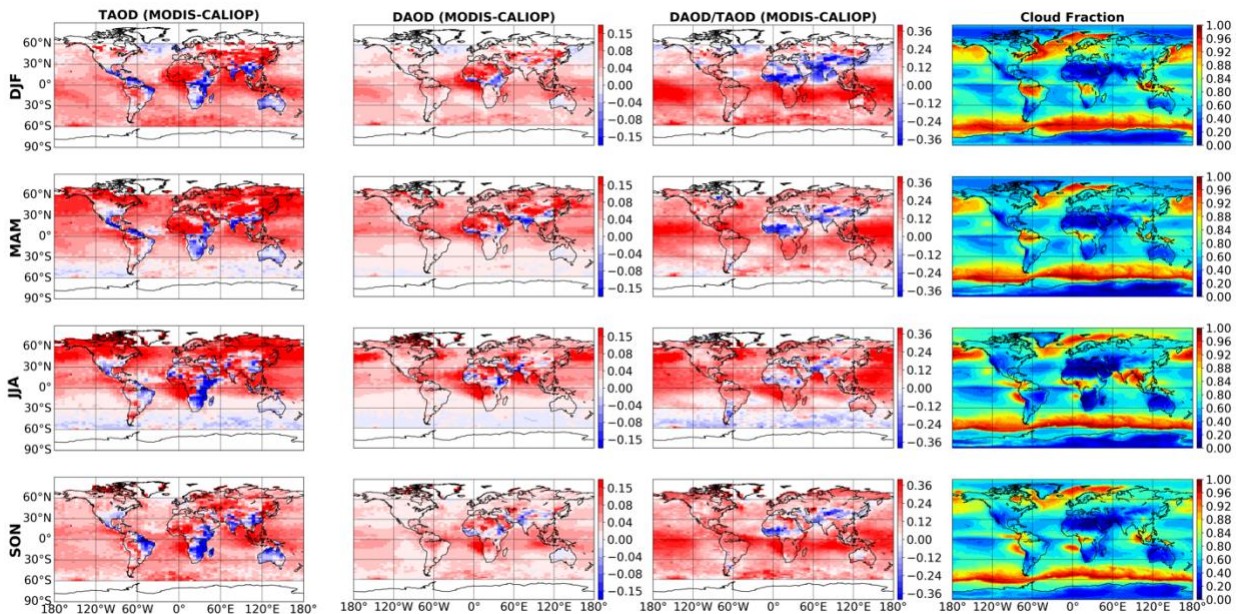

Figure 7. The difference between MODIS and CALIOP for seasonal mean TAOD (the first column), DAOD (the second column), and the fraction of DAOD in TAOD (the third column) on a basis of 13-year (2007-2019) average. The fourth column is the seasonal mean cloud fraction from MODIS L3 product.

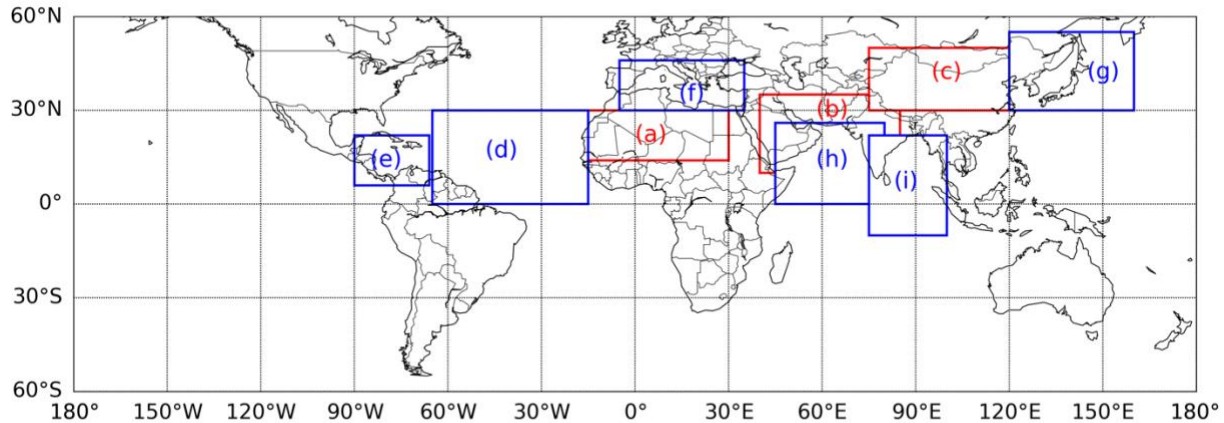

Figure 8. Major dust-laden regions including three dust source regions on land (a ~ c) and six outflow regions over ocean (e ~ i). (a) Sahara Desert (14ºN-30ºN, 15ºW-30ºE), (b) Middle East (10ºN-35ºN, 40ºE-85ºE) and (c) Eastern Asia (30ºN-50ºN,75ºE-130ºE)  (d) the tropical Atlantic Ocean–TAT (0º-30ºN, 15ºW-60ºW), (e) the Caribbean Sea–CRB (6ºN-22ºN, 60ºW-90ºW), (f) the Mediterranean Sea–MED (30ºN-46ºN, 5ºW-35ºE), (g) the northwest Pacific Ocean–NWP (30ºN-55ºN, 120ºE-160ºE), (h) the Arabian Sea–ARB (0º-26ºN,45ºE-80ºE and (i) the tropical Indian Ocean and the Bay of Bengal–IND (10ºS-22ºN,75ºE-100ºE). Note we only consider grids over land for the three dust source regions and grids over ocean for the six dust outflow regions.

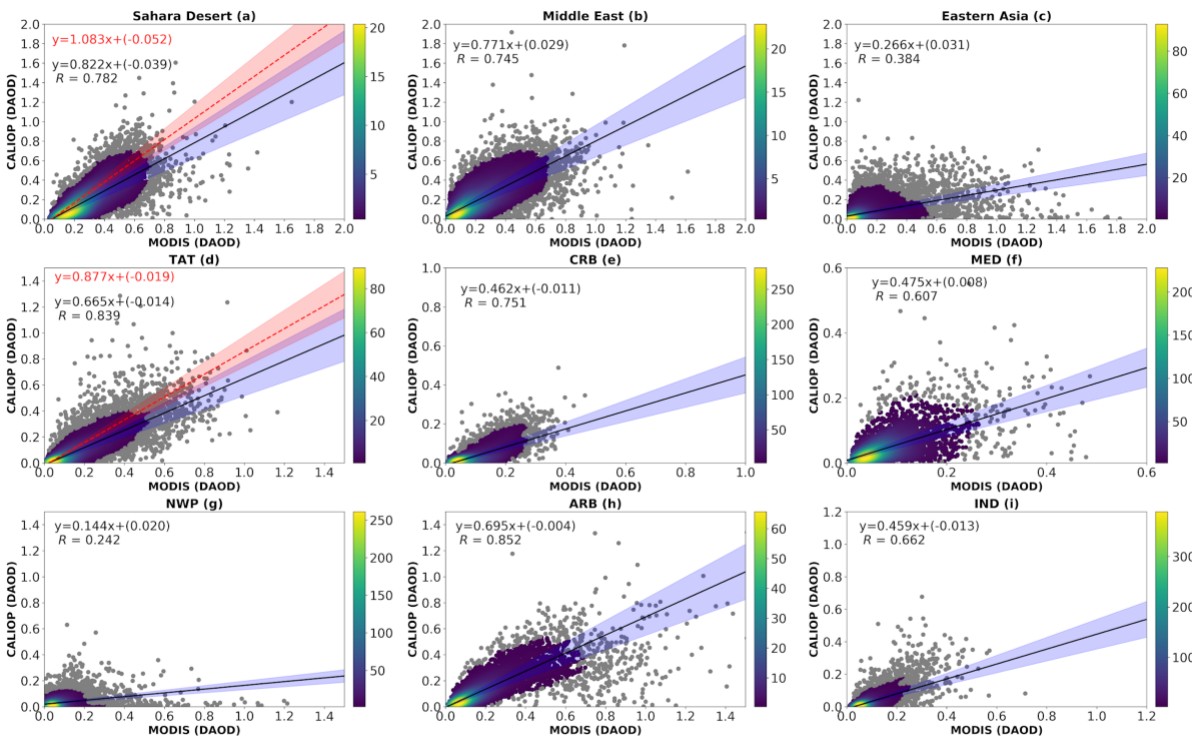

Figure 9. Comparison of CALIOP DAOD against MODIS DAOD over dust-laden regions indicated in Figure 8. Color represents the probability density using gaussian kernel density estimation. Grey points represent data points within the lowest 5% of data density. Those grey points are excluded in the linear regression analysis. The blackline and blue shadow are the linear regression for LR=44 $\pm$ 9$sr$, the red line and red shadow in (a) and (d) represent the linear regression for LR=58 $\pm$ 8$sr$. Red text in panel (a) and (d) is the linear fit based on LR=58 sr. Black text in each panel is the linear fit based on LR=44 sr. R is Pearson's linear correlation coefficient between MODIS and CALIOP DAOD.

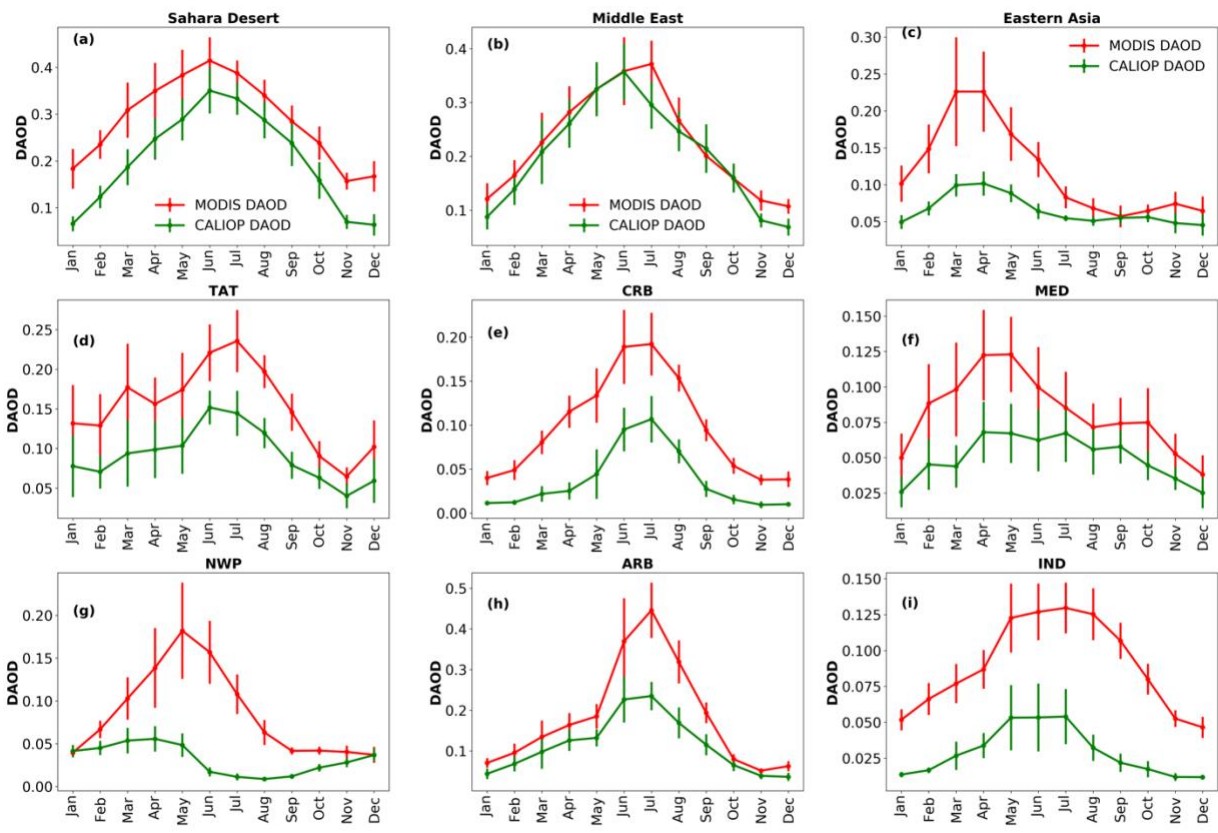

Figure 10. Monthly variation of DAOD from CALIOP (green) and MODIS (red) for major dust-laden regions indicated in Figure 4. Vertical line represents ±1 sigma (standard deviation) over the 13-year period.

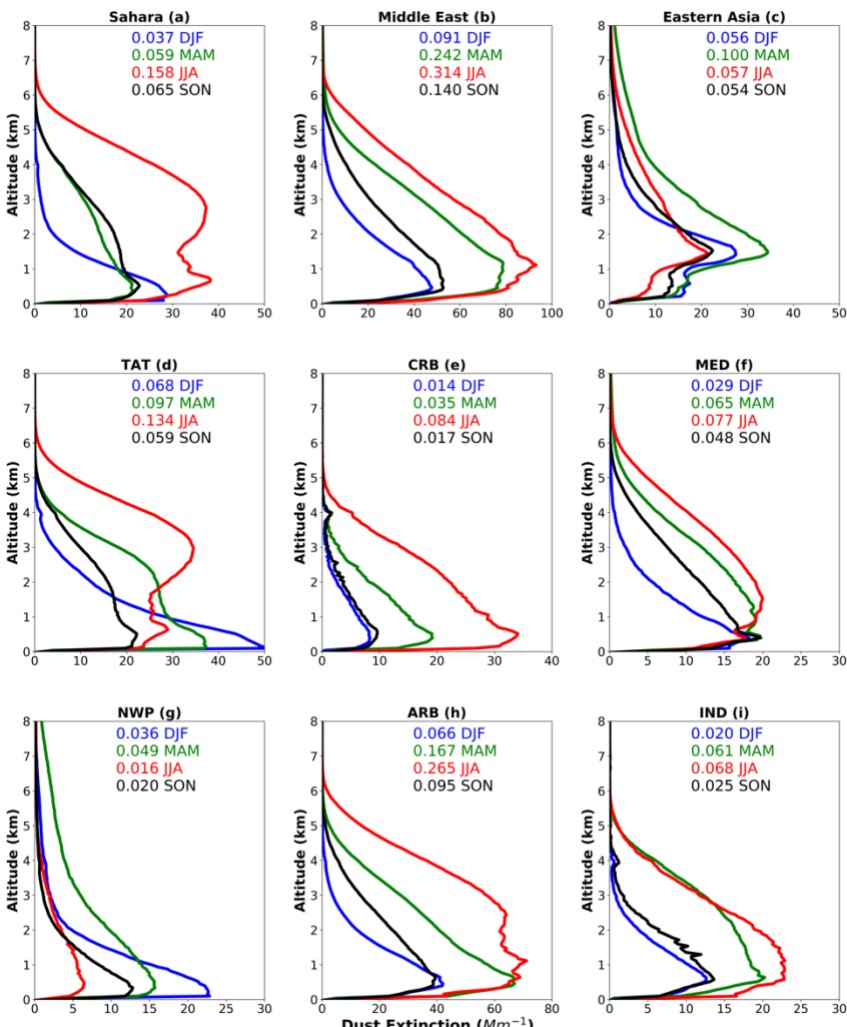

Figure 11. Vertical profiles of seasonal mean dust extinction coefficient (Mm⁻¹) in 9 dust-laden regions indicated in Figure 4. Different colors represent different seasons. The numbers on each plot are the seasonal mean DAOD for the region.

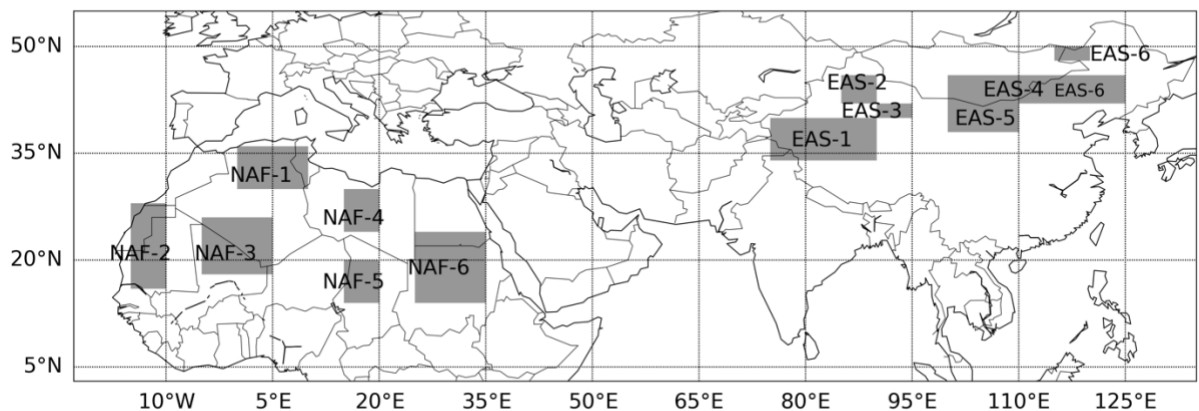

Figure 12. Six dust potential source subregions in Northern Africa (NAF) and Eastern Asia (EAS) based on Fig.1. and Fig. 2. in Formenti, et al., 2011. PSA NAF-1(30N-36N, 0-9E), PSA NAF-2 (16N-28N, 10W-15W), PSA NAF-3 (18N-26N, 5W-5E), PSA NAF-4 (24N-30N, 15E-20E), PSA NAF-5 (14N-20N, 15E-20E), PSA NAF-6 (14N-24N, 25E-35E);  EAS-1: (34N-40N, 75E-90E) ; EAS-2: (44N-46N, 85E-90E); EAS-3: (40N-42N,90E-95E and 42N-44N, 85E-90E); EAS-4: (42N-46N, 100E-115E); EAS-5: (38N-42N, 100E-110E); EAS-6: (42N-46N, 115E-125E and 48N-50N, 115E-120E)

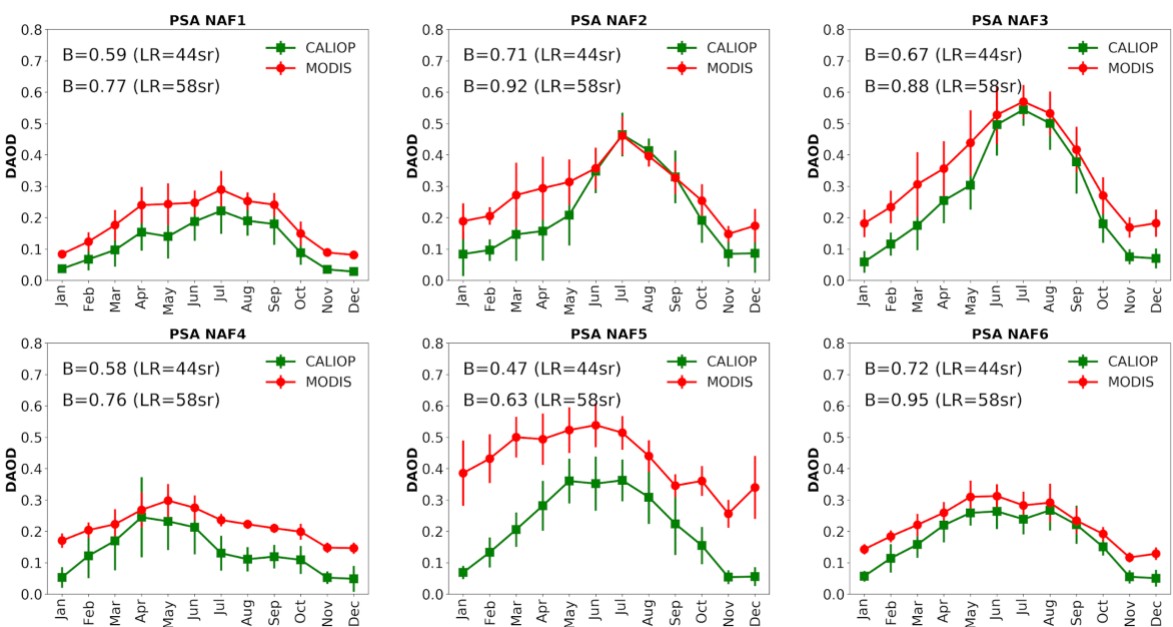

Figure 13. Annual cycle of 13-year (2007-2019) monthly mean DAOD over the six PSAs of North African dust. The CALIOP DAOD annual cycle shown in the figure is derived from backscatter coefficients using LR of 44sr. The mean bias (B) is computed as the average of CALIOP DAOD / MODIS DAOD ratios of all data pairs. B =1, >1, <1 indicates no bias, high bias and low bias. The mean bias (B) associated with CALIOP DAOD based on LR=44sr and 58sr are shown in the upper left of each panel.

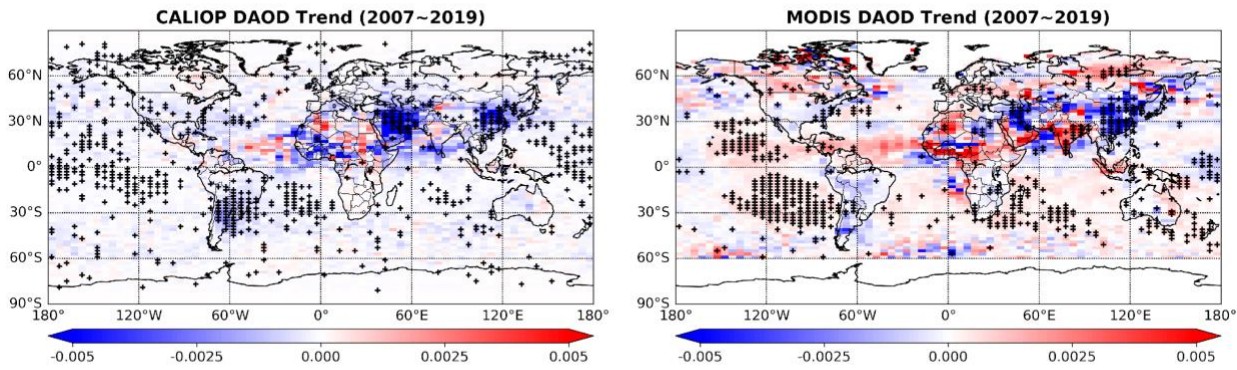

Figure 14. Global map of DAOD trend based on CALIOP (left) and MODIS (right) dust climatology data over 2007-2019 period. Red and blue represents increasing and decreasing trend, respectively. Symbol '+' denotes trends with p-value < 0.05, which are considered as statistically meaningful trend.

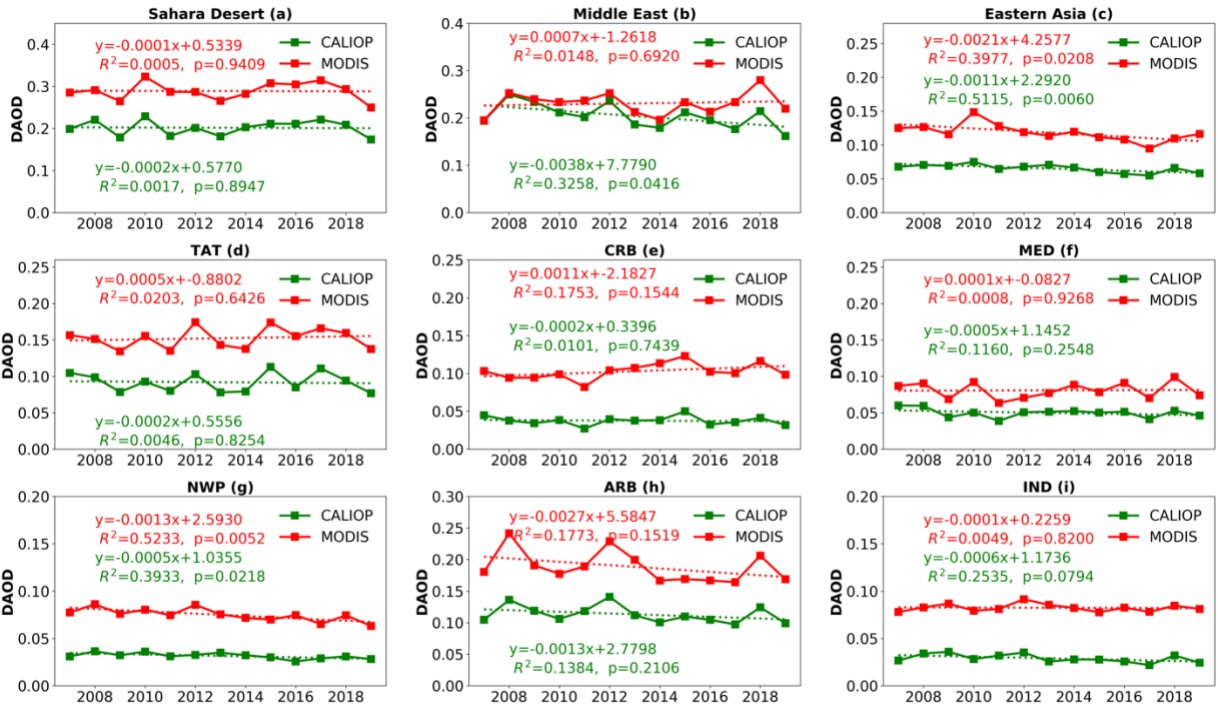

Figure 15. DAOD interannual variability over main dust source regions (a-c) and dust outflow regions (d-i) revealed by CALIOP (green curve) and MODIS (red curve) observations.

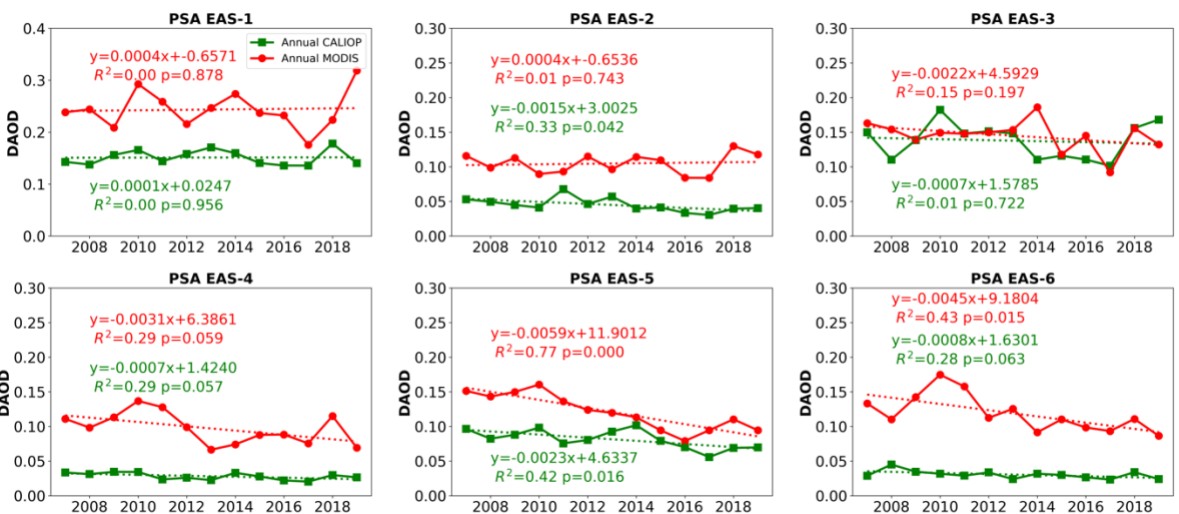

Figure 16. Interannual variability of CALIOP (green) and MODIS (red) DAOD in the six potential dust source areas in Eastern Asia (refer to Figure 8).

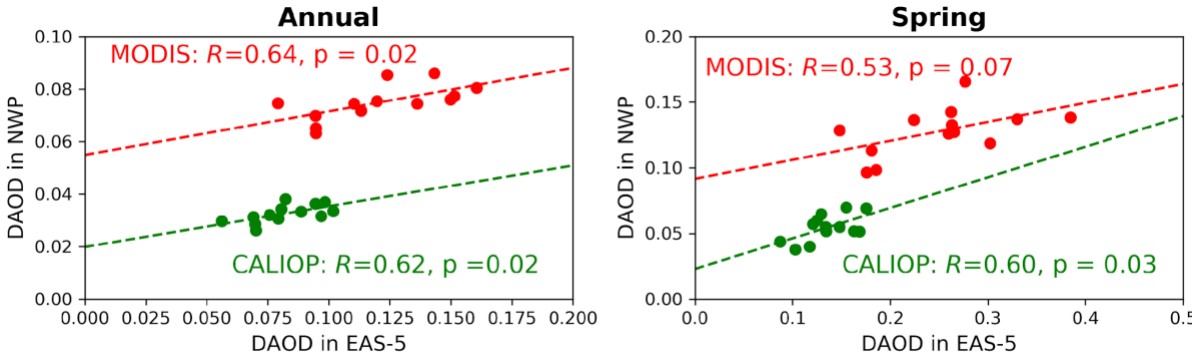

Figure 17. Correlation between DAOD in EAS-5 (Southern Gobi Desert) and DAOD in NWP for annual mean (left) and springtime average (right).

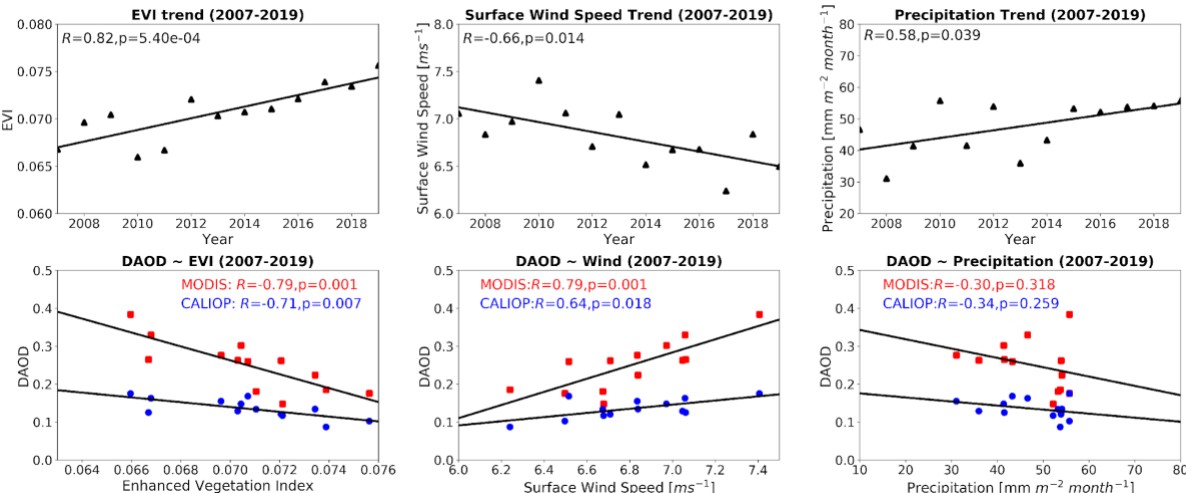

Figure 18. The trend of Enhance Vegetation Index (EVI), surface wind speed and precipitation and their correlation with DAOD in spring, EAS5 region. The 1ˢᵗ row shows the trend of EVI, surface wind speed and precipitation. The 2ⁿᵈ shows the correlation of EVI, surface wind speed and precipitation with MODIS-based DAOD and CALIOP-based DAOD respectively. In addition, the time series of EVI versus DAOD, wind speed versus DAOD, precipitation versus DAOD is shown in Figure S10 in the supplement.

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
