# Peer review of "Global Dust Optical Depth Climatology Derived from CALIOP and MODIS Aerosol Retrievals on Decadal Time Scales: Regional and Interannual Variability"

_Atmospheric Chemistry and Physics, 2021_

## Referee Comment (RC1)

Review of "Global dust optical depth climatology derived from CALIOP and MODIS aerosol retrievals on decadal time scales: regional and interannual variability" by Song et al. for ACPD

This article reports an analysis of both CALIOP and MODIS retrievals spanning over a decade in order to compile climatological records of the spatial variability of dust aerosol optical depth and its interannual (and inter-decadal) variability. For the CALIOP record, the authors also obtained vertical profiles of extinction efficiency using the methodology previously developed in Yu et al. (2015). Overall, this is insightful work that could be an important contribution to the field and the paper was a pleasure to read. I do have a few major comments relating to the methodology and the interpretation of the results. The article thus requires major revisions.

Main comments:

- There is a very large difference in the DAOD obtained from the CALIOP (0.029 on a global basis) and the MODIS record (0.063 on a global basis). The MODIS result of a global DAOD of 0.063 is about one-and-a-half to two times other estimates in recent papers (Ridley et al., 2016; Voss and Evan, 2020; Gkikas et al., 2021), and much larger than simulated by any global aerosol model (see Figure 3 in Gliss et al., 2021). Therefore, I think the authors should include more discussion of what is causing their MODIS climatology to yield a much larger DAOD than other literature estimates. That should include a more detailed error analysis, and particularly an analysis of errors affecting the MODIS estimate such as errors in the DAOD/AOD ratio (see further comments below).
- A key part of the methodology that is novel is obtaining the ratio of DAOD to AOD from MODIS data, which allows the authors to obtain a climatology of DAOD from well-established MODIS Aqua retrievals of AOD. However, more details should be provided of the procedure for obtaining DAOD/AOD, especially as errors in this procedure seem to be a likely culprit of the large differences between the CALIOP-based and the MODIS-based climatology (e.g., lines 427-9).
  - It seems that the DAOD/AOD ratio over land was calculated using the previous analysis in Pu and Ginoux (2018). Considering how important this is for the final results, this method should be summarized and explained here, and its strengths and limitations discussed.
  - It's briefly mentioned that the methodology of Yu et al. (2020) was used to obtain the DAOD/AOD ratio over ocean (line 280). Please similarly explain this method in sufficient detail for the reader to understand this.
- Related to the previous comments is that the authors should expand the discussion in Section 4 to propagate their uncertainties into their final results, obtaining error bars on the DAOD climatologies from both MODIS and CALIPSO. For CALIPSO, paragraph 1 includes an error estimate due to assumptions about the dust depolarization ratio; it seems straightforward to propagate the error of the lidar ratio into this estimate and perhaps from CALIPSO's low sensitivity (an error estimate for that is also discussed earlier in the paragraph). For MODIS, there is some quantitative analysis of errors mentioned near the end of section 4 but more

detail should be added. This would be a combination of the well-characterized errors in the AOD retrieval and errors in the DAOD/AOD ratio.

- Line 352-4: Here and in the rest of this section, the suggestion is made that CALIPSO's lower sampling frequency could be a cause for the systematically lower DAOD from CALIPSO than from MODIS. I think this is incorrect. A lower sampling frequency is just as likely to produce an overestimate of DAOD as an underestimate, so producing a systematic underestimation of a factor of two for many different regions is in my view implausible. Please correct this.
    - Line 370-1: I think this statement is similarly statistically incorrect. Why would episodic sampling of dust events produce a systematically lower DAOD, especially when you have 12 years of data and so many total retrievals?

Other comments:

- The abstract notes a climatological record over the last two decades, but the period spanned is really 2007-2019. Please correct.
- Lines 242-4: Yu et al. (2021) recently did a detailed analysis of the diurnal variability of dust AOD that should be mentioned here. And can you roughly estimate the error you expect from the difference in daytime and nighttime distributions based on Yu et al. and Kittaka '11? I think this should be small (~10%?) relative to your other errors.
- Line 247-9: This statement seems a bit vague. Could you be specific as to how large you expect the solar noise to be? If that's larger than 10% then the statement would sound less subjective to the reader.
- Line 270-2: This sentence is unclear. Could you include a supplementary graph on how sensitive your results are to this criteria of 10 retrievals per month?
- Section 3.1 is very long and a bit difficult to read. I recommend adding sub-sections to improve readability.
- Line 317-8: could you explain why the coarse size of dust contributes to the positive depolarization ratio?
- Line 319-20: If DAOD over ocean is derived from TAOD and FMF, then how do you remove the AOD due to sea salt? See my comment above regarding a need for more detail on how the DAOD/AOD ratio is calculated for MODIS.
- Is the global DAOD you report weighted by the area of each grid point? It'd be helpful to note this somewhere.
- Line 384-6: A recent paper by Huang et al. (2020) showed that Asian dust indeed gets more spherical during transport so might be good to include here.
- Line 438-9: Please elaborate on how specifically the seasonal cycles of dust emission and transport are consistent with previous results.
- Line 480-4: Here and elsewhere, the authors include some insightful explanations on how errors in cloud screening affect both CALIOP and MODIS retrievals. Are you able to quantify the difference in the AOD between CALIOP and MODIS that is due to these errors from simultaneous retrievals from both?
- Line 592: should be "desert" not "dessert" :)
- Line 610: what do you mean by appropriate, exactly?

- Line 614: here and elsewhere, could you define what you mean by "coarse mode", exactly? What particle size range does that refer to? Definitions differ for that differ in the literature.
- Line 615: although fine dust accounts for a small fraction of the mass, it accounts for a larger fraction of the DAOD, so this statement should be corrected.
- Line 616-8: this is helpful information. Could you be clear here what the bias is of the Aqua MODIS DAOD relative to AERONET, and what the error is? And since coarse mode includes sea salt, is this a one-to-one comparison?
- Line 649: could you be quantitative here about the correlation? I think whether MODIS and CALIOP-based DAOD "correlate well" is subjective and I personally was surprised the correlation was not higher.
- Line 653: I think here k is the correlation and $R^2$ is the variance explained.

**References**

Gkikas, A., et al., 2021. ModIs Dust AeroSol (MIDAS): a global fine-resolution dust optical depth data set
Gliss, J., et al., 2021. AeroCom phase III multi-model evaluation of the aerosol life cycle and optical properties using ground- and space-based remote sensing as well as surface in situ observations. Atmospheric Chemistry and Physics 21, 87-128.
Huang, Y., et al., 2020. Climate Models and Remote Sensing Retrievals Neglect Substantial Desert Dust Asphericity. Geophys. Res. Lett. 47, e2019GL086592.
Ridley, D.A., Heald, C.L., Kok, J.F., Zhao, C., 2016. An observationally-constrained estimate of global dust aerosol optical depth. Atmos. Chem. Phys. 16, 15097-15117.
Voss, K.K., Evan, A., 2020. A New Satellite-Based Global Climatology of Dust Aerosol Optical Depth. Journal of Applied Meteorology and Climatology 59, 83-102.
Yu, Y., et al., 2021. A global analysis of diurnal variability in dust and dust mixture using CATS observations. Atmospheric Chemistry and Physics 21, 1427-1447.

---

## Referee Comment (RC2)

**Comments on "Global Dust Optical Depth Climatology Derived from CALIOP and MODIS Aerosol Retrievals on Decadal Time Scales: Regional and Interannual Variability" by Song et al.**

**General comments:**

Dust is an important aerosol type in the atmosphere and has significant impacts on environment and climate. Satellite is a very useful tool to detect dust aerosol as it can provide dust distribution of large spatial coverage with long-term duration. A unique lidar platform (CALIOP) can further provide the vertical distribution of dust aerosol. This study provides a 13-year global dust optical depth (DAOD) climatology derived from CALIOP and MODIS observations and presents some interesting results for global dust spatial distribution and temporal variations. The authors also investigate the reasons for the recent decline of dust activities in East Asia. The results demonstrate the reliability of the two datasets the authors have developed.

However, the authors seem to mention a lot of uncertainties in the dataset and it seems there is no answer for a better DAOD. I personally would like to see more convincing quantitative comparison of these two datasets, such as which one is better and it is possible to suggest a more reliable global DAOD. I suggest the authors make more clear conclusions relative to the comparison.

**Major comments:**

Section 2 and 3: I am curious about the systematic difference between CALIOP and MODIS. In my understanding, the main reasons may be in the algorithms used to generate the product and the difference may be minimized if calibrated to the same data sources. I am wondering if there is some way to minimize the difference between different dataset. The authors seem to mention a lot of uncertainty and it may be helpful to separate the contributions from these factors. First, is it possible to compare the two dataset at the same time and close location (although the sample number may be small)? Second, after doing this comparison, the difference of monthly/seasonal mean can be due to the sampling and aggregate methods. I think the authors can provide a map of observation numbers for different months/seasons and a month with more temporal coverage is more likely to have a more reliable statistical result.

Section 2.2: the comparison of nighttime CALIOP and daytime MODIS product is made based on the consideration "Kittaka et al., 2011 shows that daytime and nighttime global seasonal-mean AOD distributions for JJA 2006 from CALIOP are generally similar in both outflow and source regions (see their Figure 1)." (Lines 242-244). First, Kittaka's conclusions are based on the global distribution, while this present study is specifically on dust source regions and dust outflow regions. Kittaka's conclusions should not be simply applied to present study. Second, previous studies have shown significant diurnal variations of dust event frequency in dust sources regions (e. g, Figure 17 of Yue, X., H. Wang, Z. Wang, and K. Fan (2009), Simulation of dust aerosol radiative feedback using the Global Transport Model of Dust: 1. Dust

cycle and validation, J. Geophys. Res., 114, D10202, doi:10.1029/2008JD010995.).

Line 647, in Summary and Conclusions: it seems the trends and interannual variability of DAOD are similar. I don't see the advantages and limitations clearly for each dataset. Please clarify.

**Specific comments:**
Line 21, abstract: add "(2007-2019)" after "the last two decades".

Line 27, abstract: delete "and".

Line 127: Self-consistent: Please briefly explain the word here, and not wait until Section 2.

Lines 131-132: is it any critical difference for dust between these version and previous version? This is important, as it may indicate the results shown in this study may be different from previous versions because of version updates.

Line 155: please provide some supporting information for "70% agreement".

Lines 162-163: if 40 sr is too low, is it possible to increase this value and update the product? What is the lidar ratio?

Line 174: I don't understand "or" here. Please clarify.

Sections 2.1 and 2.2: is it possible to make a table and put the comparison of key features of CALIOP and MODIS in it?

Line 242: Kittaka et al. (2011) also analyze the AOD including all aerosol types. I am wondering whether their conclusions applied to specific dust source regions and dust-effect regions.

Line 248: "in hoping that": this statements may be misleading. I think the key point is that observations do shows some significant diurnal of dust events in the source regions mainly because of the wind speed difference. Please clarify.

Line 268: I am wondering whether this new data includes all the algorithms mentioned in previous paragraph.

Line 272: how about the sampling over land? Is there a minimum number required for deriving monthly statistics?

Lines 279-280: is there any measure to keep the ocean and land product consistent with each other?

Section 3.1: this section is too long. Consider adding subtitle or dividing it into two sections.

Lines 313-314: I don't think it is because of more frequent miss but it is an expected result of using a threshold to detect dust.

Line 368: I don't think this can suggest dust activities occur relatively small scales. Since a threshold is applied in conditionally DAOD, it is easily to understand conditionally DAOD have much higher values than climatological DAOD.

Line 401: I can't see this "exception". Please check.

Lines 446-448: this can explain the secondary peak of DAOD in summer for MODIS, but it can't explain the difference of two datasets. Please clarify.

Line 480-495: it seems the authors suggest a true DAOD should fall between CALIOP and MODIS. If so, please explicitly state this in the text.

Line 523-524: broad East Asian region (ESA defined in Figure 4): I can't find ESA in Figure 4. Please clarify.

Line 531: change "DOAD" to "DAOD".

Line 535: I don't think it is "much weaker".

Line 538: please explain a little bit why EVI, MERRA2 near surface (at 10m) wind speed and precipitation are reliable for this analysis. Probably cite some references which have already demonstrated this.

Line 541: add "p>0.05" after "precipitation", as the determination of significance depends on the level of significance. I guess here you are using p=0.05 as a threshold.

Line 601: please expand DPR.

Line 603: I am wondering how the uncertainty is defined. Please add a definition.

Line 618: is it also applied to this study? If so, I am wondering whether a true DAOD is even higher than MODIS DAOD. Please clarify.

Line 640: it is possible that CALIOP misses some dust events. But this study is based on long-term statistics and the impacts should be eliminated if the observation number are substantial large. Do the authors suggest the impacts are not negligible in this study?

Line 664: add "during 2007-2019" after "DAOD".

Line 780: avoid using "decadal trend" here as there are only 13 years indeed.

Line 782: the symbol "+" can't be clearly seem.

Line 805: it is also helpful to put together the time series of EVI vs DAOD, wind vs DAOD, precipitation vs DAOD with the year for x-axis. This can be put in the supplement for references.

Lines 935-938: a duplicated reference.

---

## Community Comment (CC1)

Dear authors,

While your study is of high interest, I have the feeling that you miss some important literature on the same subject and moreover, many of the aspects mentioned should be revised, focusing specifically on the following points that I see from a first reading:

1. Differences between the CALIOP and MODIS global DODs are large. Is there any explanation about this discrepancy? Please note that the MODIS-derived global DOD is substantially higher than those reported in most of the recently published works (e.g., Ridley et al., 2016; Voss and Evan, 2020; Gkikas et al., 2021). A description is needed on how the global averages have been computed for both sensors. Do you acknowledge any weighting factors based on the grid cell surface area? According to Levy et al. (2009), the approach for the calculation of the global DOD is quite critical (see Fig. 5). Summarizing, I recommend including a table providing the corresponding global DODs given by relevant studies (relied either on observations or models) in order to check (and discuss) the consistency of your findings.

2. The manuscript could greatly benefit by previous studies that have performed similar analysis. For instance, the authors mention the climatological and conditional dust products, which have been introduced for the first time in Marinou et al., (2017) and then applied on Proestakis et al., (2018). No discussion or comparison is presented in the manuscript. Moreover, the separation methodology used in the manuscript has been extensively implemented in the framework of EARLINET (e.g. Tesche et al., 2009, 2011; Ansmann et al., Ansmann et al., 2011). Furthermore, Amiridis et al., (2013) introduced for the first time the depolarization-based separation methodology on CALIPSO. However, there is no reference or discussion on this study as well! Given that all the aforementioned studies are available in the literature, which are the innovative aspects of the present study?

3. Lines 105-109: Please update the information based on the final paper version of Gkikas et al. (2021) in which the MODIS-Aqua Collection 6.1 data, over the period 2003-2017, have been used.

4. Lines 251-264: A short description of the applied techniques for the derivation of DOD is needed, based on MODIS, over continental and marine regions. How much feasible is to discriminate mineral particles from sea-salt over oceans relying only on size parameters? It is not clear to me how you can separate dust from sea-salt over land using a very high single scattering albedo (almost equal to 1; similar to those recorded for sea-salt particles) and ignoring its spectral variation. Moreover, how much reliable the Ångström exponent is above land (see Section 4.4.5 in Levy et al. (2013))? Are you using only Deep Blue retrievals over land? In this case, how do you discriminate dust aerosols from other types when the Dark Target algorithm it is applied?

5. Section 3.1: Since you are using CALIPSO and Aqua retrievals, you can collocate them in order to eliminate the impact of the different sampling between the two satellite sensors which are flying in the A-Train constellation. Taking advantage of the almost coincident observations you can assess the assumptions made in Lines 394 – 407.

6. Trend analysis: I cannot understand why you put so much focus in EAS and NWP without discussing other regions of the planet (e.g. Middle East).

7. Uncertainty analysis: It would be important to present global maps of the DOD uncertainty both for CALIOP and MODIS in order for the reader to better understand how uncertain the obtained DOD averages are.

8. Lines 619-627: I don't agree with this statement. It is true that it is not easy to evaluate DOD retrievals against AERONET because the sun-photometric measurements are representative for the entire atmospheric column. Nevertheless, you can select either sites (even though are few of them) in desert areas (the contribution of other aerosol species is minor or negligible), or to set appropriate coincident thresholds on AOD and Ångström exponents (see for example Basart et al. (2009)) or to rely on almucantar retrievals (Gkikas et al., 2021) or to follow the approach that you are mentioning in your manuscript (Pu and Ginoux, 2018). In any case, an evaluation analysis it is needed in order to support the reliability of the satellite DODs (see also Schuster et al., 2012; Amiridis et al. 2013).

9. Table 1: Are you using the spectral SSAs or only the values at 470 nm?
10. In the manuscript, dust is distinguished from non-dust aerosols based on particle shape information (i.e., the use of particulate depolarization ratio) for CALIOP. However, the particulate depolarization ratio in L2 is too noisy, showing values for dust, dusty marine, polluted dust aerosol subtypes from negative up to 1.0 and above (see figure below).

[Figure]

Moreover, approximately 11% of all dust, dusty marine, polluted dust aerosol subtypes have particulate depolarization ratios < 0.05. Since in the methodology the dust, dusty marine, polluted dust aerosol subtypes are assumed mixtures of dust and non-dust components, how do the authors treat the negative and larger-than-one particulate depolarization cases in their Quality Assurance procedure? Do the authors consider the dusty aerosol mixtures of particulate depolarization ratio lower than 0.05 as non-dust mixtures? Which are the uncertainties introduced in the final dust product by these values? Please quantify.

11. The authors provide a CALIPSO-based dust product, based on the particulate depolarization ratio, applied to L2 backscatter coefficient profiles. Based on the manuscript it is not clear whether the methodology is applied only on the dust, dusty marine, and polluted dust aerosol subtypes, and not at the other types (e.g. elevated smoke, marine, …) at the 60m aerosol layer. Or whether an average over consecutive 60m layers is computed to remove noise. Please provide more in-depth description of the selected methodology. Moreover, which is the effect of the identified aerosol subtype misclassification on the dust product? Many important studies are mentioned by the authors (e.g. Burton et al., 2013), however the effect of the misclassification on the dust product needs discussion and quantification.

12. Based on the methodology, the dust, dusty marine, and polluted dust aerosol mixtures are distinguished into a dust and a non-dust component. Thus, at the end, there are three types of backscatter coefficient: (1) the initial backscatter coefficient of non-dust mixtures (e.g. elevated smoke, …), (2) the dust backscatter coefficient of the separated dust component, and (3) the remaining backscatter coefficient of the separation, the non-dust component. According to my understanding the extinction coefficient of (1) does not change since the methodology is not applied to non-dust mixtures. Regarding the case (2), a uniform global Lidar Ratio (LR) is implemented to calculate the dust extinction coefficient. However, the authors do not discuss the case three (3), regarding the remaining backscatter coefficient of the non-dust component. For the calculation of the non-dust extinction coefficient component, the authors should identify the non-dust aerosol subtype in the dusty aerosol mixture, in order to assign a proper LR. The authors have not provided a detailed explanation. Since the AOD is then computed by the integration of the extinction coefficient profile, the authors should either provide a solid justification of the non-dust aerosol-subtype assignment including quantification the corresponding uncertainties, or to avoid using the new AOD and the corresponding Sections, after the intermediate dust separation.

13. It is not properly discussed, how the averaging extinction coefficient procedure is computed, prior to integration for the DAOD. According to Amiridis et al. (2013) and Tackett et al. (2018), the methodology should follow first a "per-overpass" averaging within a specific grid, and accordingly integration of the mean profile, calculated by all overpasses in the grid. However, the methodology followed by the authors is not clear in this point. Please discuss, and in case a different methodology is provided justify the selected approach or revise accordingly.
14. The manuscript would greatly benefit by introducing tables of the Quality Assurance procedures, applied to both CALIPSO and MODIS, including the corresponding literature related to each filter.
15. What I am missing in the study is a validation intercomparison against ground reference lidar instruments to validate the profiles acquired (e.g. EARLINET/ACTRIS), or even an intercomparison against dust models.
16. The uncertainty analysis is not performed in-depth. Many aspects, such as the effect of non-uniform global Lidar Ratio, the presence of highly polarizing pollen, the presence of volcanic particles or the effect of depolarizing marine particles (in Low RH), the effect of topography and orography (e.g. weighting effects on the mean profiles due to mountains), negative or high positive backscatter values and how they are treated (including references) are not discussed and quantified through a proper error-propagation analysis and an estimation of the uncertainties.

---

## Author Comment (AC1)

**Responses to the RC1**

This article reports an analysis of both CALIOP and MODIS retrievals spanning over a decade in order to compile climatological records of the spatial variability of dust aerosol optical depth and its interannual (and inter-decadal) variability. For the CALIOP record, the authors also obtained vertical profiles of extinction efficiency using the methodology previously developed in Yu et al. (2015). Overall, this is insightful work that could be an important contribution to the field and the paper was a pleasure to read. I do have a few major comments relating to the methodology and the interpretation of the results. The article thus requires major revisions.

**Reply:** We would like to thank the reviewer for constructive comments. We have taken these comments seriously and revised the manuscript accordingly. An item-to-item reply to the reviewer's comments is provided in this response.

Before addressing the comments/questions in detail, we would like to first provide a summary of the major revisions made to the manuscript:

- We added more detailed discussion regarding MODIS DAOD retrieval methodologies over ocean and land in section 2.2.

- We compared our MODIS- and CALIOP-based DAOD with values reported in previous studies. The comparison is added to the revised manuscript as section 3.1. For MODIS DAOD comparison, we compare our results with previous studies in both global and regional scales; For CALIOP, there isn't global CALIOP-based DAOD retrievals to compare our result with, therefore, the comparison is limited to regional scale. Overall, these comparisons suggest that our results are in reasonable agreement with previous studies, except for Voss and Evan 2020 over ocean (which can be explained by the use of different parameterization schemes).

- We evaluated our monthly mean MODIS- and CALIOP-based DAOD product by comparing with AERONET monthly mean coarse mode AOD (COD) from 2007 to 2019. We found that MODIS DAOD is statistically higher than AERONET COD by 26.7% over land and 18.5% over ocean, while CALIOP DAOD is lower than AERONET COD by 27.9% over land and 35% over ocean. This may suggest that the true DAOD probably fall between MODIS and CALIOP DAOD retrievals. Furthermore, by following the methodology proposed by Sayer et al. 2013, we estimated that the absolute expected error of MODIS DAOD is $0.65 \times DAOD_M + 0$ over land and $0.50 \times DAOD_M + 0$ over ocean, the absolute expected error of CALIOP DAOD is $0.52 \times DAOD_C + 0.02$ over land and $0.54 \times DAOD_C + 0.02$ over ocean. This analysis was added in section 3.2.

After these revisions, we think the paper is much improved and more focused, although the general conclusions still hold.

**Main Comments:**

Q1. There is a very large difference in the DAOD obtained from the CALIOP (0.029 on a global basis) and the MODIS record (0.063 on a global basis). The MODIS result of a global DAOD of 0.063 is about one-and-a-half to two times other estimates in recent papers (Ridley et al., 2016; Voss and Evan, 2020; Gkikas et al., 2021), and much larger than simulated by any global aerosol model (see Figure 3 in Gliss et al., 2021). Therefore, I think the authors should include more discussion of what is causing their MODIS climatology to yield a much larger DAOD than other literature estimates. That should include a more detailed error analysis, and particularly an analysis of errors affecting the MODIS estimate such as errors in the DAOD/AOD ratio (see further comments below).

**Reply:** These are great questions and suggestions and also thanks for the references. Following your suggestions, we added a new section (i.e., section 3.1) in the revised manuscript to compare our study with previous ones and discuss the causes for differences. We also included several new tables (Table 2 and Table S2, S3) and figures (Figure S2, S3, S4 and S5) in both manuscript and supplementary material to support the discussion.

For global scale comparison, the (new) Table 2 in the revised manuscript lists the global mean DAOD from previous studies and our study. Ridley et al. 2016 used multiple satellite platforms (MODIS and MISR), in-situ AOD observations and four global models to estimate global mean DAOD over 2004 ~ 2008. Gkikas et al. 2021 used AOD from Aqua MODIS and DOD-to-AOD ratio from MERRA2 to estimate global mean DAOD over 2003~2017. In contrast, as shown in Table 2 our MODIS-based global (90°S~90°N) DAOD is 0.057. However, it is important to note that the global mean DAOD values from these studies are not directly comparable to our global mean results because of the methodology differences. In particular, both of aforementioned studies used model simulations to aid their global DAOD estimate, while our estimates are completely based on observations (More precisely, DAOD of the scope 60°S~60°N are completely based on observations, while outside of the scope, DAOD is assumed to be zero). In contrast, Voss and Evan 2020 (referred to VE20) used similar methods to our MODIS-based methodology and limited the global scope to 50°S~60°N, this is directly comparable to our global (60°S~60°N) mean MODIS DAOD values listed in Table 2. Below we focused on explaining the difference between our MODIS-based DAOD and values reported in Voss and Evan 2020.

As shown in (new) Table 2 of the revised manuscript, our DAOD based on MODIS over land (DAOD=0.103) is almost identical with that in VE20 (DAOD=0.1). Over ocean, our MODIS-based result (DAOD=0.055) is significantly larger than VE20 (DAOD=0.03). As we mentioned before, VE20 and our MODIS-based DAOD retrieval used the similar method. However, different parameters are used in the two MODIS over ocean retrieval methodologies, which is the main reason causing the non-negligible difference in our over-ocean mean DAOD. As shown in Eq (2) and Eq (3) in the revised manuscript, $f_c, f_d, f_m$ are required to estimate DAOD. We use MODIS over ocean retrievals to determine $f_c, f_d, f_m$ and hence derive DAOD in self-consistent manner, while Voss and Evan (2020) determine these parameters based on observations in AERONET stations dominated by each aerosol type. The use of different characteristic parameters in the estimation of DAOD over ocean is the main reason for the difference between the two studies.

As we explained in the supplementary materials, after combining Eq (2) and Eq. (3) in the manuscript, we get the following equation for DAOD over ocean:

$$\tau_d = \frac{(\tau - \tau_m)f_c + \tau_m f_m - \tau f}{f_c - f_d} \ , \tag{1}$$

, where $f_c, f_d, f_m$ are the fine mode fraction of combustion, dust and marine aerosols, respectively. The values for these parameters used in our study and in VE20 are listed in the Table S4 of the supplementary material. It turns out that we used significantly larger $f_c$ and $f_m$, while a slightly smaller $f_d$, in comparison with VE20. Because the derived DAOD is positively proportional to these parameters, the use of larger $f_c$ and $f_m$, is probably the reason for a larger DAOD in our study.

Moreover, for regional scale comparison, we compared regional mean DAOD in Ridley et al. 2016 and Proestakis et al. 2018 with our results. As we discussed before, the global mean DAOD in Ridley et al. 2016 is not directly comparable to our results due to the different retrieval methodology. To compare with Ridley et al. 2016, we selected the same 14 dust-laden regions provided in their paper (see their Figure 1) and plotted our DAOD results with the values reported in their Table 3. As aforementioned, in Ridley et al. 2016 the DAOD in these dust-laden regions is constrained by AERONET measurements and satellite retrievals, and therefore more comparable with our results. The comparison plots are provided in the Figure S2 and S3 of the supplementary material. Overall, our MODIS-based DAOD agrees very well with their results. Note that in Ridley et al. (2016), DAOD outside of the dust-laden regions came largely from model simulations and imperfect parameterizations of dust transport and removal processes in the model may have led to significant difference from our MODIS-based DAOD. Table 1 in Proestakis et al. (2018) provides domain mean DAOD for six regions in Asia based on CALIOP observations. We selected the same 6 regions in East Asian and compared the regional mean DAOD between the two studies. As shown in Figure S4 and S5 of the supplementary material, values from Proestakis et al. (2018) are in excellent agreement with our CALIOP-based DAOD.

Overall, these regional and global comparisons suggest that our results are in reasonable agreement with previous studies, except for VE20 over ocean (which can be explained by the use of different parameterization schemes). On the other hand, the comparison results also reveal that MODIS-based DAOD is generally larger than CALIOP-based DAOD (See Figure S3 and S5 the supplementary materials). But the two methods were not systematically compared in previous studies, which has motivated us to carry out this study.

Q2. A key part of the methodology that is novel is obtaining the ratio of DAOD to AOD from MODIS data, which allows the authors to obtain a climatology of DAOD from well-established MODIS Aqua retrievals of AOD. However, more details should be provided of the procedure for obtaining DAOD/AOD, especially as errors in this procedure seem to be a likely culprit of the large differences between the CALIOP-based and the MODIS-based climatology (e.g., lines 427-9).

- It seems that the DAOD/AOD ratio over land was calculated using the previous analysis in Pu and Ginoux (2018). Considering how important this is for the final results, this method should be summarized and explained here, and its strengths and limitations discussed.
- It's briefly mentioned that the methodology of Yu et al. (2020) was used to obtain the DAOD/AOD ratio over ocean (line 280). Please similarly explain this method in sufficient detail for the reader to understand this.

**Reply:** Thanks for the suggestion. we have explained the methods in the manuscript to help readers understand the methodology and facilitate discussion of strengths and weaknesses of the methods.

For MOIDS over-ocean DAOD retrieval, an approach was developed in previous studies to separate DAOD from other types of aerosol by using aerosol optical depth ($\tau$) and fine mode fraction (f) retrieved from MODIS Dark Target retrieval over ocean. Both $\tau$ and $f$ refer to properties at 550 nm hereafter, unless specified otherwise. In this approach, both $\tau$ and fine-mode AOD ($f\tau$) are assumed to be composed of marine aerosol, dust and combustion aerosols, i.e.,

(2)
$$\tau = \tau_m + \tau_d + \tau_c \ ,$$

(3)
$$f\tau = f_m\tau_m + f_d\tau_d + f_c\tau_c \ ,$$

Where the subscripts m, d, and c represent marine aerosol, dust and combustion aerosol, respectively. Based on Eq. (2) and (3), $\tau_d$ can be calculated from MODIS-retrieved $\tau$ and $f$, with appropriate parameterizations for $f_m, f_d, f_c$ and $\tau_m$. More specifically, $f_m, f_d, f_c$ were determined from retrieved $f$ in selected regions and seasons for which a specific aerosol type dominates, $\tau_m$ was parameterized as a function of wind speed (details can be found in Kaufman et al., 2005; Yu et al., 2009, 2020).

Over land, MODIS aerosol properties including AOD, Ångström exponent, SSA are retrieved from the Deep Blue (DB) algorithm. DAOD over land is derived from the AOD using one criterion based on size distribution (to distinguish fine and coarse modes) and the other criterion based on absorption (to distinguish between scattering sea salt and absorbing dust). To apply first criterion, we use the following formula depending quadratically on the Ångström exponent ($\alpha$), a measure of the wavelength dependence of optical depth or extinction (Eck et al., 1999):

(4)
$$COD_M = AOD \times (0.98 - 0.5089\alpha + 0.051\alpha^2) \ ,$$

Where $COD_M$ is the coarse mode fraction (aerodynamic diameters larger than $1\mu m$) of AOD retrieved from MODIS, with a contribution from absorbing (DAOD) and scattering aerosols (sea salt aerosol optical depth). This relationship is derived from the formula of Anderson et al. 2005 derived from in-situ data. The second criterion requires the single-scattering albedo at 470 nm to be less than 0.99 for the retrieval of DAOD (more details can be found in Pu and Ginoux, 2018). This description was added in the revised manuscript (section 2.2).

Q3. Related to the previous comments is that the authors should expand the discussion in Section 4 to propagate their uncertainties into their final results, obtaining error bars on the DAOD climatology from both MODIS and CALIPSO. For CALIPSO, paragraph 1 includes an error estimate due to assumptions about the dust depolarization ratio; it seems straightforward to propagate the error of the lidar ratio into this estimate and perhaps from CALIPSO's low sensitivity (an error estimate for that is also discussed earlier in the paragraph). For MODIS, there is some quantitative analysis of errors mentioned near the end of section 4 but more detail should be added. This would be a combination of the well-characterized errors in the AOD retrieval and errors in the DAOD/AOD ratio.

**Reply:** Good point. It is important to quantify uncertainties associated with our MODIS- and CALIOP-based DAOD.

In CALIOP-based DAOD retrieval, we assume dust lidar ratio (LR) to be $44 \pm 9 \ sr$ at 532nm. This range of LR is comparable to previous studies and basically covers the range of

typical dust LR. The $\pm 9\ sr$ LR uncertainties induces around $\pm\ 20\%$ uncertainties in DAOD (the shaded area in Figure 9 in the revised manuscript).

For DAOD uncertainty from depolarization ratio (DPR), we added Figure S6 in the supplementary material to show the global distribution of DPR induced DAOD uncertainties. We found that the uncertainty is much lower in dust dominant regions than other regions. The averaged uncertainty for regions with DAOD>0.05 is 20%, while the averaged uncertainty for other regions is 38%.

It is difficult to quantify DAOD uncertainty induced by each source in one study. Instead, we estimate absolute expected error of MODIS- and CALIOP-based DAOD by comparing with AERONET in-situ measurements of monthly mean coarse mode optical depth (COD). In this process, we consider AERONET COD as a proxy of DAOD since so far there is not a valid method to derive DAOD from AERONET AOD.

For over-land dust retrievals between 2007 and 2019, there were 16653 MODIS, CALIOP monthly mean DAOD retrievals collocated with observations from 761 AERONET sites located within a 1-degree MODIS and CALIOP grid cell (Figure 1 in the revised manuscript). Overall, MODIS DAOD is biased high compared with AERONET COD, with absolute bias $of\ B_a = 0.01$ and relative bias $of\ B_r = 26.7\%$. On the other hand, CALIOP DAOD is generally biased low with $B_a = -0.02$ and $B_r = -27.9\%$. For over-ocean dust retrievals, between 2007 and 2019, there were 7755 MODIS, CALIOP monthly mean DAOD retrievals collocated with 311 AERONET sites located within a 1-degree MODIS and CALIOP grid cell (Figure 3 in the revised manuscript). MODIS DAOD is biased high compared with AERONET COD with absolute bias $B_a = 0.01$ and relative bias $B_r = 18.1\%$. On contrary, CALIOP is generally biased low with $B_a = -0.02$ and $B_r = -35\%$. We estimate that the absolute expected error of MODIS DAOD is 0.65×DAOD$_M$+0 over land and 0.50×DAOD$_M$+0 over ocean, the absolute expected error of CALIOP DAOD is 0.52×DAOD$_C$+0.02 over land and 0.54×DAOD$_C$+0.02 over ocean. This analysis has been added in section 3.2.

Q4. Line 352-4: Here and in the rest of this section, the suggestion is made that CALIPSO's lower sampling frequency could be a cause for the systematically lower DAOD from CALIPSO than from MODIS. I think this is incorrect. A lower sampling frequency is just as likely to produce an overestimate of DAOD as an underestimate, so producing a systematic underestimation of a factor of two for many different regions is in my view implausible. Please correct this.

- Line 370-1: I think this statement is similarly statistically incorrect. Why would episodic sampling of dust events produce a systematically lower DAOD, especially when you have 12 years of data and so many total retrievals?

**Reply:** This is a good point. Statistically, CALIOP's lower frequency indeed is not a reasonable explanation for its systematically lower DAOD than MODIS. We corrected our statement in this part as below:

The CALIOP-based DAOD is rather low in regions that are known to be dusty in certain seasons, such as southwestern United States, South America, Australia and South Africa. These regions do stand out in MODIS DAOD maps (the second column in Figure 5). Then we plot DAOD-to-TAOD ratio based on DAOD and TAOD retrievals from two sensors (the last two columns in Figure 5). These regions indeed show up in the DAOD-to-TAOD ratio plot based on both sensors (i.e., the last two columns in Figure 5). This implies that in those regions both sensorspecific methodologies are able to distinguish dust aerosol from sensor-detected total aerosol to some extent so that the DAOD to TAOD ratio stands out in those regions for both sensors.

In addition, the conditionally sampled DAOD product is kept in the revised manuscript as an independent part instead of using this as an explanation of low CALIOP-DAOD, because the comparison between climatological and conditional DAOD could provide important information about frequency and intensity of dust events.

**Other comments:**

Q1. The abstract notes a climatological record over the last two decades, but the period spanned is really 2007-2019. Please correct.

**Reply:** Thanks. We corrected it as 'from 2007 to 2019'.

Q2. Lines 242-4: Yu et al. (2021) recently did a detailed analysis of the diurnal variability of dust AOD that should be mentioned here. And can you roughly estimate the error you expect from the difference in daytime and nighttime distributions based on Yu et al. and Kittaka '11? I think this should be small (~10%?) relative to your other errors.

**Reply:** Thanks for the suggestion and the reference. We added a paragraph to Section 2.1 to discuss why we use nighttime observation for CALIOP DAOD retrievals. We cited Yu et al. 2021 in this section. We would like to note here that, Yu et al. (2021) actually found a significant day-night inconsistency in their CATS AOD retrievals after comparisons with collocated AERONET retrievals (see their discussion in section 3.1). Because of this, they concluded that "*To account for this day–night inconsistency in CATS data quality, the diurnal variability in dust and dust mixture characteristics is currently examined separately for daytime and nighttime periods*". In other words, their daytime and nighttime DAOD, even though plotted together, are NOT directly comparable. But you are right that, the contrast between daytime and nighttime DAOD based on these plots is roughly 10-15%, which is smaller than other uncertainties in CALIOP retrievals as analyzed in section 3. Again, it has to be emphasized that this contrast is partly due to the day–night inconsistency in CATS data quality.

Q3. Line 247-9: This statement seems a bit vague. Could you be specific as to how large you expect the solar noise to be? If that's larger than 10% then the statement would sound less subjective to the reader.

**Reply:** It is hard to isolate the uncertainty induced by solar noise from all other uncertainty sources. However, we think our choice of using nighttime CALIOP observations is still valid.

First, based on our discussion in the above question Q2, we know that it is very difficult to extract true daytime and nighttime DAOD difference from lidar observations. The diurnal variation of DAOD in Yu et al. 2021 are actually investigated for daytime and nighttime separately due to the significant difference in data quality.

Second, we also retrieve DAOD based on CALIOP daytime observations and further analyze the difference between CALIOP daytime and nighttime DAOD datasets (Figure S1). We found that CALIOP daytime DAOD is generally much greater than nighttime DAOD in open ocean regions where dust aerosol is not expected to appear (see the third column in Figure S1). This means CALIOP daytime DAOD has a much lower quality than nighttime DAOD, which is mainly due to solar contamination in daytime CALIOP observations. Considering the low data quality of CALIOP daytime DAOD dataset, we choose to use the nighttime CALIOP product that is free of solar noise.

Third, using CALIOP daytime DAOD dataset would not change our result in this study, specifically CALIOP DAOD would be still systematically smaller than MODIS DAOD. CALIOP daytime DAOD is generally smaller than nighttime in dust-laden regions. Generally, CALIOP

nighttime DAOD is smaller than MODIS DAOD especially in some dust-laden regions. If we change to use CALIOP daytime DAOD, then its difference with MODIS DAOD would be even larger.

Considering all aforementioned issues, we decide to use higher quality nighttime CALIOP DAOD dataset in the analysis.

**Q4. Line 270-2: This sentence is unclear. Could you include a supplementary graph on how sensitive your results are to this criterion of 10 retrievals per month?**

**Reply:** Figure 1 below shows the impact of requiring a minimum of 10 DAOD retrievals in a month on the derived seasonal mean DAOD over ocean. The first column shows seasonal mean DAOD (2007~2019) directly from MODIS retrieval. The second column shows the seasonal mean DAOD of MODIS retrievals with a minimum of 10 DAOD retrievals in a month. The third column is the difference between the first two columns. Over dust dominant regions such as the North Atlantic Ocean, there is no impact. However, over some heavy cloudy regions such as the North Pacific Ocean, DAOD in column 1 is significantly higher than column 2 during MAM and JJA. In those regions, MODIS dust retrieval could bias high due to cloud contamination, screening out infrequent sampling of DAOD could minimize the cloud contamination in those regions to some extent.

[Figure]

Figure 1 The first column shows seasonal mean DAOD (2007~2019) directly from MODIS retrieval. The second column shows the seasonal mean DAOD of MODIS retrievals with a minimum of 10 DAOD retrievals in a month. The third column is the difference between the first two columns.

**Q5. Section 3.1 is very long and a bit difficult to read. I recommend adding sub-sections to improve readability.**

**Reply:** Thanks for the suggestion. In the revised manuscript, we break the original Section into two sections as section 4.1 and 4.2.

**Q6. Line 317-8: could you explain why the coarse size of dust contributes to the positive depolarization ratio?**

**Reply:** We added two references at this point to explain that the linear depolarization ratio of dust particles depends on the dust particle size, one is based on theoretical studies (Gasteiger et al., 2011), the other one is based on an experimental basis (Järvinen et al., 2016).

Järvinen et al. 2016 shows that the strongest size-dependence was observed for fine-mode particles as their depolarization ratios increased almost linearly with particle median diameter from 0.03 to 0.3, whereas the coarse-mode particle depolarization values stayed rather constant with a mean linear depolarization ratio of 0.27.

**Q7. Line 319-20: If DAOD over ocean is derived from TAOD and FMF, then how do you remove the AOD due to sea salt? See my comment above regarding a need for more detail on how the DAOD/AOD ratio is calculated for MODIS.**

**Reply:** We added more details regarding MODIS DAOD retrievals over ocean and land in Section 2.2. The more detailed description could answer how to distinguish DAOD from others for over-ocean and overland retrievals (see answer for Q2 in main comments).

**Q8. Is the global DAOD you report weighted by the area of each grid point? It'd be helpful to note this somewhere.**

**Reply:** This is a great point. The description of the way we calculated global mean DAOD is added in caption of Table 4.

'Since Earth is a sphere, grid-cell surface area decreases toward the poles. We weight each grid-cell surface area into ocean, land and global DAOD average'. This description was added in the caption of Table 4.

**Q9. Line 384-6: A recent paper by Huang et al. (2020) showed that Asian dust indeed gets more spherical during transport so might be good to include here.**

**Reply:** Thanks for the suggestion. The reference paper is included.

**Q10. Line 438-9: Please elaborate on how specifically the seasonal cycles of dust emission and transport are consistent with previous results.**

**Reply:** More discussion about comparison with previous studies are added in the revised manuscript.

For example, Prospero et al. 2002 shows that dust activity peaks in May-July in North Africa and Middle East, while peaks in spring in China. These seasonal cycles are consistent with our results shown in the first row of Figure 6. Yu et al. 2015a shows that DAOD peaks in June-July-August in La Parguera, which is consistent with the seasonal cycle in CRB in this study.

**Q11. Line 480-4: Here and elsewhere, the authors include some insightful explanations on how errors in cloud screening affect both CALIOP and MODIS retrievals. Are you able to quantify the difference in the AOD between CALIOP and MODIS that is due to these errors from simultaneous retrievals from both?**

**Reply:** Thanks for the suggestion, but this is extremely difficult and beyond the scope of this study. Cloud screening process is designed and implemented by the operational algorithm teams, which is difficult for us to analyze. Moreover, our DAOD retrievals are at 2x5 degree resolution, so it is not possible to obtain "simultaneous retrievals" based on the current data. The current paper is already too long, we will leave this investigation to future study.

**Q12. Line 592: should be "desert" not "dessert" :)**
**Reply:** Thanks. The typo is corrected. :)

**Q13. Line 610: what do you mean by appropriate, exactly?**
**Reply:** Thanks for the question. This 'appropriate' could be explained by our description of MODIS DAOD retrieval in Section 2.2. For MODIS over-ocean DAOD retrieval, we have two equations Eq. (2) and (3) shown in the answer for Q2 in main comments. Based on Eq. (2) and (3), $\tau_d$ can be calculated from MODIS-retrieved $\tau$ and $f$, with appropriate parameterizations for $f_m, f_d, f_c$ and $\tau_m$. More specifically, the appropriate parameterizations here represent that $f_m, f_d, f_c$ were determined from retrieved $f$ in selected regions and seasons for which a specific aerosol type dominates, $\tau_m$ was parameterized as a function of wind speed (details can be found in Kaufman et al., 2005; Yu et al., 2009, 2020).
In addition, a reference was added in the revised manuscript for the parameterization values.

**Q14. Line 614: here and elsewhere, could you define what you mean by "coarse mode", exactly? What particle size range does that refer to? Definitions differ for that differ in the literature.**
**Reply:** This is a great question.
For AERONET product, the fine and coarse mode optical depths are defined optically (rather than in terms of a microphysical cutoff of the associated particle size distribution at some specific radius) and essentially depend on the fact that the coarse mode spectral variation is approximately neutral (O'Neill et al., 2003). In the paper, we mentioned that MODIS overland DAOD retrieval represents the coarse-mode fraction of dust only, the coarse mode here refers to the portion of the actual aerosol that exists at low-RH aerodynamic diameters larger than 1 µm (Anderson et al., 2005). The definition of coarse mode for AERONET and MODIS retrievals are added in the revised manuscript.

**Q15. Line 615: although fine dust accounts for a small fraction of the mass, it accounts for a larger fraction of the DAOD, so this statement should be corrected.**
**Reply:** This is a great point.
Based on the coarse-mode definition in MODIS overland retrieval (mentioned in Q14), basically MODIS overland retrieval exclude submicron dust aerosol. Then we double checked Kok et al. 2017 and corrected our statement.
This part is changed to 'The exclusion of submicron dust aerosol could induce around 3% underestimation of the global dust mass load and around 15% underestimation of the global DAOD (see Figure S1 in Kok et al. 2017).'

**Q16. Line 616-8: this is helpful information. Could you be clear here what the bias is of the Aqua MODIS DAOD relative to AERONET, and what the error is? And since coarse mode includes sea salt, is this a one-to-one comparison?**
**Reply:** Yes, the comparison with AERONET measurements is helpful information. Therefore, we added our own analysis regarding comparison between our monthly mean DAOD datasets with AERONET monthly mean coarse-mode AOD (COD) and further derived absolute expected error. we estimated that the absolute expected error of MODIS DAOD is $0.65 \times DAOD_M + 0$ over land and $0.50 \times DAOD_M + 0$ over ocean, the absolute expected error of CALIOP DAOD is $0.52 \times DAOD_C + 0.02$ over land and $0.54 \times DAOD_C + 0.02$ over ocean. This analysis was added in section 3.2.

Actually, comparison with AERONET COD is not one-to-one comparison because so far there is not a valid method to derive DAOD from AERONET measurements. However, over land, especially dust source regions, dust aerosols are predominantly in coarse mode, therefore, AERONET COD could be considered as a good proxy of DAOD over land. Over ocean, the exclusion of fine mode DAOD could be partially cancelled by the inclusion of coarse sea salt AOD in AERONET COD retrievals. Therefore, AERONET COD is considered as a proxy of DAOD over ocean as well.

Q17. Line 649: could you be quantitative here about the correlation? I think whether MODIS and CALIOP-based DAOD "correlate well" is subjective and I personally was surprised the correlation was not higher.

**Reply:** The coefficients of correlation (R values) are added in this part for each region to make this statement quantitative.

This part is changed to 'Through the comparison, we found generally CALIOP-based DAOD correlates well with MODIS-based DAOD over dust-laden regions such as Sahara (R=0.78), TAT (R=0.84), CRB (R=0.75) and ARB (R=0.85)'

Q18. Line 653: I think here k is the correlation and R^2 is the variance explained.

**Reply:** We change to use R instead of $R^2$ in the revised version. Correlation coefficient (R) is a parameter to indicate the degree of correlation between two variables. R ranges from $-1$ to $+1$. R= $-1$ means that there is a perfect negative relationship between the two variables. For every positive increase in one variable, there is a decrease of a fixed proportion in the other. R= $+1$ means that there is a perfect positive relationship between the two variables. For every positive increase in one variable, there is an increase of a fixed proportion in the other. $R^2$ is the coefficient of determination, which shows percentage variation in y-axis variable can be explained by the dependence of x-axis variable using the particular regression model.

We think the slope 'k' indicates how does one variable change with the other. For example, we have two variables $x = [2,4,6,8,10]$ and $y = [1,2,3,4,5]$. Then we have $y=0.5x$. In this case, even though k = 0.5, but the two variables correlate very well with correlation coefficient R=1.

References:

Anderson, T. L., Wu, Y., Chu, D. A., Schmid, B., Redemann, J., & Dubovik, O. (2005). Testing the MODIS satellite retrieval of aerosol fine-mode fraction. *Journal of Geophysical Research D: Atmospheres*, *110*(18), 1–16. https://doi.org/10.1029/2005JD005978

Eck, T. F., Holben, B. N., Reid, J. S., Dubovik, O., Smirnov, A., O'Neill, N. T., … Kinne, S. (1999). Wavelength dependence of the optical depth of biomass burning, urban, and desert dust aerosols. *Journal of Geophysical Research Atmospheres*, *104*(D24), 31333–31349. https://doi.org/10.1029/1999JD900923

Gasteiger, J., Wiegner, M., Groß, S., Freudenthaler, V., Toledano, C., Tesche, M., & Kandler, K. (2011). Modelling lidar-relevant optical properties of complex mineral dust aerosols. *Tellus, Series B: Chemical and Physical Meteorology*, *63*(4), 725–741. https://doi.org/10.1111/j.1600-0889.2011.00559.x

Gelaro, R., McCarty, W., Suárez, M. J., Todling, R., Molod, A., Takacs, L., … Zhao, B. (2017). The modern-era retrospective analysis for research and applications, version 2 (MERRA-2). *Journal of Climate*, *30*(14), 5419–5454. https://doi.org/10.1175/JCLI-D-16-0758.1

Gkikas, A., Proestakis, E., Amiridis, V., Kazadzis, S., Di Tomaso, E., Tsekeri, A., … Pérez Garciá-Pando, C. (2021). ModIs Dust AeroSol (MIDAS): A global fine-resolution dust optical depth data set. *Atmospheric Measurement Techniques*, *14*(1), 309–334. https://doi.org/10.5194/amt-14-309-2021

Hsu, N. C., M. J. Jeong, C. Bettenhausen, A. M. Sayer, R. Hansell, C. S. Seftor, J. Huang, and S. C. Tsay. 2013. "Enhanced Deep Blue Aerosol Retrieval Algorithm: The Second Generation." *Journal of Geophysical Research: Atmospheres* 118(16):9296–9315

Järvinen, E., Kemppinen, O., Nousiainen, T., Kociok, T., Möhler, O., Leisner, T., & Schnaiter, M. (2016). Laboratory investigations of mineral dust near-backscattering depolarization ratios. *Journal of Quantitative Spectroscopy and Radiative Transfer*, *178*, 192–208. https://doi.org/10.1016/j.jqsrt.2016.02.003

Kaufman, Y. J., Koren, I., Remer, L. A., Tanré, D., Ginoux, P., & Fan, S. (2005). Dust transport and deposition observed from the Terra-Moderate Resolution Imaging Spectroradiometer (MODIS) spacecraft over the Atlantic Ocean. *Journal of Geophysical Research D: Atmospheres*, *110*(10), 1–16. https://doi.org/10.1029/2003JD004436

Kok, J. F., Ridley, D. A., Zhou, Q., Miller, R. L., Zhao, C., Heald, C. L., … Haustein, K. (2017). Smaller desert dust cooling effect estimated from analysis of dust size and abundance. *Nature Geoscience*, *10*(4), 274–278. https://doi.org/10.1038/ngeo2912

Kurosaki, Y., & Mikami, M. (2003). Recent frequent dust events and their relation to surface wind in East Asia. *Geophysical Research Letters*, *30*(14). https://doi.org/10.1029/2003GL017261

Lee, E. H., & Sohn, B. J. (2011). Recent increasing trend in dust frequency over Mongolia and Inner Mongolia regions and its association with climate and surface condition change. *Atmospheric Environment*, *45*(27), 4611–4616. https://doi.org/10.1016/j.atmosenv.2011.05.065

Mielonen, T., Arola, A., Komppula, M., Kukkonen, J., Koskinen, J., De Leeuw, G., & Lehtinen, K. E. J. (2009). Comparison of CALIOP level 2 aerosol subtypes to aerosol types derived from AERONET inversion data. *Geophysical Research Letters*, *36*(18). https://doi.org/10.1029/2009GL039609

O'Neill, N. T., Eck, T. F., Smirnov, A., Holben, B. N., & Thulasiraman, S. (2003). Spectral discrimination of coarse and fine mode optical depth. *Journal of Geophysical Research: Atmospheres*, *108*(17). https://doi.org/10.1029/2002jd002975

Proestakis, E., Amiridis, V., Marinou, E., Georgoulias, A. K., Solomos, S., Kazadzis, S., … Van Der A, R. J. (2018). Nine-year spatial and temporal evolution of desert dust aerosols over South and East Asia as revealed by CALIOP. *Atmospheric Chemistry and Physics*, *18*(2), 1337–1362. https://doi.org/10.5194/acp-18-1337-2018

Pu, B., & Ginoux, P. (2018). How reliable are CMIP5 models in simulating dust optical depth? *Atmospheric Chemistry and Physics*, *18*(16), 12491–12510. https://doi.org/10.5194/acp-18-12491-2018

Qian, W., Quan, L., & Shi, S. (2002). Variations of the dust storm in China and its climatic control. *Journal of Climate*, *15*(10), 1216–1229. https://doi.org/10.1175/1520-0442(2002)015<1216:VOTDSI>2.0.CO;2

Ridley, A. D., Heald, L. C., Kok, F. J., & Zhao, C. (2016). An observationally constrained estimate of global dust aerosol optical depth. *Atmospheric Chemistry and Physics*, *16*(23), 15097–15117. https://doi.org/10.5194/acp-16-15097-2016

Sayer, A. M., Hsu, N. C., Bettenhausen, C., & Jeong, M. J. (2013). Validation and uncertainty estimates for MODIS Collection 6 "deep Blue" aerosol data. *Journal of Geophysical Research Atmospheres*, *118*(14), 7864–7872. https://doi.org/10.1002/jgrd.50600

Yu, H. B., Chin, M., Bian, H. S., Yuan, T. L., Prospero, J. M., Omar, A. H., … Zhang, Z. B. (2015). Quantification of trans-Atlantic dust transport from seven-year (2007-2013) record of CALIPSO lidar measurements. *Remote Sensing of Environment*, *159*, 232–249. https://doi.org/10.1016/j.rse.2014.12.010

Yu, H., Chin, M., Remer, L. A., Kleidman, R. G., Bellouin, N., Bian, H., & Diehl, T. (2009). Variability of marine aerosol fine-mode fraction and estimates of anthropogenic aerosol component over cloud-free oceans from the Moderate Resolution Imaging Spectroradiometer (MODIS). *Journal of Geophysical Research Atmospheres*, *114*(10). https://doi.org/10.1029/2008JD010648

Yu, H., Yang, Y., Wang, H., Tan, Q., Chin, M., Levy, R., … Shi, Y. (2020). Interannual Variability and Trends of Combustion Aerosol and Dust in Major Continental Outflows Revealed by MODIS Retrievals and CAM5 Simulations During 2003–2017. *Atmospheric Chemistry and Physics Discussions*, 1–38. https://doi.org/10.5194/acp-2019-621

---

## Author Comment (AC2)

**Responds to the RC2**

Dust is an important aerosol type in the atmosphere and has significant impacts on environment and climate. Satellite is a very useful tool to detect dust aerosol as it can provide dust distribution of large spatial coverage with long-term duration. A unique lidar platform (CALIOP) can further provide the vertical distribution of dust aerosol. This study provides a 13-year global dust optical depth (DAOD) climatology derived from CALIOP and MODIS observations and presents some interesting results for global dust spatial distribution and temporal variations. The authors also investigate the reasons for the recent decline of dust activities in East Asia. The results demonstrate the reliability of the two datasets the authors have developed.

However, the authors seem to mention a lot of uncertainties in the dataset and it seems there is no answer for a better DAOD. I personally would like to see more convincing quantitative comparison of these two datasets, such as which one is better and it is possible to suggest a more reliable global DAOD. I suggest the authors make more clear conclusions relative to the comparison.

**Reply:** We thank the reviewer for constructive and insightful comments. We have addressed these comments in the revision. An item-to-item reply to the reviewer's comments is provided below.

Before addressing specific comments/questions, we would like to first provide a summary of the major revisions made to the manuscript:

- We added more detailed discussion regarding MODIS DAOD retrieval methodologies over ocean and land in section 2.2.
- We compared our MODIS- and CALIOP-based DAOD with values reported in previous studies based on MODIS and CALIOP, respectively. The comparison is added to the revised manuscript as section 3.1. For MODIS DAOD comparison, we compare our results with previous studies in both global and regional scales; For CALIOP, there isn't global CALIOP-based DAOD retrievals to compare our result with, therefore, the comparison is limited to regional scale. Overall, these comparisons suggest that our results are in reasonable agreement with previous studies, except for Voss and Evan 2020 over ocean (which can be explained by the use of different parameterization schemes).
- We evaluated our monthly mean MODIS- and CALIOP-based DAOD product by comparing with AERONET monthly mean coarse mode AOD (COD) from 2007 to 2019. We found that MODIS DAOD is statistically higher than AERONET COD by 26.7% over land and 18.5% over ocean, while CALIOP DAOD is lower than AERONET COD by 27.9% over land and 35% over ocean. This may suggest that the true DAOD probably fall between MODIS and CALIOP DAOD retrievals. Furthermore, by following the methodology proposed by Sayer et al. 2013, we estimated that the absolute expected error of MODIS DAOD is $0.65 \times DAOD_M + 0$ over land and $0.50 \times DAOD_M + 0$ over ocean, the absolute expected error of CALIOP DAOD is $0.52 \times DAOD_C + 0.02$ over land and $0.54 \times DAOD_C + 0.02$ over ocean. This analysis was added in section 3.2.

After these revisions, we think the paper is much improved and more focused, although the general conclusions still hold.

**Major Comments:**

Q1. Section 2 and 3: I am curious about the systematic difference between CALIOP and MODIS. In my understanding, the main reasons may be in the algorithms used to generate the product and the difference may be minimized if calibrated to the same data sources. I am wondering if there is some way to minimize the difference between different dataset. The authors seem to mention a lot of uncertainty and it may be helpful to separate the contributions from these factors. First, is it possible to compare the two dataset at the same time and close location (although the sample number may be small)? Second, after doing this comparison, the difference of monthly/seasonal mean can be due to the sampling and aggregate methods. I think the authors can provide a map of observation numbers for different months/seasons and a month with more temporal coverage is more likely to have a more reliable statistical result.

**Reply:** Thanks for the question.

First, this work is to present a climatological DAOD dataset derived from CALIOP observations and its comparison with MODIS-based DAOD dataset. Due to the spatial and temporal limit of CALIOP sampling, the DAOD is initially derived as monthly mean level 3 product. Therefore, the best comparison we could do in this study is in monthly mean level.

Second, in our CALIOP DAOD retrieval, we calculated monthly mean dust extinction coefficient at each altitude for each grid and then derived column integrated monthly mean DAOD. Refer to your suggestion, we added sample numbers at each altitude for each grid in our CALIOP monthly mean DAOD dataset.

Q2. Section 2.2: the comparison of nighttime CALIOP and daytime MODIS product is made based on the consideration "Kittaka et al., 2011 shows that daytime and nighttime global seasonal-mean AOD distributions for JJA 2006 from CALIOP are generally similar in both outflow and source regions (see their Figure 1)." (Lines 242- 244). First, Kittaka's conclusions are based on the global distribution, while this present study is specifically on dust source regions and dust outflow regions. Kittaka's conclusions should not be simply applied to present study. Second, previous studies have shown significant diurnal variations of dust event frequency in dust sources regions (e. g, Figure 17 of Yue, X., H. Wang, Z. Wang, and K. Fan (2009), Simulation of dust aerosol radiative feedback using the Global Transport Model of Dust: 1. Dust cycle and validation, J. Geophys. Res., 114, D10202, doi:10.1029/2008JD010995.).

**Reply:** Thanks for the insight.

It is correct that Kittaka's result may not be simply applied to this study, which mainly focuses on dust source and outflow regions.

To justify our comparison between CALIOP nighttime DAOD with MODIS daytime DAOD.

First, we also retrieve DAOD based on CALIOP daytime observations and further analyze the difference between CALIOP daytime and nighttime DAOD datasets (Figure S1). We found that CALIOP daytime DAOD is generally much greater than nighttime DAOD in open ocean regions where dust aerosol is not expected to appear (see the third column in Figure S1). This means CALIOP daytime DAOD has a much lower quality than nighttime DAOD, which is mainly due to solar contamination in daytime CALIOP observations. Considering the low data quality of CALIOP daytime DAOD dataset, we choose to use the nighttime CALIOP product that is free of solar noise.

Second, Xu et al. 2009 (denoted as Xu2009) indeed shows that dust mobilization is more active during the local daytime than nighttime. However, more frequent dust uplift does not mean higher DAOD in the atmosphere. DAOD also depends on dust dry/wet deposition. Figure 16 (a) in

Xu2009 shows diurnal variation of global mean dust uplift, dry/wet deposition and dust burden. They conclude that dust burden shows the least variation with a standard deviation of only 1% of its mean value. Although they did not discuss DAOD diurnal variation in the paper, we assume that DAOD has similar magnitude of variation.

Third, using the CALIOP daytime DAOD dataset would not change the conclusion of this study: that is CALIOP DAOD is systematically lower than MODIS DAOD. CALIOP daytime DAOD is generally smaller than nighttime in dust-laden regions. Generally, CALIOP nighttime DAOD is smaller than MODIS DAOD especially in some dust-laden regions. If we change to use CALIOP daytime DAOD, then its difference with MODIS DAOD would be even larger.

Considering all aforementioned issues, we decide to use high-quality nighttime CALIOP DAOD dataset in our analysis.

Line 647, in Summary and Conclusions: it seems the trends and interannual variability of DAOD are similar. I don't see the advantages and limitations clearly for each dataset. Please clarify.

**Reply:** Thanks for the question.

CALIOP and MODIS DAOD are systematically different. In most dust laden regions, CALIOP DAOD is statistically smaller than MODIS (Figure 5 in the paper). Even though the DAOD magnitude from two retrievals are very different as discussed in the previous sections, both CALIOP and MODIS show similar DAOD interannual variability and trends. Therefore, the interannual trend of DAOD is trustworthy since both retrievals show very similar trends.

**Specific comments:**

Q1. Line 21, abstract: add "(2007-2019)" after "the last two decades".
**Reply:** Done

Q2. Line 27, abstract: delete "and".
**Reply:** Done

Q3. Line 127: Self-consistent: Please briefly explain the word here, and not wait until Section 2.
**Reply:** The following sentence in the manuscript is a brief explanation of self-consistent by citing Yu et al. 2009 as an example.
For example, in MOIDS over-ocean DAOD retrieval, both AOD ($\tau$) and fine-mode AOD ($f\tau$) are assumed to be composed of marine aerosol, dust and combustion aerosols, i.e.,

$$\tau = \tau_m + \tau_d + \tau_c \ , \qquad (2)$$

$$f\tau = f_m\tau_m + f_d\tau_d + f_c\tau_c \ , \qquad (3)$$

Where the subscripts m, d, and c represent marine aerosol, dust and combustion aerosol, respectively. Based on Eq. (2) and (3), $\tau_d$ can be calculated from MODIS-retrieved $\tau$ and $f$, with appropriate parameterizations for $f_m, f_d, f_c$ and $\tau_m$. In this study, $f_m, f_d, f_c$ were determined from MODIS retrieved $f$ in selected regions and seasons for which a specific aerosol type dominates, $\tau_m$ was parameterized as a function of wind speed (details can be found in Kaufman et al., 2005; Yu et al., 2009, 2020). In this case, the self-consistent means that the selection of $f_m, f_d, f_c$ are also from MODIS retrieval rather than from other sources such as AERONET. The self-consistent use of MODIS data could minimize the introduction of additional biases due to discrepancies in FMF between MODIS and AERONET.

Q4. Lines 131-132: is it any critical difference for dust between these version and previous version? This is important, as it may indicate the results shown in this study may be different from previous versions because of version updates.
**Reply:** There are numerous updates from V3 to V4 for CALIOP retrievals and similarly from V5 to V6 for MODIS retrievals, including instrument calibration updates, algorithm adjustments and modifications of QA flags. It is impossible and beyond the scope of this study to keep track of all these changes and investigate the impacts on DAOD retrievals. In this study, we use the latest versions of CALIOP and MODIS retrievals and report the corresponding DAOD results. Interested readers can compare our results and previous studies based on earlier versions of retrievals. The differences can be investigated, if significant, in future studies.

Q5. Line 155: please provide some supporting information for "70% agreement".
**Reply:** The '70% agreement' is also supported by the reference paper in the following sentence. Therefore, we combine the two sentences in the revised version as 'The comparison between aerosol subtypes in CALIOP level 2 V2.01 and NASA Aerosol Robotic Network (AERONET) aerosol types shows that 70% of the CALIOP and AERONET aerosol types are in agreement and best agreement is achieved for dust and polluted dust (Mielonen et al., 2009).'

Q6. Lines 162-163: if 40 sr is too low, is it possible to increase this value and update the product? What is the lidar ratio?

**Reply:** In CALIOP Version 4 product, the greater lidar ratio of 44sr is used for dust aerosols. We cannot update the product with a use of the new lidar ratio because the CALIOP retrieval uses the lidar ratio to first correct light attenuation and then convert the attenuation-corrected backscatter to the extinction. Extinction does not have a linear relationship with lidar ratio. Lidar ratio is defined as Extinction-to-backscatter ratio. This explanation of lidar ratio is added in the revised manuscript.

Q7. Line 174: I don't understand "or" here. Please clarify.

**Reply:** Sorry for the confusion. It means that some of these studies consider 'dust' subtype as dust; others consider both 'dust' and 'polluted dust' subtypes as dust.

Q8. Sections 2.1 and 2.2: is it possible to make a table and put the comparison of key features of CALIOP and MODIS in it?

**Reply:** Section 2.1 and 2.2 describe CALIOP and MODIS dust retrieval methods, respectively. We added more detailed information for MODIS dust retrieval over ocean and land. Table 1 in the manuscript is the summary of key features of the two DAOD retrievals (see below).

Table 1. Summary of DAOD retrievals from MODIS and CALIOP

| Sensors | Retrieve Scope | Relevant variables used to derive DAOD | References |
|---------|----------------|----------------------------------------|------------|
| MODIS | Ocean | AOD, fine-mode AOD | Yu et al. (2009, 2020) |
| MODIS | Land | AOD, SSA at 470nm, Angstrom exponent | Pu and Ginoux et al. (2018) |
| CALIOP | Globe | Profiles of backscatter, extinction, depolarization ratio | Yu et al. (2015a) |

Q9. Line 242: Kittaka et al. (2011) also analyze the AOD including all aerosol types. I am wondering whether their conclusions applied to specific dust source regions and dust- effect regions.

**Reply:** Exactly, Kittaka et al. 2011 analyze the AOD including all aerosol types. In the revised manuscript, we also retrieve DAOD based on CALIOP daytime observations and further analyze the difference between CALIOP daytime and nighttime DAOD datasets (Figure S1). We found that CALIOP daytime DAOD is generally much greater than nighttime DAOD in open ocean regions where dust aerosol is not expected to appear (see the third column in Figure S1). This means CALIOP daytime DAOD has a lower quality than nighttime DAOD, which is mainly due to solar contamination in daytime CALIOP observations. Considering the low data quality of CALIOP daytime DAOD dataset, we choose to use the nighttime CALIOP product that is free of solar noise. The bottom line is that CALIOP daytime and nighttime difference may not be physical.

Q10. Line 248: "in hoping that": this statement may be misleading. I think the key point is that observations do shows some significant diurnal of dust events in the source regions mainly because of the wind speed difference. Please clarify.

**Reply:** Thanks for the insight.

As discussed in our reply to the Q2 in the major comments. Xu et al. 2009 (denoted as Xu2009) shows that dust mobilization is more active during the local daytime than nighttime. However, more frequent dust uplift does not mean higher DAOD in the atmosphere. DAOD also depends on dust dry/wet deposition. Figure 16 (a) in Xu2009 shows diurnal variation of global mean dust uplift, dry/wet deposition and dust burden. They conclude that dust burden shows the least variation with a standard deviation of only 1% of its mean value. We believe that DAOD would have a variation similar to dust burden.

For some DAOD diurnal variation studies based on CATS observations, there is a significant difference between daytime and nighttime CATS AOD quality, because CATS daytime observation has a higher lidar calibration uncertainty at 1064nm and is subject to solar contamination. To account for this day-night inconsistency in CATS AOD quality, the diurnal variability is examined separately for daytime and nighttime periods. Therefore, the true daytime-nighttime DAOD difference is still not fully understood.

To justify our comparison between CALIOP nighttime DAOD with MODIS daytime DAOD. We also retrieve DAOD based on CALIOP daytime observations and further analyze the difference between CALIOP daytime and nighttime DAOD datasets (Figure S1). We found that CALIOP daytime DAOD is generally much greater than nighttime DAOD in open ocean regions where dust aerosol is not expected to appear (see the third column in Figure S1). This means CALIOP daytime DAOD has a much lower quality than nighttime DAOD, which is mainly due to solar contamination in daytime CALIOP observations. Considering the low data quality of CALIOP daytime DAOD dataset, we choose to use the nighttime CALIOP product that is free of solar noise.

Q11. Line 268: I am wondering whether this new data includes all the algorithms mentioned in previous paragraph.
**Reply:** Yes, the three retrieval methods (global CALIOP, MODIS over ocean and MODIS over Land) introduced in section 2 are all used in this study.

Q12. Line 272: how about the sampling over land? Is there a minimum number required for deriving monthly statistics?
**Reply:** Thanks for the question. For MODIS overland retrieval, there is no minimum of the number of samples to derive monthly values.

Q13. Lines 279-280: is there any measure to keep the ocean and land product consistent with each other?
**Reply:** For MODIS, the observations are consistent over ocean and over land. However, different aerosol retrieval algorithms are used for over ocean (Dark Target) and over land (Deep Blue) due to their different surface characteristics. Therefore, MODIS-based DAOD retrieval methods, which make use of parameters retrieved from MODIS aerosol retrievals, are also different for over ocean and over land (see details in section 2).
For CALIOP, we use a consistent algorithm to retrieve DAOD over land and ocean.

Q14. Section 3.1: this section is too long. Consider adding subtitle or dividing it into two sections.
**Reply:** Thanks for the suggestion. In the revised manuscript, we break the original Section into two sections as section 4.1 and 4.2.

**Reply:** Thanks for the comment. We corrected our statement here. We deleted the statement of more frequent miss of dust by CALIOP.

This part is to discuss DAOD in regions that are known to be dusty in certain seasons, such as southwestern United States, South America (Patagonian Desert), Australia, and South Africa. In those regions, CALIOP-based climatological DAOD is rather low.

The low DAOD based on CALIOP in these regions could be the results of two factors. One is that the low aerosol detected by CALIOP (this could be indicated by TAOD comparison between CALIOP and MODIS in Figure 7), the other one is the different algorithm used for CALIOP and MODIS to distinguish dust aerosol from other types of aerosol. We found that these regions indeed show up in the DAOD-to-TAOD ratio plot based on both sensors (i.e., the last two columns in Figure 5). This means that in those regions both sensor-specific methodologies are able to distinguish dust aerosol from sensor-detected total aerosol to some extent so that the DAOD to TAOD ratio stands out in those regions for both sensors.

**Reply:** Thanks for the insight.

In this study, climatological DAOD includes all cloud-free cases in the average of dust extinction and DAOD regardless of the presence of dust, while conditional DAOD is calculated by only averaging those cases where dust is detected (i.e., DAOD and dust extinction are non-zero).

The comparison of climatological and conditional DAOD could provide useful information on the intensity and frequency of dust events. For example, if we consider an extreme case in which dust aerosol is persistently lofted in the atmosphere. Then climatological DAOD would be the same as conditional DAOD in this case, because dust aerosol is always present in the satellite observations. Whereas, if dust events occur quite infrequently then the conditional DAOD would be much larger than climatological DAOD.

Therefore, in regions with a large difference between climatological and conditional DAOD (3$^{rd}$ column in Figure 6), dust activities are highly episodic or occur in relatively small scales, so that there are many cloud-free dust-free cases detected by CALIOP.

**Reply:** Thanks for the comment. In Figure 7 in the revised manuscript, cloud fraction during the winter (DJF) in the Northwest of Pacific Ocean region in the 1$^{st}$ row and 4$^{th}$ column is greater than 0.9. In this circumstance, we expect that MODIS DAOD is larger than CALIOP DAOD. The 1$^{st}$ row and 2$^{nd}$ column shows that MODIS DAOD is similar or even smaller than CALIOP DAOD. This is an exception.

**Reply:** This is a great point. Thanks.

We further analyze DAOD interannual variability over Taklamakan Desert to check if the statement here is reasonable. The figure below shows that MODIS DAOD in Taklamakan Desert

peaks in middle spring (April) and decreases in summer. Therefore, MODIS doesn't capture the secondary maximum dust activity over Taklamakan, the high DAOD over NWP retrieved from MODIS observations in summer therefore is mainly attributed to cloud contamination.

[Figure]

Q19. Line 480-495: it seems the authors suggest a true DAOD should fall between CALIOP and MODIS. If so, please explicitly state this in the text.
**Reply:** Yes, thanks for the suggestion. In Section 3 of the revised manuscript, after comparing our DAOD with AERONET COD, we conclude that it is highly probable that the true DAOD falls between MODIS and CALIOP DAOD.

Q20. Line 523-524: broad East Asian region (ESA defined in Figure 4): I can't find ESA in Figure 4. Please clarify.
**Reply:** Thanks. The typo was corrected.

Q21. Line 531: change "DOAD" to "DAOD".
**Reply:** Done. Thanks

Q22. Line 535: I don't think it is "much weaker".
**Reply:** Yes, we change 'much weaker' to 'weaker'. We think the correlation of DAOD is weaker for MODIS because on one hand correlation coefficient is smaller for MODIS (R=0.53) than CALIOP (R=0.6), on the other hand P-value for MODIS (p=0.07) is higher than the threshold 0.05, which means the trend is not statistically significant.
In the revised manuscript, we changed the sentence as 'In spring, the correlation of DAOD from two regions is good based on CALIOP ($R = 0.6, p = 0.03$), while a weaker correlation ($R = 0.53, p = 0.07$) was found based on MODIS.'

Q23. Line 538: please explain a little bit why EVI, MERRA2 near surface (at 10m) wind speed and precipitation are reliable for this analysis. Probably cite some references which have already demonstrated this.
**Reply:** Here we cited three papers (Qian et al. 2002, Kurosaki & Mikami, 2003, Lee & Sohn, 2011) to indicate the dependence of dust events on precipitation, surface wind speed and vegetation, respectively.

**Reply:** (p>0.05) was added. Thanks.

**Reply:** DPR (Depolarization ratio) was added in the revised manuscript.

**Reply:**

To quantify the uncertainty caused by DPR selection, we also calculated DAOD in the lowest ($\delta_d$=0.30 and $\delta_{nd}$=0.07) and the highest ($\delta_d$=0.20 and $\delta_{nd}$=0.02) dust fraction scenarios. The uncertainty induced by DPR is region dependent (Figure S6). The uncertainty is much lower in dust dominant regions than other regions. The averaged uncertainty for regions with DAOD>0.05 is 20%, while the averaged uncertainty for other regions is 38%.

The DAOD uncertainty induced from DPR is updated in the revised manuscript. The DAOD uncertainty map and uncertainty definition are added in the supplement as Figure S6.

[Figure]

Figure S6. 2007~2019 Seasonal mean DAOD uncertainties induced by DPR assumptions. For each season in each grid, DAOD uncertainty is defined as $\frac{(DAOD^{high}-DAOD^{low})/2}{DAOD^{mean}}$%, where $DAOD^{high}$ is derived from high dust scenario with $\delta_d = 0.20$ and $\delta_{nd} = 0.02$, $DAOD^{low}$ is derived from low dust scenario with $\delta_d = 0.30$ and $\delta_{nd} = 0.07$, $DAOD^{mean}$ is the average of the two scenarios.

Reply: We realized that this is not applied to this study. Our study is focusing on monthly mean climatological DAOD. Therefore, we did our own analysis to compare our monthly mean DAOD with AERONET monthly mean coarse mode AOD (COD).

First, we evaluated our monthly mean MODIS- and CALIOP-based DAOD product by comparing with AERONET monthly mean coarse mode AOD (COD) from 2007 to 2019. We found that

MODIS DAOD is statistically higher than AERONET COD by 26.7% over land and 18.5% over ocean, while CALIOP DAOD is lower than AERONET COD by 27.9% over land and 35% over ocean. We suggest that the true DAOD are highly probable to fall between MODIS and CALIOP DAOD retrievals. Furthermore, referring to the methodology proposed by Sayer et al. 2013, we estimated that the absolute expected error of MODIS DAOD is $0.65 \times DAOD_M + 0$ over land and $0.50 \times DAOD_M + 0$ over ocean, the absolute expected error of CALIOP DAOD is $0.52 \times DAOD_C + 0.02$ over land and $0.54 \times DAOD_C + 0.02$ over ocean. This part of analysis is added in Section 3.2 in the revised manuscript.

Q28. Line 640: it is possible that CALIOP misses some dust events. But this study is based on long-term statistics and the impacts should be eliminated if the observation number are substantial large. Do the authors suggest the impacts are not negligible in this study?
**Reply:** This is a great point. Thanks for pointing this out.
In the revised manuscript, we removed this statement.

Q29. Line 664: add "during 2007-2019" after "DAOD".
**Reply:** Done. Thanks

Q30. Line 780: avoid using "decadal trend" here as there are only 13 years indeed.
**Reply:** 'decadal trend' was changed to 'interannual trend'. Thanks.

Q31. Line 782: the symbol "+" can't be clearly seem.
**Reply:** The figure was updated with a larger '+'sign.

Q32. Line 805: it is also helpful to put together the time series of EVI vs DAOD, wind vs DAOD, precipitation vs DAOD with the year for x-axis. This can be put in the supplement for references.
**Reply:** The time series figure was added in supplement (Figure S10).

[Figure]

**Figure S10.** Inter-spring series of EVI, surface wind speed and precipitation along with inter-spring series of DAOD from MODIS (red curves) and CALIOP (blue curves). R is Pearson's linear correlation coefficient between each variables and time series. Positive R indicates the variable increase with time, and vice versa.

Q33. Lines 935-938: a duplicated reference.
**Reply:** Corrected. Thanks!

References:

Anderson, T. L., Wu, Y., Chu, D. A., Schmid, B., Redemann, J., & Dubovik, O. (2005). Testing the MODIS satellite retrieval of aerosol fine-mode fraction. *Journal of Geophysical Research D: Atmospheres*, *110*(18), 1–16. https://doi.org/10.1029/2005JD005978

Eck, T. F., Holben, B. N., Reid, J. S., Dubovik, O., Smirnov, A., O'Neill, N. T., … Kinne, S. (1999). Wavelength dependence of the optical depth of biomass burning, urban, and desert dust aerosols. *Journal of Geophysical Research Atmospheres*, *104*(D24), 31333–31349. https://doi.org/10.1029/1999JD900923

Gasteiger, J., Wiegner, M., Groß, S., Freudenthaler, V., Toledano, C., Tesche, M., & Kandler, K. (2011). Modelling lidar-relevant optical properties of complex mineral dust aerosols. *Tellus, Series B: Chemical and Physical Meteorology*, *63*(4), 725–741. https://doi.org/10.1111/j.1600-0889.2011.00559.x

Gelaro, R., McCarty, W., Suárez, M. J., Todling, R., Molod, A., Takacs, L., … Zhao, B. (2017). The modern-era retrospective analysis for research and applications, version 2 (MERRA-2). *Journal of Climate*, *30*(14), 5419–5454. https://doi.org/10.1175/JCLI-D-16-0758.1

Gkikas, A., Proestakis, E., Amiridis, V., Kazadzis, S., Di Tomaso, E., Tsekeri, A., … Pérez Garciá-Pando, C. (2021). ModIs Dust AeroSol (MIDAS): A global fine-resolution dust optical depth data set. *Atmospheric Measurement Techniques*, *14*(1), 309–334. https://doi.org/10.5194/amt-14-309-2021

Hsu, N. C., M. J. Jeong, C. Bettenhausen, A. M. Sayer, R. Hansell, C. S. Seftor, J. Huang, and S. C. Tsay. 2013. "Enhanced Deep Blue Aerosol Retrieval Algorithm: The Second Generation." *Journal of Geophysical Research: Atmospheres* 118(16):9296–9315

Järvinen, E., Kemppinen, O., Nousiainen, T., Kociok, T., Möhler, O., Leisner, T., & Schnaiter, M. (2016). Laboratory investigations of mineral dust near-backscattering depolarization ratios. *Journal of Quantitative Spectroscopy and Radiative Transfer*, *178*, 192–208. https://doi.org/10.1016/j.jqsrt.2016.02.003

Kaufman, Y. J., Koren, I., Remer, L. A., Tanré, D., Ginoux, P., & Fan, S. (2005). Dust transport and deposition observed from the Terra-Moderate Resolution Imaging Spectroradiometer (MODIS) spacecraft over the Atlantic Ocean. *Journal of Geophysical Research D: Atmospheres*, *110*(10), 1–16. https://doi.org/10.1029/2003JD004436

Kok, J. F., Ridley, D. A., Zhou, Q., Miller, R. L., Zhao, C., Heald, C. L., … Haustein, K. (2017). Smaller desert dust cooling effect estimated from analysis of dust size and abundance. *Nature Geoscience*, *10*(4), 274–278. https://doi.org/10.1038/ngeo2912

Kurosaki, Y., & Mikami, M. (2003). Recent frequent dust events and their relation to surface wind in East Asia. *Geophysical Research Letters*, *30*(14). https://doi.org/10.1029/2003GL017261

Lee, E. H., & Sohn, B. J. (2011). Recent increasing trend in dust frequency over Mongolia and Inner Mongolia regions and its association with climate and surface condition change. *Atmospheric Environment*, *45*(27), 4611–4616. https://doi.org/10.1016/j.atmosenv.2011.05.065

Mielonen, T., Arola, A., Komppula, M., Kukkonen, J., Koskinen, J., De Leeuw, G., & Lehtinen, K. E. J. (2009). Comparison of CALIOP level 2 aerosol subtypes to aerosol types derived from AERONET inversion data. *Geophysical Research Letters*, *36*(18). https://doi.org/10.1029/2009GL039609

O'Neill, N. T., Eck, T. F., Smirnov, A., Holben, B. N., & Thulasiraman, S. (2003). Spectral discrimination of coarse and fine mode optical depth. *Journal of Geophysical Research: Atmospheres*, *108*(17). https://doi.org/10.1029/2002jd002975

Proestakis, E., Amiridis, V., Marinou, E., Georgoulias, A. K., Solomos, S., Kazadzis, S., … Van Der A, R. J. (2018). Nine-year spatial and temporal evolution of desert dust aerosols over South and East Asia as revealed by CALIOP. *Atmospheric Chemistry and Physics*, *18*(2), 1337–1362. https://doi.org/10.5194/acp-18-1337-2018

Pu, B., & Ginoux, P. (2018). How reliable are CMIP5 models in simulating dust optical depth? *Atmospheric Chemistry and Physics*, *18*(16), 12491–12510. https://doi.org/10.5194/acp-18-12491-2018

Qian, W., Quan, L., & Shi, S. (2002). Variations of the dust storm in China and its climatic control. *Journal of Climate*, *15*(10), 1216–1229. https://doi.org/10.1175/1520-0442(2002)015<1216:VOTDSI>2.0.CO;2

Ridley, A. D., Heald, L. C., Kok, F. J., & Zhao, C. (2016). An observationally constrained estimate of global dust aerosol optical depth. *Atmospheric Chemistry and Physics*, *16*(23), 15097–15117. https://doi.org/10.5194/acp-16-15097-2016

Sayer, A. M., Hsu, N. C., Bettenhausen, C., & Jeong, M. J. (2013). Validation and uncertainty estimates for MODIS Collection 6 "deep Blue" aerosol data. *Journal of Geophysical Research Atmospheres*, *118*(14), 7864–7872. https://doi.org/10.1002/jgrd.50600

Yu, H. B., Chin, M., Bian, H. S., Yuan, T. L., Prospero, J. M., Omar, A. H., … Zhang, Z. B. (2015). Quantification of trans-Atlantic dust transport from seven-year (2007-2013) record of CALIPSO lidar measurements. *Remote Sensing of Environment*, *159*, 232–249. https://doi.org/10.1016/j.rse.2014.12.010

Yu, H., Chin, M., Remer, L. A., Kleidman, R. G., Bellouin, N., Bian, H., & Diehl, T. (2009). Variability of marine aerosol fine-mode fraction and estimates of anthropogenic aerosol component over cloud-free oceans from the Moderate Resolution Imaging Spectroradiometer (MODIS). *Journal of Geophysical Research Atmospheres*, *114*(10). https://doi.org/10.1029/2008JD010648

Yu, H., Yang, Y., Wang, H., Tan, Q., Chin, M., Levy, R., … Shi, Y. (2020). Interannual Variability and Trends of Combustion Aerosol and Dust in Major Continental Outflows Revealed by MODIS Retrievals and CAM5 Simulations During 2003–2017. *Atmospheric Chemistry and Physics Discussions*, 1–38. https://doi.org/10.5194/acp-2019-621

---

## Author Comment (AC3)

**Responds to CC1**

Dear authors,
While your study is of high interest, I have the feeling that you miss some important literature on the same subject and moreover, many of the aspects mentioned should be revised, focusing specifically on the following points that I see from a first reading:

**Reply:** We appreciate that you spent time reading our paper. An item-to-item reply to your comments is provided below.

Before addressing your comments/questions below, we would like to first provide a summary of the major revisions made to the manuscript:

- We added more detailed discussion regarding MODIS DAOD retrieval methodologies over ocean and land in section 2.2.
- We compared our MODIS- and CALIOP-based DAOD with values reported in previous studies based on MODIS and CALIOP, respectively. The comparison is added to the revised manuscript as section 3.1. For MODIS DAOD comparison, we compare our results with previous studies in both global and regional scales; For CALIOP, there isn't global CALIOP-based DAOD retrievals to compare our result with, therefore, the comparison is limited to regional scale. Overall, these comparisons suggest that our results are in reasonable agreement with previous studies, except for Voss and Evan 2020 over ocean (which can be explained by the use of different parameterization schemes).
- We evaluated our monthly mean MODIS- and CALIOP-based DAOD product by comparing with AERONET monthly mean coarse mode AOD (COD) from 2007 to 2019. We found that MODIS DAOD is statistically higher than AERONET COD by 26.7% over land and 18.5% over ocean, while CALIOP DAOD is lower than AERONET COD by 27.9% over land and 35% over ocean. This may suggest that the true DAOD probably fall between MODIS and CALIOP DAOD retrievals. Furthermore, by following the methodology proposed by Sayer et al. 2013, we estimated that the absolute expected error of MODIS DAOD is $0.65 \times DAOD_M + 0$ over land and $0.50 \times DAOD_M + 0$ over ocean, the absolute expected error of CALIOP DAOD is $0.52 \times DAOD_C + 0.02$ over land and $0.54 \times DAOD_C + 0.02$ over ocean. This analysis was added in section 3.2.

After these revisions, we think the paper is much improved and more focused, although the general conclusions still hold.

Q1. Differences between the CALIOP and MODIS global DODs are large. Is there any explanation about this discrepancy? Please note that the MODIS-derived global DOD is substantially higher than those reported in most of the recently published works (e.g., Ridley et al., 2016; Voss and Evan, 2020; Gkikas et al., 2021). A description is needed on how the global averages have been computed for both sensors. Do you acknowledge any weighting factors based on the grid cell surface area? According to Levy et al. (2009), the approach for the calculation of the global DOD is quite critical (see Fig. 5). Summarizing, I recommend including a table providing the corresponding global DODs given by relevant studies (relied either on observations or models) in order to check (and discuss) the consistency of your findings.

**Reply:** Thanks for the suggestion. We added a Table 2 in the revised manuscript (also shown below) to compare our DAOD retrievals with values reported in previous studies and discussed reasons for the differences.

Table 2. Compare global mean DAOD retrievals in this study with some relevant studies (Note the definition of global scope is different for different studies).

| Region | | DAOD@550nm | Reference |
|---|---|---|---|
| 90°S~90°N | Global | 0.03±0.005 | Ridley et al. 2016 Use multiple satellite platforms, in-situ AOD observations and four global models |
| 90°S~90°N | Global | 0.033 | Gkikas et al 2021 Use AOD from Aqua MODIS and DOD-to-AOD ratio from MERRA2 |
| 50°S~60°N | Over Ocean | 0.03±0.06 | Voss and Evan 2020 Over Ocean: use method in Kaufman et al 2005 Over Land: use method in Ginoux et al. 2012 |
| | Over Land | 0.1 | |
| 60°S~60°N | Over Ocean | 0.055, 0.020 | This Study MODIS-based, CALIOP-based DAOD (To calculate global mean DAOD for scope 90°S~90°N, we assume zero DAOD outside of region 60°S~60°N. We weight each grid-cell surface area into ocean, land and global DAOD average) |
| | Over Land | 0.103, 0.068 | |
| 90°S~90°N | Global | 0.057, 0.028 | |

For global scale comparison, the (new) Table 2 in the revised manuscript lists the global mean DAOD from previous studies and our study. Ridley et al. 2016 used multiple satellite platforms (MODIS and MISR), in-situ AOD observations and four global models to estimate global mean DAOD over 2004 ~ 2008. Gkikas et al. 2021 used AOD from Aqua MODIS and DOD-to-AOD ratio from MERRA2 to estimate global mean DAOD over 2003~2017. In contrast, as shown in Table 2 our MODIS-based global (90°S~90°N) DAOD is 0.057. However, difference in the global mean DAOD values from these studies should be expected as we use different methodology. In particular, both of aforementioned studies used model simulations to aid their global DAOD estimate, while our estimates are completely based on observations (More precisely, DAOD of the scope 60°S~60°N are completely based on observations, while outside of the scope, DAOD is assumed to be zero). In contrast, Voss and Evan 2020 (referred to VE20) used similar methods to our MODIS-based methodology and limited the global scope to 50°S~60°N, this is directly comparable to our global (60°S~60°N) mean MODIS DAOD values listed in Table 2. Below we focused on explaining the difference between our MODIS-based DAOD and values reported in Voss and Evan 2020.

As shown in (new) Table 2 of the revised manuscript, our DAOD based on MODIS over land (DAOD=0.103) is almost identical with that in VE20 (DAOD=0.1). Over ocean, our MODIS-based result (DAOD=0.055) is significantly larger than VE20 (DAOD=0.03). As we mentioned before, VE20 and our MODIS-based DAOD retrieval used the similar method. However, different parameters are used in the two MODIS over ocean retrieval methodologies, which is the main reason causing the non-negligible difference in our over-ocean mean DAOD. As shown in Eq (2) and Eq (3) in the revised manuscript, $f_c, f_d, f_m$ are required to estimate DAOD. We use MODIS over ocean retrievals to determine $f_c, f_d, f_m$, while Voss and Evan 2020 determine those parameters based on AERONET stations dominated by each aerosol type. We believe the use of different parameters in the estimation of DAOD over ocean is the main reason causing the difference between the two studies.

As we explained in the supplementary materials, after combining Eq (2) and Eq. (3) in the manuscript, we get the following equation for DAOD over ocean:

$$\tau_d = \frac{(\tau - \tau_m)f_c + \tau_m f_m - \tau f}{f_c - f_d} \ , \tag{1}$$

, where $f_c, f_d, f_m$ are the fine mode fraction of combustion, dust and marine aerosols, respectively. The values for these parameters used in our study and in VE20 are listed in the Table S4 of the supplementary material. It turns out that we used significantly larger $f_c$ and $f_m$, while a slightly smaller $f_d$, in comparison with VE20. Because the derived DAOD is positively proportional to these parameters, the use of larger $f_c$ and $f_m$, is probably the reason for a larger DAOD in our study.

Moreover, for regional scale comparison, we compared regional mean DAOD in Ridley et al. 2016 and Proestakis et al. 2018 with our MODIS and CALIOP results, respectively.

As we discussed before, the global mean DAOD in Ridley et al. 2016 may differ from our results due to the different retrieval methodology. To compare with Ridley et al. 2016, we selected the same 14 dust-laden regions provided in their paper (see their Figure 1) and plotted our DAOD results with the values reported in their Table 3. As aforementioned, in Ridley et al. 2016 the DAOD in these dust-laden regions is based on AERONET measurements and satellite retrievals, and therefore more comparable with our results. The comparison plots are provided in the Figure S2 and S3 of the supplementary material. Overall, our MODIS-based DAOD agrees very well with their results. Note that in their method, only in these dust laden regions the DAOD is constrained by observations (MODIS, MISR and AERONET) while the rest of the world is based on model simulation. Therefore, the comparison indicates that the two studies are in good agreement in terms of MODIS-based DAOD.

Table 1 in Proestakis et al., 2018 provides domain mean DAOD for six regions in Asia based on CALIOP observations. We selected the same 6 regions in East Asian and compared the regional mean DAOD between the two studies. As shown in Figure S4 and S5 of the supplementary material, Proestakis et al. 2018 are in excellent agreement with our CALIOP-based DAOD.

Overall, these comparisons suggest that our results are in reasonable agreement with previous studies, except for VE20 over ocean (which can be explained by the use of different parameterization schemes). On the other hand, the comparison results also reveal that MODIS-

based is generally larger than CALIOP-based DAOD (See Figure S3 and S5 the supplementary materials). But the two methods were not systematically compared in previous studies, which is an important motivation of this study.

The description of the way we calculated global mean DAOD is added in caption of Table 4. Since Earth is a sphere, grid-cell surface area decreases toward the poles. We weight each grid-cell surface area into ocean, land and global DAOD average.

Q2. The manuscript could greatly benefit by previous studies that have performed similar analysis. For instance, the authors mention the climatological and conditional dust products, which have been introduced for the first time in Marinou et al., (2017) and then applied on Proestakis et al., (2018). No discussion or comparison is presented in the manuscript. Moreover, the separation methodology used in the manuscript has been extensively implemented in the framework of EARLINET (e.g. Tesche et al., 2009, 2011; Ansmann et al., Ansmann et al., 2011). Furthermore, Amiridis et al., (2013) introduced for the first time the depolarization-based separation methodology on CALIPSO. However, there is no reference or discussion on this study as well! Given that all the aforementioned studies are available in the literature, which are the innovative aspects of the present study?

**Reply:** Thanks for the suggestions and references. In the revised paper, we made the following modifications

First, we cited Marinou et al. 2017 and Proestakis et al. 2018 when introducing our conditional DAOD product in section 4. As discussed in Q1, the comparison with previous studies such as Ridley et al. 2016 and Proestakis et al. 2018 of regional mean DAOD were added in Section 3.1.

Second, we add a few sentences in section 2.1 about the depolarization-based dust separation algorithm that include the mentioned references: 'The depolarization-based dust separation algorithm is based on the method developed by Shimizu et al. (2004), Hayasaka et al. (2007) and Tesche et al. 2009. The algorithm has been implemented in the framework of surface lidar network such as European Aerosol Research Lidar Network (EARLINET) (Ansmann et al. 2011) and also applied to CALIOP observations (Yu et al., 2012; Amiridis et al. 2013; Yu et al., 2015a).'. Third, we would like to clarify that the innovative aspects of this study include:

(a) The previous depolarization-based dust separation based on CALIOP observations are mostly regional studies. While our study extends to a global scale.

(b) We systematically compare depolarization-based (shape-based) DAOD from CALIOP with size-based DAOD from MODIS and discuss their differences.

(c) We further investigate DAOD interannual variability and trends in major dust source and outflow regions based on two DAOD retrievals.

Q3. Lines 105-109: Please update the information based on the final paper version of Gkikas et al. (2021) in which the MODIS-Aqua Collection 6.1 data, over the period 2003-2017, have been used.

**Reply:** Thanks. The information was updated in the revised manuscript as 'Gkikas et al. 2021 developed a global fine resolution (0.1° x 0.1°) DAOD dataset for the period 2006-2017 by scaling MODIS retrieved Collection 6.1 Aerosol Optical Depth (AOD) with the DAOD-to-AOD ratios provided by MERRA-2 (Modern-Era Retrospective analysis for Research and Applications, Version 2) reanalysis (Gelaro et al., 2017).'

Q4. Lines 251-264: A short description of the applied techniques for the derivation of DOD is needed, based on MODIS, over continental and marine regions. How much feasible is to discriminate mineral particles from sea-salt over oceans relying only on size parameters? It is not clear to me how you can separate dust from sea-salt over land using a very high single scattering albedo (almost equal to 1; similar to those recorded for sea-salt particles) and ignoring its spectral variation. Moreover, how much reliable the Ångström exponent is above land (see Section 4.4.5 in Levy et al. (2013))? Are you using only Deep Blue retrievals over land? In this case, how do you discriminate dust aerosols from other types when the Dark Target algorithm it is applied?

**Reply:** To answer this question, we provide a more detailed description for our MODIS-based dust retrieval in the revised manuscript (i.e., Section 2.2).

For MOIDS over-ocean DAOD retrieval, an approach was developed in previous studies to separate DAOD from other types of aerosol by using aerosol optical depth ($\tau$) and fine mode fraction (f) retrieved from MODIS Dark Target retrieval over ocean. Both $\tau$ and $f$ refer to properties at 550 nm. In this approach, both $\tau$ and fine-mode AOD ($f\tau$) are assumed to be composed of marine aerosol, dust and combustion aerosols, i.e.,

$$\tau = \tau_m + \tau_d + \tau_c \ , \tag{2}$$

$$f\tau = f_m\tau_m + f_d\tau_d + f_c\tau_c \ , \tag{3}$$

Where the subscripts m, d, and c represent marine aerosol, dust and combustion aerosol, respectively. Note that marine aerosol refers to all aerosols originated from ocean, including not only sea salt but also DMS-produced sulfate and organic aerosol Based on Eq. (2) and (3), $\tau_d$ can be calculated from MODIS-retrieved $\tau$ and $f$, with appropriate parameterizations for $f_m, f_d, f_c$ and $\tau_m$. More specifically, $f_m, f_d, f_c$ were determined from retrieved $f$ in selected regions and seasons for which a specific aerosol type dominates, $\tau_m$ was parameterized as a function of wind speed (details can be found in Kaufman et al., 2005; Yu et al., 2009, 2020). We don't use size parameters to discriminate dust from sea-salt.

Over land, MODIS aerosol properties including AOD, Ångström exponent, SSA are retrieved from the Deep Blue (DB) algorithm. Dark target aerosol products over land are not used in this study DAOD over land is derived from the AOD using one criterion based on size distribution (to distinguish fine and coarse modes) and the other criterion based on absorption (to distinguish between scattering sea salt and absorbing dust). To apply first criterion, we use the following formula established by  Anderson et al. 2005 using in-situ data:

$$COD_M = AOD \times (0.98 - 0.5089\alpha + 0.051\alpha^2) \ , \tag{1}$$

Where $\alpha$ is the Ångström exponent (a measure of the wavelength dependence of optical depth) which has been shown to be highly sensitive to particle size (Eck et al. 1999), $COD_M$ is the coarse mode fraction (aerodynamic diameters larger than $1\mu m$) of AOD retrieved from MODIS, with a contribution from absorbing (DAOD) and scattering aerosols (sea salt aerosol optical depth). This relationship is derived from the formula of Anderson et al. 2005 derived from in-situ data. The second criterion requires the single-scattering albedo at 470 nm to be less than 0.99 for the retrieval of DAOD (more details can be found in Pu and Ginoux, 2018). Nevertheless, marine aerosol would be negligible in broad continental regions except in coastal areas.

**Reply:** Thanks for the suggestion. However, this work focuses on a climatological monthly mean DAOD product on a global scale derived from CALIOP observations and its comparison with MODIS-based DAOD retrievals. Because we are using CALIOP nighttime data with high quality, it is challenging to collocate CALIOP with MODIS/Aqua.

**Reply:** We examined possible trends of dust optical depth on a global scale, in the dust belt and the major dust outflow regions. What we found was that in EAS and NWP, both MODIS and CALIOP showed statistically significant trends (see Table 5 in the revised manuscript). For other regions, either there is no statistically significant trend from two sensors or only one sensor shows a statistically meaningful trend. Therefore, we focused on understanding factors contributing to the dust trends in EAS and NWOP.

**Reply:** Thanks for the suggestion. This is an important question. However, we don't have all the information needed to quantify all the DAOD uncertainties as discussed in our manuscript, the choice of depolarization ratio (DPR) for dust aerosols and non-dust aerosols also introduces uncertainty in DAOD. The uncertainty induced by DPR is region dependent. We added a map plot Figure S6 in the supplementary material to show the uncertainty induced from the DPR assumption. However, the uncertainty source of MODIS- and CALIOP-based DAOD are from many sources, it's impossible for us to quantify all of them within one study. In this revised manuscript, we have assessed DAOD uncertainty through comparing satellite derived DAOD with AERONET observed coarse mode AOD

**Reply:** Thanks for the suggestion. We have evaluated our MODIS- and CALIOP-based monthly mean DAOD by comparing with AERONET monthly mean coarse mode optical depth (COD) from 2007 to 2019 and put our analysis in the revised manuscript (i.e., section 3.2). We found that MODIS DAOD is statistically higher than AERONET COD by 26.7% over land and 18.5% over ocean, while CALIOP DAOD is lower than AERONET COD by 27.9% over land and 35% over ocean. We suggest that the true DAOD may fall between MODIS and CALIOP DAOD retrievals.

Furthermore, referring to the methodology proposed by Sayer et al. 2013, we estimated that the absolute expected error of MODIS DAOD is 0.65×DAOD$_M$+0 over land and 0.50×DAOD$_M$+0 over ocean, the absolute expected error of CALIOP DAOD is 0.52×DAOD$_C$+0.02 over land and 0.54×DAOD$_C$+0.02 over ocean.

Q9. Table 1: Are you using the spectral SSAs or only the values at 470 nm?
**Reply:** Thanks for pointing out this inconsistency. In over-land MODIS DAOD retrieval, we require SSA at 470 nm to be less than 0.99 to separate dust from sea salt. The information in Table 1 was corrected.

Q10. In the manuscript, dust is distinguished from non-dust aerosols based on particle shape information (i.e., the use of particulate depolarization ratio) for CALIOP. However, the particulate depolarization ratio in L2 is too noisy, showing values for dust, dusty marine, polluted dust aerosol subtypes from negative up to 1.0 and above (see figure below).

[Figure]

Moreover, approximately 11% of all dust, dusty marine, polluted dust aerosol subtypes have particulate depolarization ratios < 0.05. Since in the methodology the dust, dusty marine, polluted dust aerosol subtypes are assumed mixtures of dust and non-dust components, how do the authors treat the negative and larger-than-one particulate depolarization cases in their Quality Assurance procedure? Do the authors consider the dusty aerosol mixtures of particulate depolarization ratio lower than 0.05 as non-dust mixtures? Which are the uncertainties introduced in the final dust product by these values? Please quantify.
**Reply:** We understand that CALIOP observations of depolarization ratio are quite noisy at their native resolutions. In our study, if CALIOP observed particulate depolarization ratio (DPR) <0, then we make it to be 0, if it is >1, then we make it to be 1. In our approach, we don't use CALIOP aerosol type information. We check all the detected aerosol features and use the observed depolarization ratio to separate dust from non-dust aerosol. The figure on the right shows the relationship between dust fraction ($f_d$) and CALIOP observed DPR. The red curve is for high dust scenario, which is the results of combination of $\delta_d = 0.2$ and $\delta_{nd} = 0.02$. The blue curve is for low dust scenario, which is the results of combination of $\delta_d = 0.3$ and

$\delta_{nd} = 0.07$. The black line is the mean of high and low dust scenarios. The DAOD derived for different dust scenarios (high, low and mean) are all included in our product.

To quantify the uncertainty caused by DPR selection, we also calculated DAOD in the lowest ($\delta_d = 0.30$ and $\delta_{nd} = 0.07$) and the highest ($\delta_d = 0.20$ and $\delta_{nd} = 0.02$) dust fraction scenarios. The uncertainty induced by DPR is region dependent (see Figure S6 in the supplementary material). The uncertainty is much lower in dust dominant regions than other regions. The averaged uncertainty for regions with DAOD>0.05 is 20%, while the averaged uncertainty for other regions is 38%.

Q11. The authors provide a CALIPSO-based dust product, based on the particulate depolarization ratio, applied to L2 backscatter coefficient profiles. Based on the manuscript it is not clear whether the methodology is applied only on the dust, dusty marine, and polluted dust aerosol subtypes, and not at the other types (e.g. elevated smoke, marine, ...) at the 60m aerosol layer. Or whether an average over consecutive 60m layers is computed to remove noise. Please provide more in-depth description of the selected methodology. Moreover, which is the effect of the identified aerosol subtype misclassification on the dust product? Many important studies are mentioned by the authors (e.g. Burton et al., 2013), however the effect of the misclassification on the dust product needs discussion and quantification.

**Reply:** In our CALIOP-based DAOD retrieval, the methodology was applied to all CALIOP detected cloud-free aerosol layers. Therefore, this DAOD retrieval does not depend on CALIOP standard aerosol subtype classification.

Q12. Based on the methodology, the dust, dusty marine, and polluted dust aerosol mixtures are distinguished into a dust and a non-dust component. Thus, at the end, there are three types of backscatter coefficient: (1) the initial backscatter coefficient of non-dust mixtures (e.g. elevated smoke, ...), (2) the dust backscatter coefficient of the separated dust component, and (3) the remaining backscatter coefficient of the separation, the non-dust component. According to my understanding the extinction coefficient of (1) does not change since the methodology is not applied to non-dust mixtures. Regarding the case (2), a uniform global Lidar Ratio (LR) is implemented to calculate the dust extinction coefficient. However, the authors do not discuss the case three (3), regarding the remaining backscatter coefficient of the non-dust component. For the calculation of the non-dust extinction coefficient component, the authors should identify the non-dust aerosol subtype in the dusty aerosol mixture, in order to assign a proper LR. The authors have not provided a detailed explanation. Since the AOD is then computed by the integration of the extinction coefficient profile, the authors should either provide a solid justification of the non-dust aerosol- subtype assignment including quantification the corresponding uncertainties, or to avoid using the new AOD and the corresponding Sections, after the intermediate dust separation.

**Reply:** Thanks for the comment.

In our CALIOP-based DAOD retrieval, our methodology was applied to all types of aerosol layers. Therefore, there are two types of backscatter coefficient: (1) the backscatter coefficient for dust component (2) the backscatter coefficient for non-dust component. Meanwhile, in addition to backscatter coefficient profile, we also have extinction coefficient profile for total aerosol from CALIOP level 2 product. The extinction coefficient profile here is used to calculate total AOD.

In this study, we focus on dust aerosol, therefore, we assign a global uniform LR (44sr, which is consistent with LR used in CALIOP standard product) for dust component to calculate dust extinction coefficient vertical profile. Then DAOD could be calculated by integrating dust extinction coefficient profile for each column.

For non-dust component, it is not our focus in this study. We did not assign any LR for non-dust components, which is impossible (marine aerosol and smoke pollution can differ in LR by about a factor of 3). The total AOD shown in this study is calculated by integrating the total extinction profile from CALIOP L2 product.

Q13. It is not properly discussed, how the averaging extinction coefficient procedure is computed, prior to integration for the DAOD. According to Amiridis et al. (2013) and Tackett et al. (2018), the methodology should follow first a "per-overpass" averaging within a specific grid, and accordingly integration of the mean profile, calculated by all overpasses in the grid. However, the methodology followed by the authors is not clear in this point. Please discuss, and in case a different methodology is provided justify the selected approach or revise accordingly.

**Reply:** Thanks for the suggestion. We added more discussion on how we average extinction coefficient in Section 2. Below is our updated description.

In each 2º (latitude) ×5º (longitude) grid, at each altitude, dust backscatter coefficient is derived by multiplying CALIOP total backscatter coefficient with the calculated $f_d$. Then we apply LR to the dust backscatter coefficient to get the dust extinction coefficient for each overpass. The monthly mean dust extinction coefficient is calculated at each altitude for grids with larger than 5 samples within the month. Then DAOD is calculated by integrating the monthly mean dust extinction coefficient profile for each grid.

Q14. The manuscript would greatly benefit by introducing tables of the Quality Assurance procedures, applied to both CALIPSO and MODIS, including the corresponding literature related to each filter.

**Reply**: Thanks for the suggestion. The table containing the Quality Assurance procedures are added in the supplementary material as Table S1.

Table S1. Summary of Quality Assurance procedures in CALIOP- and MODIS-based DAOD retrievals.

|  | Quality Assurance (references) |
|---|---|
| CALIOP | (a) Select cloud-free columns or columns with high-level optically thin clouds using CALIOP L2 cloud layer product. (Yu et al. 2015a)
 (b) Use CAD score between –90 and –100 (Yu et al. 2019)
 (c) Use EXT_QC values of 0, 1, 18, and 16 (Winker et al. 2013) |
| MODIS (Ocean) | QAC>=0 (Levy et al. 2013), AOD <0 was excluded |
| MODIS (Land) | Retrieved aerosol properties with a standard deviation less than 0.15 among 10x10 pixels are assumed cloud free and are flagged with the highest quality flag (QA=3). Here we use products of QA=3 following the recommendation of Hsu et al. (2013) |

Q15. What I am missing in the study is a validation intercomparison against ground reference lidar instruments to validate the profiles acquired (e.g. EARLINET/ACTRIS), or even an intercomparison against dust models.

**Reply:** Thanks for the suggestion. We added a section for comparison with AERONET retrievals. Although a comparison with EARLINET/ACTRIS or against dust models would be nice, it is beyond the scope of this study and may be pursued in future research. The focus of this paper is to

assess consistency and inconsistency between CALIOP shape-based DAOD and MODIS size-based DAOD.

**Reply:** Thanks for the comment. There are multiple uncertainty sources, we do not think it is possible to quantify each of them in this study. We agree that the presence of pollen, volcanic ash, and cube-like sea salt particles, all with elevated DPR, would have led to an overestimate of CALIOP DAOD. This is now discussed in the paper. However, it is impossible to quantify the overestimate.

For the uncertainty analysis, in the revised manuscript, we discuss the uncertainty induced from LR assumption and DPR in Section 3.2.

This study assumes dust lidar ratio to be $44 \pm 9 \; sr$ at 532 nm, which is the value used in the CALIOP V4 product (Kim et al. 2018) and is comparable to previous studies and basically covers the range of typical dust lidar ratios. The $\pm 9 \; sr$ induces $\pm 20\%$ DAOD uncertainties as shown in the shaded area in Figure 9.

To quantify the uncertainty caused by DPR selection, we also calculated DAOD in the lowest ($\delta_d$=0.30 and $\delta_{nd}$=0.07) and the highest ($\delta_d$ =0.20 and $\delta_{nd}$=0.02) dust fraction scenarios. The uncertainty induced by DPR is region dependent (Figure S6). The uncertainty is much lower in dust dominant regions than other regions. The averaged uncertainty for regions with DAOD>0.05 is 20%, while the averaged uncertainty for other regions is 38%.

Moreover, as shown in the answer of Q8, we estimated the absolute expected error of our DAOD products by comparing with AERONET COD.

References:

Anderson, T. L., Wu, Y., Chu, D. A., Schmid, B., Redemann, J., & Dubovik, O. (2005). Testing the MODIS satellite retrieval of aerosol fine-mode fraction. *Journal of Geophysical Research D: Atmospheres*, *110*(18), 1–16. https://doi.org/10.1029/2005JD005978

Eck, T. F., Holben, B. N., Reid, J. S., Dubovik, O., Smirnov, A., O'Neill, N. T., … Kinne, S. (1999). Wavelength dependence of the optical depth of biomass burning, urban, and desert dust aerosols. *Journal of Geophysical Research Atmospheres*, *104*(D24), 31333–31349. https://doi.org/10.1029/1999JD900923

Gasteiger, J., Wiegner, M., Groß, S., Freudenthaler, V., Toledano, C., Tesche, M., & Kandler, K. (2011). Modelling lidar-relevant optical properties of complex mineral dust aerosols. *Tellus, Series B: Chemical and Physical Meteorology*, *63*(4), 725–741. https://doi.org/10.1111/j.1600-0889.2011.00559.x

Gelaro, R., McCarty, W., Suárez, M. J., Todling, R., Molod, A., Takacs, L., … Zhao, B. (2017). The modern-era retrospective analysis for research and applications, version 2 (MERRA-2). *Journal of Climate*, *30*(14), 5419–5454. https://doi.org/10.1175/JCLI-D-16-0758.1

Gkikas, A., Proestakis, E., Amiridis, V., Kazadzis, S., Di Tomaso, E., Tsekeri, A., … Pérez Garciá-Pando, C. (2021). ModIs Dust AeroSol (MIDAS): A global fine-resolution dust optical depth data set. *Atmospheric Measurement Techniques*, *14*(1), 309–334. https://doi.org/10.5194/amt-14-309-2021

Hsu, N. C., M. J. Jeong, C. Bettenhausen, A. M. Sayer, R. Hansell, C. S. Seftor, J. Huang, and S. C. Tsay. 2013. "Enhanced Deep Blue Aerosol Retrieval Algorithm: The Second Generation." *Journal of Geophysical Research: Atmospheres* 118(16):9296–9315

Järvinen, E., Kemppinen, O., Nousiainen, T., Kociok, T., Möhler, O., Leisner, T., & Schnaiter, M. (2016). Laboratory investigations of mineral dust near-backscattering depolarization ratios. *Journal of Quantitative Spectroscopy and Radiative Transfer*, *178*, 192–208. https://doi.org/10.1016/j.jqsrt.2016.02.003

Kaufman, Y. J., Koren, I., Remer, L. A., Tanré, D., Ginoux, P., & Fan, S. (2005). Dust transport and deposition observed from the Terra-Moderate Resolution Imaging Spectroradiometer (MODIS) spacecraft over the Atlantic Ocean. *Journal of Geophysical Research D: Atmospheres*, *110*(10), 1–16. https://doi.org/10.1029/2003JD004436

Kok, J. F., Ridley, D. A., Zhou, Q., Miller, R. L., Zhao, C., Heald, C. L., … Haustein, K. (2017). Smaller desert dust cooling effect estimated from analysis of dust size and abundance. *Nature Geoscience*, *10*(4), 274–278. https://doi.org/10.1038/ngeo2912

Kurosaki, Y., & Mikami, M. (2003). Recent frequent dust events and their relation to surface wind in East Asia. *Geophysical Research Letters*, *30*(14). https://doi.org/10.1029/2003GL017261

Lee, E. H., & Sohn, B. J. (2011). Recent increasing trend in dust frequency over Mongolia and Inner Mongolia regions and its association with climate and surface condition change. *Atmospheric Environment*, *45*(27), 4611–4616. https://doi.org/10.1016/j.atmosenv.2011.05.065

Levy, R. C., Mattoo, S., Munchak, L. A., Remer, L. A., Sayer, A. M., Patadia, F., and Hsu, N. C.: The Collection 6 MODIS aerosol products over land and ocean, Atmos. Meas. Tech., 6, 2989–3034, https://doi.org/10.5194/amt-6-2989-2013, 2013.

Mielonen, T., Arola, A., Komppula, M., Kukkonen, J., Koskinen, J., De Leeuw, G., & Lehtinen, K. E. J. (2009). Comparison of CALIOP level 2 aerosol subtypes to aerosol types derived from AERONET inversion data. *Geophysical Research Letters*, *36*(18).

https://doi.org/10.1029/2009GL039609

O'Neill, N. T., Eck, T. F., Smirnov, A., Holben, B. N., & Thulasiraman, S. (2003). Spectral discrimination of coarse and fine mode optical depth. *Journal of Geophysical Research: Atmospheres*, *108*(17). https://doi.org/10.1029/2002jd002975

Proestakis, E., Amiridis, V., Marinou, E., Georgoulias, A. K., Solomos, S., Kazadzis, S., … Van Der A, R. J. (2018). Nine-year spatial and temporal evolution of desert dust aerosols over South and East Asia as revealed by CALIOP. *Atmospheric Chemistry and Physics*, *18*(2), 1337–1362. https://doi.org/10.5194/acp-18-1337-2018

Pu, B., & Ginoux, P. (2018). How reliable are CMIP5 models in simulating dust optical depth? *Atmospheric Chemistry and Physics*, *18*(16), 12491–12510. https://doi.org/10.5194/acp-18-12491-2018

Qian, W., Quan, L., & Shi, S. (2002). Variations of the dust storm in China and its climatic control. *Journal of Climate*, *15*(10), 1216–1229. https://doi.org/10.1175/1520-0442(2002)015<1216:VOTDSI>2.0.CO;2

Ridley, A. D., Heald, L. C., Kok, F. J., & Zhao, C. (2016). An observationally constrained estimate of global dust aerosol optical depth. *Atmospheric Chemistry and Physics*, *16*(23), 15097–15117. https://doi.org/10.5194/acp-16-15097-2016

Sayer, A. M., Hsu, N. C., Bettenhausen, C., & Jeong, M. J. (2013). Validation and uncertainty estimates for MODIS Collection 6 "deep Blue" aerosol data. *Journal of Geophysical Research Atmospheres*, *118*(14), 7864–7872. https://doi.org/10.1002/jgrd.50600

Yu, H. B., Chin, M., Bian, H. S., Yuan, T. L., Prospero, J. M., Omar, A. H., … Zhang, Z. B. (2015). Quantification of trans-Atlantic dust transport from seven-year (2007-2013) record of CALIPSO lidar measurements. *Remote Sensing of Environment*, *159*, 232–249. https://doi.org/10.1016/j.rse.2014.12.010

Yu, H., Chin, M., Remer, L. A., Kleidman, R. G., Bellouin, N., Bian, H., & Diehl, T. (2009). Variability of marine aerosol fine-mode fraction and estimates of anthropogenic aerosol component over cloud-free oceans from the Moderate Resolution Imaging Spectroradiometer (MODIS). *Journal of Geophysical Research Atmospheres*, *114*(10). https://doi.org/10.1029/2008JD010648

Yu, H., Yang, Y., Wang, H., Tan, Q., Chin, M., Levy, R., … Shi, Y. (2020). Interannual Variability and Trends of Combustion Aerosol and Dust in Major Continental Outflows Revealed by MODIS Retrievals and CAM5 Simulations During 2003–2017. *Atmospheric Chemistry and Physics Discussions*, 1–38. https://doi.org/10.5194/acp-2019-621

---

## Author Comment (AC4)

**Supplementary material**

**Global Dust Optical Depth Climatology Derived from CALIOP and MODIS Aerosol Retrievals on Decadal Time Scales: Regional and Interannual Variability**

**Qianqian Song1,2, Zhibo Zhang1,2,\*, Hongbin Yu3, Paul Ginoux4, Jerry Shen3,**

- 1. Physics Department, UMBC, Baltimore, Maryland, USA
- 2. Joint Center of Earth Systems Technology, UMBC, Baltimore, Maryland, USA
- 3. Earth Sciences Division, NASA Goddard Space Flight Center, Greenbelt, Maryland, USA
- 4. NOAA Geophysical Fluid Dynamics Laboratory, Princeton, New Jersey, USA.

Correspondence to: Zhibo Zhang, Zhibo.Zhang@umbc.edu

**who worked as summer intern at NASA Goddard Space Flight Center during June–August 2020.**

**DAOD derived from CALIOP and MODIS observations**

---

## Author Response (AR2)

**Responds to Reviewers**

**Reviewer 1**

This paper derived two global monthly mean dust aerosol optical depth climatological datasets based on CALIOP and on MODIS observations. However, I still have some major comments on the methodology. I suggest this paper to be published after major revision.

Major comments: (1) For MODIS dataset, two different separation methods of DAOD are used over land and ocean. However, the separation method seems to overestimate the DAOD over ocean. For example, over the Northern Atlantic, the DAOD should decrease when dust transports away from the dust source region. However, the new dataset shows an artificial increase from land to ocean near the coastal region. The authors should check carefully that the different methods over land and ocean used in this paper can provide consistent results. Furthermore, the new MODIS dataset shows quite unrealistic large DAODs over land and ocean in high latitudes of Northern Hemisphere in some seasons. What's the reason for this?

**Reply:** Thanks for the questions and comments.

Indeed, because the two MODIS AOD products, i.e., Dark Target (DT) over ocean (and dark vegetation surface) and Deep Blue (DB) over land, are developed based on different methodologies and implemented by different groups, they have discontinuity problem in transition regions, e.g., along the coastline region. This issue has been reported in previous studies such as Yu et al. 2021. For MODIS dataset in this study, DAOD is derived from DT AOD retrieval over ocean and DB AOD retrieval over land. Therefore, the discontinuity in MODIS AOD would propagate to DAOD.

In addition to the DT vs. DB issue, we also used different methods to derive DAOD from AOD over land and ocean, which adds another level of uncertainty or maybe additional discontinuity. To identify the discontinuity problem in this study and understand the potential causes, we further investigate the variation of meridional mean AOD and DAOD over latitude range 5°N-30°N (Figure 1) and 20°S-5°N (Figure 2) along longitude from 30°E to 60°W.

Figure 1 represents the longitudinal variation of the meridional mean TAOD and DAOD over Northern Africa between 5°N and 30°N with the stars indicating land-ocean transition point. In this region, the MODIS and CALIOP AOD have a similar pattern and there is not an obvious discontinuity in TAOD and DAOD over the coastal region.

However, when checking the TAOD and DAOD in the Southern Africa between 20°S and 5°N, we noted an abrupt increase in MODIS-based TAOD and DAOD in coastal region (indicated by star makers) for all seasons. As shown in Figure 2, the meridional mean TAOD based on CALIOP (blue line in Figure2) in this region has a peak over land (i.e., east of the star). In contrast, the MODIS based TAOD peaks at the transition point, which seems to be a result of the abovementioned DB-to-DT discontinuity problem.

The problem is even more severe for the DAOD (lower panel of Figure2); summer (JJA) and fall (SON) seasons in particular. This is a sharp and unrealistic peak around the transition point (10°E). Note that CALIOP derived DAOD is drastically different and rather smooth.

Cloud contamination in MODIS aerosol retrievals is likely the cause of this issue. As discussed in the manuscript and pointed in many previous studies, MODIS aerosol retrieval is more susceptible to cloud contamination. Cloud contamination can lead to an overestimation of TAOD but underestimation of FMF, therefore, overestimation in DAOD (see line 605~611 in the manuscript for details). To support this point, we derived the frequency of dust and cloud coexisting within 5-km CALIOP aerosol and cloud profile products. To this end, we first identify the dusty profiles using the lidar depolarization-based method described in the manuscript (see Section 2.1). Then we search for the presence of any cloud in the same profile. The dust-cloud coexisting frequency is defined as ratio of the number of dusty profiles with cloud presence with respect to the total dusty profiles.

Figure 3 shows meridional mean dust-cloud co-existing frequency in the two regions, i.e., Northern (5°N ~ 30°N) and Southern (20°S~5°N) Africa. It is evident that clouds are more frequently coexisting with dust over Southern African coastal region than Northern Africa. As such, TAOD and DAOD retrieval based on MODIS are more susceptible to cloud contamination over coastal region in Southern Africa than in Northern Africa, which causes more obvious discontinuity in TAOD and DAOD in Southern Africa.

Regarding to unrealistic large DAOD in Northern Hemisphere (NH) high latitudes in some seasons: for over-ocean DAOD, cloud contamination and limited sampling are major issues. For example, in summertime and over North Pacific, we found that DAOD was too high. We know that the cloud fraction is high in summer, which limits the aerosol retrieval and introduces cloud contamination issue.

[Figure]

Figure 1. Meridional mean TAOD (upper panel) and DAOD (lower panel) over latitude range 5°-30°N. The star locates at the longitude that represents the most frequent coastal line longitude over the latitude interval. In other words, it is transition point between MODIS DB AOD over land and MODIS DT AOD over ocean.

[Figure]

Figure 2. The same as Figure 1, except for latitude range 20°S-5°N.

[Figure]

Figure 3. Meridional mean frequency of coexisting cloud and dust cases of latitude interval 20°S-5°N and 5°N-30°N based on CALIPSO retrievals. The star locates at the longitude that represents the most frequent coastal line longitude over the latitude interval. More specifically, the star on the right represents coastal line over latitude 20°S-5°N, the one on the left represents coastal line over latitude 5°N-30°N.

Clearly, both the cloud contamination problem and DB-DT discontinuity problem can cause significant errors and uncertainties in the TAOD and DAOD. We pointed this out explicitly as major limitation of MODIS-based results, in the discussion of Figure 5 (Section 4.1) of the revised manuscript. However, fixing these issues is far beyond the scope of this study and frankly out of our capability. We will leave them to the operational MODIS AOD retrieval teams.

(2) For CALIOP dataset, some mean dust extinction coefficient profiles in Figure 11 show quite sharp decreasing trend in the low levels over some regions, for example the profiles below ~0.5km in Figure 11(b). Could the authors provide mean dust extinction coefficient profile results from other literatures for comparison? And, as those profiles are all shown as low as 0km, how did the authors handle with bins below surface in the averaging process?

Reply: Thanks for the comment.
We compare dust extinction vertical profile from this study with the results reported in Yu et al. 2015. By comparing the top panel (bottom panel) of Figure 4 in this document with the blue curves in Figure 5 (Figure 6) bottom panel in Yu et al. 2015, we could see that dust extinction

from two studies agree very well. The minor difference could be due to the different data version used in the two studies.

We checked dust extinction for altitude lower than 0km, it is always zero. Therefore, we show dust extinction vertical profile starting from 0km in the manuscript, which is also consistent with that in Yu et al. 2015.

[Figure]

Figure 4. Vertical profile of 10°N-20°N averaged dust extinction coefficient (km⁻¹) at 15°W, 35°W and 75°W in MAM (top panel) and JJA (bottom panel) 2012. Dotted blue cure, solid blue curve and black thick curve are for low, high, and mean dust scenarios. Dust optical depth in mean dust scenarios and in (low, high) scenarios are noted in the plots.

**Reviewer 2**

Review of the revised paper "Global dust optical depth climatology derived from CALIOP and MODIS aerosol retrievals on decadal time scales: regional and interannual variability" by Song et al. for ACPD. The authors have gone through great lengths to address the (voluminous) comments from myself, the other reviewer, and the community comment by Vassilis Amiridis. In particular, the paper is much improved by a more detailed discussion of the differences between the CALIPSO- and MODIS-based DOD estimates, differences of their estimates with literature studies over the past few years, and a comparison against AERONET data. In addition, there is a more detailed discussion of sources of uncertainty. As such, the authors addressed my main concerns well, and I recommend publication after considering two small remaining comments. Remaining concerns:
- This paper presents two new products, namely a DOD product based on CALIPSO and a DOD product based on MODIS. It looks like only the CALIPSO product is freely available to the community. Could you also make the MODIS-based DOD product freely available?

**Reply:** Thanks for the comments. We've uploaded MODIS-based DAOD products $(1^{\circ} \times 1^{\circ}$ and $5^{\circ} \times 2^{\circ}$ resolution) to the google drive. Now both CALIPSO and MODIS DAOD products are publicly available at 'https://drive.google.com/drive/folders/1aQVupe7govPwR6qmsqUbR4fJQsp1DBCX?usp=sharing'. The link is provided in Data availability part.

Line 403: please cite these previous studies.
**Reply:** In this part, we added a reference (Kar et al. 2018) regarding CALPSO Version 4 nighttime calibration at 532nm. In addition, we corrected the citation 'Chamara et al. 2017' to 'Rajapakshe et al. 2017'.

**Reviewer 3**

Suggestions for revision or reasons for rejection (will be published if the paper is accepted for final publication)
1. The authors have used "interannual trends" in several places within the manuscript. I suggest not using this term as it is not commonly used.
**Reply:** Thanks. We choose to use more commonly used terms such as 'interannual variability' and 'trend' in the revised manuscript.

2. Lines 765-766: Please provide some information on EVI, MERRA-2 wind speed and precipitation to show their reliability for your analysis. In particular, a reanalysis of MERRA-2 is not a real observation. Please explain why they are not biased. Probably cite some references if you can find some.
**Reply:** Thanks for the comment. We add a reference Carvalho et al. 2019 for the assessment of MERRA-2 surface wind speeds and Reichle et al. 2017 for MERRA-2 land surface precipitation assessment.

3. Table 5: I think "inter-seasonal trend" is not a commonly-used word. Please check.
**Reply:** Thanks. In the revised manuscript, we use 'DAOD trend' in Table 5.

**References**

Carvalho, D. (2019). An Assessment of NASA's GMAO MERRA-2 Reanalysis Surface Winds, *Journal of Climate*, *32*(23), 8261-8281. Retrieved Aug 5, 2021, from https://journals.ametsoc.org/view/journals/clim/32/23/jcli-d-19-0199.1.xml

Kar, J., Vaughan, M. A., Lee, K.-P., Tackett, J. L., Avery, M. A., Garnier, A., Getzewich, B. J., Hunt, W. H., Josset, D., Liu, Z., Lucker, P. L., Magill, B., Omar, A. H., Pelon, J., Rogers, R. R., Toth, T. D., Trepte, C. R., Vernier, J.-P., Winker, D. M., and Young, S. A.: CALIPSO lidar calibration at 532 nm: version 4 nighttime algorithm, Atmos. Meas. Tech., 11, 1459–1479, https://doi.org/10.5194/amt-11-1459-2018, 2018.

Reichle, R. H., Liu, Q., Koster, R. D., Draper, C. S., Mahanama, S. P. P., & Partyka, G. S. (2017). Land Surface Precipitation in MERRA-2, *Journal of Climate*, *30*(5), 1643-1664. Retrieved Aug 5, 2021, from https://journals.ametsoc.org/view/journals/clim/30/5/jcli-d-16-0570.1.xml

Yu, H., Tan, Q., Zhou, L., Zhou, Y., Bian, H., Chin, M., Ryder, C. L., Levy, R. C., Pradhan, Y., Shi, Y., Song, Q., Zhang, Z., Colarco, P. R., Kim, D., Remer, L. A., Yuan, T., Mayol-Bracero, O., and Holben, B. N.: Observation and modeling of a historic African dust intrusion into the Caribbean Basin and the southern U.S. in June 2020, Atmos. Chem. Phys. Discuss. [preprint], https://doi.org/10.5194/acp-2021-73, in review, 2021.

Yu, H. B., M. Chin, H. S. Bian, T. L. Yuan, J. M. Prospero, A. H. Omar, L. A. Remer, D. M. 1405 Winker, Y. K. Yang, Y. Zhang, and Z. B. Zhang. 2015a. "Quantification of Trans-Atlantic Dust Transport from Seven-Year (2007-2013) Record of CALIPSO Lidar Measurements." *Remote Sensing of Environment* 159:232–49.